# Learning Two-Player Mixture Markov Games: Kernel Function Approximation and Correlated Equilibrium

**Chris Junchi Li**[*]
University of California, Berkeley
junchili@berkeley.edu

**Dongruo Zhou**[*]
University of California, Los Angeles
drzhou@cs.ucla.edu

**Quanquan Gu**
University of California, Los Angeles
qgu@cs.ucla.edu

**Michael I. Jordan**
University of California, Berkeley
jordan@cs.berkeley.edu

## Abstract

We consider learning Nash equilibria in two-player zero-sum Markov Games with nonlinear function approximation, where the action-value function is approximated by a function in a Reproducing Kernel Hilbert Space (RKHS). The key challenge is how to do exploration in the high-dimensional function space. We propose a novel online learning algorithm to find a Nash equilibrium by minimizing the duality gap. At the core of our algorithms are upper and lower confidence bounds that are derived based on the principle of optimism in the face of uncertainty. We prove that our algorithm is able to attain an $O(\sqrt{T})$ regret with polynomial computational complexity, under very mild assumptions on the reward function and the underlying dynamic of the Markov Games. We also propose several extensions of our algorithm, including an algorithm with a Bernstein-type bonus that can achieve a tighter regret bound, and another algorithm for model misspecification that can be applied to neural network function approximation.

## 1 Introduction

Multi-agent reinforcement learning (MARL) has been the focus of research across a range of research communities [Shapley, 1953, Littman, 1994]. The case of two-player Markov Games (MG) has been of particular interest. In this case, two players select their actions based on the current state simultaneously and independently. One player (the max-player) aims to maximize the return based on the reward provided by the environment, while the other (the min-player) aims to minimize it. A series of recent results have established polynomial sample complexity/regret guarantees that depend on the cardinality of state/action spaces for two-player zero-sum MGs [Wei et al., 2017, Bai and Jin, 2020, Bai et al., 2020, Liu et al., 2021, Jia et al., 2019, Sidford et al., 2020, Cui and Yang, 2021, Lagoudakis and Parr, 2002, Perolat et al., 2015, Pérolat et al., 2016a,b, 2017, Jin et al., 2021b].

Meanwhile, most of the recent successful applications of MARL deal with *large state/action spaces* that may be continuous or a fine-grained discretization of a continuous space. Examples include GO [Silver et al., 2016], autonomous driving [Shalev-Shwartz et al., 2016], TexasHold'em poker [Brown and Sandholm, 2019], and AlphaStar for the game Starcraft [Vinyals et al., 2019]. In order to tackle problems with large state/action spaces, researchers have designed MARL algorithms based on *function approximation* which approximate the original high-dimensional value

---

[*]Equal contribution.

36th Conference on Neural Information Processing Systems (NeurIPS 2022).

function/policy by a function approximator. For instance, Xie et al. [2020] and Chen et al. [2022] studied RL for two-player zero-sum MGs with *linear function approximation*, where it is assumed that there are a set of *linear features* that span the transition kernel and reward function spaces. In contrast to RL with linear function approximation, RL with *nonlinear function approximation* (e.g., kernel and neural network approximation) aims to take advantage of the superior representational power of nonlinear function compared to linear parameterizations. For example, Jin et al. [2022] studied neural-network-based RL in the setting of *MGs with low multi-agent Bellman eluder dimension*, obtaining algorithms that have polynomial dependence on the complexity of the underlying function class. Although this yields a strong theoretical guarantee, the algorithm that they propose is not computationally efficient due to the constructed highly nonconvex confidence sets. The following question is still open: *Can we design a computationally and statistically efficient RL algorithm for learning two-player Markov Games with nonlinear function approximation?*

In this paper, we give an affirmative answer to this question for a class of episodic Markov Games, dubbed *mixture Markov Games*, when using a nonlinear approximation function in a Reproducing Kernel Hilbert Space (RKHS). We propose a novel kernel-based MARL algorithmic framework for general episodic two-player zero-sum MGs which provides provable regret guarantees. We summarize the contributions of our work as follows:

- We propose a `KernelCCE-VTR` algorithm for two-player zero-sum MGs. In particular, at each episode, `KernelCCE-VTR` uses kernel function approximation to approximate the optimal value function and constructs corresponding confidence sets, following the *"Optimism-in-Face-of-Uncertainty"* principle [Abbasi-Yadkori et al., 2011] to select an action based on the current state. In contrast to algorithms in Jin et al. [2022], which construct implicit confidence sets that are in general computationally intractable, our algorithm `KernelCCE-VTR` crafts a computationally efficient exploration bonus based on the gram matrix of the kernel function.
- Under the assumption that the transition dynamics belongs to some RKHS, we show that our algorithm `KernelCCE-VTR` is able to find a Nash equilibrium of the game with a $\widetilde{O}(d_{\mathcal{F}} H^2 \sqrt{T})$ regret bound on the duality gap, where $H$ is the horizon, $T$ is the number of the episodes, and $d_{\mathcal{F}}$ represents the complexity of the function class $\mathcal{F}$. We also propose an extension of `KernelCCE-VTR` that utilizes *weighted kernel ridge regression* and a *Bernstein-type bonus* to achieve $\widetilde{O}(d_{\mathcal{F}} H^{3/2} \sqrt{T})$ regret. When $\mathcal{F}$ reduces to the $d$-dimensional linear function class, our regret reduces to $\widetilde{O}(d H^{3/2} \sqrt{T})$, which almost matches the lower bound in Chen et al. [2022].
- We also study the general case where the transition dynamics belongs to some RKHS up to a misspecification error. We show that our `KernelCCE-VTR` can achieve a similar regret as in the well-specified case. In particular, we study the neural network function approximation case which can be regarded as a special instance of the misspecified RKHS case and derive the corresponding regret bound.

**Notation.** We use lowercase letters to denote scalars, and lower and uppercase bold letters to denote vectors and matrices. We use $\| \cdot \|$ to indicate Euclidean norm, and for a semi-positive definite matrix $\mathbf{\Sigma}$ and any vector $\mathbf{x}$, $\|\mathbf{x}\|_{\mathbf{\Sigma}} := \|\mathbf{\Sigma}^{1/2}\mathbf{x}\| = \sqrt{\mathbf{x}^\top \mathbf{\Sigma} \mathbf{x}}$. For real $t$ and interval $[a, b]$, we use $\Pi_{[a,b]}[t]$ to indicate the projection of $t$ onto $[a, b]$, i.e. $\Pi_{[a,b]}[t] = \max(a, \min(b, t))$. For positive integer $N$ we sometimes define $[N] = \{1, \ldots, N\}$ for compactness. We also adopt the standard big-$O$ and big-$\Omega$ notations: say $a_n = O(b_n)$ if and only if there exists $C > 0, N > 0$, for any $n > N$, $a_n \leq C b_n$; $a_n = \Omega(b_n)$ if $a_n \geq C b_n$. The notations $\widetilde{O}$ and $\widetilde{\Omega}$ are adopted when the $C$ above hides a polylogarithmic factor.

## 2   Related Work

**Online RL with function approximation.** MARL with function approximation can be seen as an extension of RL with function approximation on MDPs. There are several lines of work studying RL with function approximation. The first line of work studies the so-called linear MDP which assumes the reward function and transition dynamics are linear functions of a feature mapping defined on the state and action spaces [Yang and Wang, 2020, Jin et al., 2020, Zanette et al., 2020]. These works propose model-free algorithms with sublinear regret on the number of episodes $K$. The second line of work studies the linear mixture MDP which assumes the transition kernel is a linear combination of several base models [Modi et al., 2020, Jia et al., 2020, Zhou et al., 2021b,a]. These studies proposed model-based RL algorithms that estimate the transition kernel with finite sample complexity

or sublinear regret guarantees. The third line of work studies general function approximation which assumes that either the value function or the transition kernel can be approximated by a general class of functions [Osband and Van Roy, 2014, Jiang et al., 2017, Sun et al., 2019, Wang et al., 2020, Yang et al., 2020, Du et al., 2021, Jin et al., 2021a]. Algorithms proposed in this line enjoy finite regret or sample complexity bounds that depend on some general complexity measures such as Eluder dimension [Russo and Van Roy, 2013, Osband and Van Roy, 2014], Bellman rank [Jiang et al., 2017], witness rank [Sun et al., 2019], information gain [Yang et al., 2020], bilinear class [Du et al., 2021] and Bellman eluder dimension [Jin et al., 2021a].

**Learning two-player MGs with function approximation.** There is a large body of literature on MARL for two-player MGs with function approximation. These works can be generally categorized into MARL with *linear function approximation* and MARL with *general function approximation*. For example, for linear function approximation, Xie et al. [2020] studied zero-sum simultaneous-move MGs where both the reward and transition kernel can be parameterized as linear functions of some feature mappings. They proposed an OMVI-NI algorithm with an $\widetilde{O}(\sqrt{d^3 H^3 T})$ regret, where $d$ is the number of the feature dimension, $H$ is the episode length and $T$ is the total number of rounds. Chen et al. [2022] studied the linear mixture MGs and proposed a nearly minimax optimal Nash-UCRL-VTR algorithm with an $\widetilde{O}(dH\sqrt{T})$ regret and an $\Omega(dH\sqrt{T})$ matching lower bound. In contrast to this work, our `KernelCCE-VTR` does not assume the underlying transition dynamic or reward function has a linear structure. For MARL with general function approximation, Jin et al. [2022] studied the two-player zero-sum MGs with low multi-agent Bellman Eluder dimension and proposed a "Golf with Exploiter" algorithm using a general function class. They showed their algorithm enjoys an $\widetilde{O}(H\sqrt{dK \log N})$ regret, where $d$ is the multi-agent Bellman eluder dimension, $K$ is the number of episodes. Huang et al. [2022] studied two-player MGs with a finite minimax Eluder dimension and proposed a method called ONEMG with an $\widetilde{O}(H\sqrt{dK \log N})$ regret, where $d$ is the minimax Eluder dimension. To obtain the desired function approximator, both Golf with Exploiter and ONEMG need to solve a constrained optimization problem, which is computationally intractable even in the linear function approximation setting. In contrast to Jin et al. [2022] and Huang et al. [2022], our proposed algorithms are computationally efficient and nearly optimal by using the Bernstein-type bonus. Qiu et al. [2021] also studied kernel function approximation for two-player MGs. However, there are two key differences between our work and theirs. First, Qiu et al. [2021] studied MGs where the expectation of the value function is in some RKHS; we, on the other hand, assume that the transition dynamics of the MG lie in an RKHS. Second, while the regret result in Qiu et al. [2021] depends on the covering number of the function space, our regret is *independent* of the covering number.

## 3 Preliminaries

In this section, we present the necessary definitions that will be adopted throughout the paper. Section 3.1 describes simultaneous-move games in the setting of zero-sum two-player Markov Games (MG) and recaps the concepts of equilibrium and duality gap that are employed in the game theory literature. Section 3.2 provides necessary definitions and notation for approximating action value function with functions belonging to a reproducing kernel Hilbert space (RKHS) via modeling the transition probability.

### 3.1 Two-player Markov Games

A simpler instance of Markov Games, referred to as turn-based games, can be seen as a special case of simultaneous-move games.[2] In a zero-sum two-player simultaneous-move Markov Game, the dynamical structure can be captured by an MG, denoted $(\mathcal{S}, \mathcal{A}_1, \mathcal{A}_2, r, \mathbb{P}, H)$, where $\mathcal{S}$ is the space of available states of the environment, $\mathcal{A}_1$ is the action space of the first player and $\mathcal{A}_2$ is the action space of the second player. $H$ is the time horizon representing the maximum step of each round of play. The reward function $r : \{r_h(x, a, b) : h \in [H]\}$ is a sequence of mappings from $\mathcal{S} \times \mathcal{A}_1 \times \mathcal{A}_2$ to $[-1, 1]$. The transition matrix $\mathbb{P} : \{\mathbb{P}_h(\cdot|x, a, b) : h \in [H]\}$ gives for each state actions triplet $(x, a, b)$ and at each time $h$ the stochastic response of the environment to the next $x' \in \mathcal{S}$. Here by "simultaneous move" we refer to the setting where at each round of game the two players $P_1$ and $P_2$ take actions $a \in \mathcal{A}_1, b \in \mathcal{A}_2$ simultaneously at a given state $x \in \mathcal{S}$, in contrast with the turn-based

---

[2]We present a discussion of the implications of our results for turn-based games in the supplementary materials.

game where $r_h$ and $\mathbb{P}_h$ are defined for a state-action pair $(x, a)$ where the action can be taken by either players. In the context of this paper, for simplicity of notation we let $\mathcal{A}_1 = \mathcal{A}_2 = \mathcal{A}$, while the results can be easily generalized to the case when $\mathcal{A}_1 \neq \mathcal{A}_2$. Similar definitions of a zero-sum two-player simultaneous-move episodic Markov Games can be found in Wei et al. [2017], Perolat et al. [2018], Xie et al. [2020].

In the above setting, two players $P_1$ and $P_2$ take actions according to their own strategies. We use $\pi := \{\pi_h\}_{h \in [H]}$ to denote the stochastic policy of $P_1$ and use $\nu := \{\nu_h\}_{h \in [H]}$ to denote the stochastic policy of $P_2$. We note that at time $h$, $\pi_h : \mathcal{S} \mapsto \Delta_{\mathcal{A}}$ maps the current state $x_h$ to a probability distribution of the actions, and similarly for $\nu_h$. Given two agents' policies, $\pi, \nu$, across $h$ steps, the state value function is defined as the expected total reward through $H$ steps where at step $h \in [H]$ player $P_1$ follows policy $\pi_h(\cdot|x_h)$ and player $P_2$ follows policy $\nu_h(\cdot|x_h)$,

$$V_h^{\pi,\nu}(x) := \mathbb{E}_{\pi,\nu}\left[\sum_{t=h}^{H} r_t(s_t, a_t, b_t) \,\middle|\, x_h = x\right], \quad V_{H+1}^{\pi,\nu}(x) := 0,$$

and where $V^{\pi,\nu}(x) := V_1^{\pi,\nu}(x)$. Note that the expectation is taken over all stochasticity in $\pi_h, \nu_h$ and $\mathbb{P}_h$. The action value function is defined as

$$Q_h^{\pi,\nu}(x, a, b) := \mathbb{E}_{\pi,\nu}\left[\sum_{t=h}^{H} r_t(x_t, a_t, b_t) \,\middle|\, x_h = x, a_h = a, b_h = b\right], \quad Q_{H+1}^{\pi,\nu}(x, a, b) := 0,$$

and $Q^{\pi,\nu}(x, a, b) := Q_1^{\pi,\nu}$. From the definition of two value functions, we observe that for any $x \in \mathcal{S}$, the state value function given policy pair $(\pi, \nu)$ is the expectation of the corresponding action value function

$$V_h^{\pi,\nu}(x) := \mathbb{E}_{(a,b)\sim(\pi,\nu)} Q_h^{\pi,\nu}(x, a, b),$$

where the expectation is taken over the action distribution induced by the policy pair. Throughout this paper, we use superscripts do denote the number of episodes and subscripts to denote the number of horizon steps.

**Nash equilibrium and duality gap.** In a zero-sum two-player Markov Game, $P_1$ wants to maximize the expected reward $V^{\pi,\nu}(x)$ via the choice of the policy $\pi$. On the contrary, $P_2$ wants to minimize $V^{\pi,\nu}(x)$ by properly choosing $\nu$. For fixed $\nu$, we define the best response policy with respect to $V$ and $\nu$ as $\mathrm{br}(\nu)$ and define $V_h^{*,\nu} := V_h^{\mathrm{br}(\nu),\nu}$ and $Q_h^{*,\nu} := Q_h^{\mathrm{br}(\nu),\nu}$, We define $V_h^{\pi,*} := V_h^{\pi,\mathrm{br}(\pi)}$ and $Q_h^{\pi,*} := Q_h^{\pi,\mathrm{br}(\pi)}$ similarly. A Nash equilibrium is a pair of policies $(\pi^*, \nu^*)$ that are the best response policy for each other, which we write as $V^{\pi^*,*}(x) = V^{\pi^*,\nu^*}(x) = V^{*,\nu^*}(x)$. For notational simplicity we write $V^* := V^{\pi^*,\nu^*}, Q^* := Q^{\pi^*,\nu^*}$. By definition of the best response policy, we obtain weak duality:

$$V_h^{\pi,*}(x) \leq V_h^*(x) \leq V_h^{*,\nu}(x).$$

For any policy pair $(\pi, \nu)$, we define the duality gap as $V_1^{*,\nu^t}(x_1^t) - V_1^{\pi^t,*}(x_1^t)$. We call a pair an $\epsilon$-*approximate Nash equilibrium (NE)* if $V_1^{*,\nu^t}(x_1^t) - V_1^{\pi^t,*}(x_1^t) \leq \epsilon$. We also define the *regret* in the MG setting as follows:

$$\mathrm{Regret}(T) := \sum_{t=1}^{T} V_1^{*,\nu^t}(x_1^t) - V_1^{\pi^t,*}(x_1^t).$$

**Coarse correlated equilibrium.** We introduce the *Coarse Correlated Equilibrium (CCE)* notion which will be used in our proposed algorithms. Given payoff matrices $Q_1, Q_2 : \mathcal{S} \times \mathcal{A} \times \mathcal{A} \mapsto \mathbb{R}$ and the state $x$, we define the CCE of the game as a joint distribution $\sigma$ on $\mathcal{A} \times \mathcal{A}$ satisfying:

$$\mathbb{E}_{(a,b)\sim\sigma}\left[Q_1(x, a, b)\right] \geq \mathbb{E}_{b\sim\mathcal{P}_2\sigma}\left[Q_1(x, a', b)\right], \quad \forall a' \in \mathcal{A}, \tag{3.1}$$

$$\mathbb{E}_{(a,b)\sim\sigma}\left[Q_2(x, a, b)\right] \leq \mathbb{E}_{a\sim\mathcal{P}_1\sigma}\left[Q_2(x, a, b')\right], \quad \forall b' \in \mathcal{A}, \tag{3.2}$$

where $\mathcal{P}_1\sigma$ denotes the marginal of $\sigma$ on the first coordinate (min-player) and $\mathcal{P}_2\sigma$ denotes the marginal of $\sigma$ on the second coordinate (max-player). We use $\texttt{FIND\_CCE}(Q_1, Q_2, x)$ to denote $\sigma$. When $\sigma$ can be written as a product of two policies over action space $\mathcal{A}$, it is a Nash equilibrium [Xie et al., 2020]. To compute a CCE given $Q_1, Q_2, x$, please see Appendix I.

## 3.2 Nonlinear Function Approximation by Reproducing Kernel Hilbert Spaces

For simplicity of notation, we use $z = (x, a, b)$ to denote a state-action-action triplet (or state-action tuple) in $\mathcal{Z} := \mathcal{S} \times \mathcal{A} \times \mathcal{A}$. An RKHS $\mathcal{H}$ with kernel $\mathbf{K}(\cdot, \cdot) : \mathcal{Z} \times \mathcal{Z} \mapsto \mathbb{R}$ is a general form of linear function class. Every RKHS $\mathcal{H}$ consists of functions on $\mathcal{Z}$, with a feature mapping $\phi : \mathcal{Z} \mapsto \mathcal{H}$, such that $\forall \boldsymbol{f} \in \mathcal{H}$ and $\forall z \in \mathcal{Z}$, $\boldsymbol{f}(z) = \langle \boldsymbol{f}, \phi(z) \rangle_{\mathcal{H}}$. The kernel $K$ is thus defined for every $x, y \in \mathcal{Z} \times \mathcal{Z}$ as $\mathbf{K}(x, y) = \langle \phi(x), \phi(y) \rangle_{\mathcal{H}}$. We call $\phi$ the *feature mapping* induced by the RKHS $\mathcal{H}$ with kernel $K$. In the following sections, we sometimes use $f^\top g$ as a simplification of $\langle \boldsymbol{f}, \boldsymbol{g} \rangle_{\mathcal{H}}$ when $\boldsymbol{f}, \boldsymbol{g} \in \mathcal{H}$. We make no distinction in notation between the vector inner product and the product $\langle \cdot, \cdot \rangle_{\mathcal{H}}$; the distinction can be read out from the nature of the two objects in the product. For every RKHS $\mathcal{H}$, there exists a natural eigenvalue decomposition in $\mathcal{L}^2(\mathcal{Z})$. RKHS approximation is a generalization of the linear function approximation of finite dimension $d$ which can be infinite-dimensional. In the following, we define the so-called *kernel mixture MG*, which can be regarded as an extension from the linear mixture MDP [Jia et al., 2020, Ayoub et al., 2020, Zhou et al., 2021a] and linear mixture MG [Chen et al., 2022] to their kernel counterpart.

**Kernel mixture MG.** In a kernel mixture MG model, we model the transition probability $\mathbb{P}_h(s'|z) : \mathcal{Z} \mapsto \Delta(\mathcal{S})$ as an element in an RKHS $\mathcal{H}$ with feature mapping $\phi(s'|z) : \mathcal{Z} \times \mathcal{S} \to \mathcal{H}$, such that for an unknown true parameter $\boldsymbol{\theta}_h^* \in \mathcal{H}$, $\mathbb{P}_h(s'|z) = \langle \phi(s'|z), \boldsymbol{\theta}_h^* \rangle_{\mathcal{H}}$ for all $s' \in \mathcal{S}$ and $z \in \mathcal{Z}$. A similar MG structure called kernel MG has been studied by Qiu et al. [2021], which assumes that the transition probability satisfies $\mathbb{P}_h(s'|z) = \langle \phi(z), \boldsymbol{\mu}_h(s') \rangle$ for some $\phi(\cdot), \boldsymbol{\mu}_h(\cdot) \in \mathcal{H}$. The single-agent MDP counterparts of kernel MGs and kernel mixture MGs are linear MDPs and linear mixture MDPs. Zhou et al. [2021b] have shown that linear MDPs and linear mixture MDPs are different classes of MDPs and one cannot be covered by each other. Following a similar argument, we can also show that kernel mixture MGs and kernel MGs are different classes of MGs and cannot imply each other.

At time $h$, for any estimate of the value function $V_h(\cdot) : \mathcal{S} \mapsto \mathbb{R}$, we note that the expectation of value function at time $h + 1$, $\mathbb{P}_h V_{h+1}$ is an element in the RKHS $\mathbb{P}_h V_{h+1}(z) = \langle \phi_{V_{h+1}}(z), \boldsymbol{\theta}_h^* \rangle_{\mathcal{H}}$, where $\phi_{V_{h+1}}(z) := \sum_{s' \in \mathcal{S}} V_{h+1}(s') \phi(s'|z)$ integrates the product of the feature mapping with the estimated value of $s'$ over $\mathcal{S}$. It is worth noting that the quantity $\phi_V(\cdot)$ plays an important role in previous linear mixture model-based algorithms [Jia et al., 2020, Ayoub et al., 2020, Zhou et al., 2021a, Chen et al., 2022]. We assume that for any bounded value function $V(\cdot) : \mathcal{S} \mapsto [-1, 1]$ and any $z \in \mathcal{Z}$, $\|\phi_V(z)\|_{\mathcal{H}} \leq 1$. Given that the reward function $r_h(z)$ is known, we obtain through the Bellman equation that

$$Q_h^{*,\nu}(\cdot) = r_h(\cdot) + (\mathbb{P}_h V_{h+1}^{*,\nu})(\cdot) = r_h(\cdot) + \left\langle \phi_{V_{h+1}^{*,\nu}}(\cdot), \boldsymbol{\theta}_h^* \right\rangle_{\mathcal{H}}, \tag{3.3}$$

$$Q_h^{\pi,*}(\cdot) = r_h(\cdot) + (\mathbb{P}_h V_{h+1}^{\pi,*})(\cdot) = r_h(\cdot) + \left\langle \phi_{V_{h+1}^{\pi,*}}(\cdot), \boldsymbol{\theta}_h^* \right\rangle_{\mathcal{H}}. \tag{3.4}$$

**Weighted kernel function.** In this work, we consider a general RKHS $\mathcal{H}$ and do not assume that we can access the feature mapping $\phi$ directly. Instead, we assume that we can access the *weighted kernel function* $\boldsymbol{k}_{V_1, V_2}(\cdot, \cdot)$, which is defined as follows:

**Definition 1.** For any function pairs $V_1, V_2 : \mathcal{S} \to [0, 1]$ which map states to real numbers, the weighted kernel function $\boldsymbol{k}_{V_1, V_2}(\cdot, \cdot)$ is defined as follows: $\forall z_1, z_2 \in \mathcal{Z}$,

$$\boldsymbol{k}_{V_1, V_2}(z_1, z_2) := \sum_{s_1, s_2 \in \mathcal{S}} V_1(s_1) V_2(s_2) \langle \phi(s_1|z_1), \phi(s_2|z_2) \rangle_{\mathcal{H}}.$$

It is easy to see from Definition 1 that

$$\boldsymbol{k}_{V_1, V_2}(z_1, z_2) = \left\langle \sum_{s_1 \in \mathcal{S}} V_1(s_1) \phi(s_1|z_1), \sum_{s_2 \in \mathcal{S}} V_2(s_2) \phi(s_2|z_2) \right\rangle_{\mathcal{H}} = \langle \phi_{V_1}(z_1), \phi_{V_2}(z_2) \rangle_{\mathcal{H}},$$

which suggests that the weighted kernel function $\boldsymbol{k}_{V_1, V_2}(\cdot, \cdot)$ indeed captures the interaction (in inner product relation) between $\phi_{V_1}(z_1)$ and $\phi_{V_2}(z_2)$. We assume that we can access an integration oracle that computes $\boldsymbol{k}_{V_1, V_2}(z_1, z_2)$ for any function $V_1, V_2$ and state-action tuples $z_1, z_2$ efficiently.

## 4 Algorithm

In this section, we introduce our value-targeted iteration algorithm for the zero-sum two-player Markov Game setting with RKHS function approximation. We follow the *value-targeted regression* framework and the confidence set design as in UCRL [Jia et al., 2020, Ayoub et al., 2020], and combine the CCE technique [Xie et al., 2020] to deal with the zero-sum sub-game brought by upper confidence bound (UCB) and lower confidence bound (LCB) value functions. These techniques enable us to adapt the results from the linear setting to the nonlinear RKHS regime [Chowdhury and Gopalan, 2017, Yang et al., 2020, Zhou et al., 2020] to get a structure-dependent regret bound that is both computationally simple and statistically efficient.

---

**Algorithm 1** KernelCCE-VTR

---

1: **Input:** bonus parameter $\beta > 0$.
2: **for** episode $t = 1, 2, \ldots, T$ **do**
3:     **for** step $h = H, H-1, \ldots, 1$ **do**
4:         Calculate $\overline{Q}_h^t(\cdot, \cdot, \cdot), \underline{Q}_h^t(\cdot, \cdot, \cdot)$ as in (4.3)

5:         Let $\sigma_h^t(\cdot) = \texttt{FIND\_CCE}(\overline{Q}_h^t, \underline{Q}_h^t, \cdot)$
6:         Let $\overline{V}_h^t(\cdot) = \mathbb{E}_{(a,b) \sim \sigma_h^t(\cdot)} \overline{Q}_h^t(\cdot, a, b)$ and
        $\underline{V}_h^t(\cdot) = \mathbb{E}_{(a,b) \sim \sigma_h^t(\cdot)} \underline{Q}_h^t(\cdot, a, b)$
7:     **end for**
8:     Receive initial state $x_1^t$
9:     **for** step $h = 1, 2, \ldots, H$ **do**
10:        Sample $(a_h^t, b_h^t) \sim \sigma_h^t(x_h^t)$
11:        $P_1$ takes action $a_h^t$, $P_2$ takes action $b_h^t$
12:        Observe next state $x_{h+1}^t$
13:     **end for**
14: **end for**

---

To find an equilibrium $(\pi^*, \nu^*)$ of the value function $V_1^{\pi,\nu}(x_1)$, we design an algorithm using value-targeted regression (VTR) and upper/lower confidence bound-based exploration. As the min-player aims to minimize the value function while the max-player targets to maximize the value function, we use the upper confidence bound to encourage the exploration of the max-player and use the lower confidence bound to encourage the exploration of the min-player. Thus we need to define two value functions for the min/max-players respectively, i.e., $\overline{Q}_h^t, \underline{Q}_h^t, \overline{V}_h^t, \underline{V}_h^t$, where we adopt the overline notation for the over-estimation by the max-player and the underline notation for the under-estimation by the min-player. In the following, we only describe how to estimate the value functions for the max-player, while the value functions for the min-player can be estimated analogously. At each round of the game, we solve the following ridge regression problem for minimizing the Bellman error:

$$\overline{\theta}_h^t = \min_{\theta \in \mathcal{H}} \sum_{\tau=1}^{t-1} \left[ \overline{V}_{h+1}^\tau(x_{h+1}^\tau) - \left\langle \phi_{\overline{V}_{h+1}^\tau}(z_h^\tau), \theta \right\rangle_{\mathcal{H}} \right]^2 + \lambda \|\theta\|_{\mathcal{H}}^2. \tag{4.1}$$

Note that in (4.1), $\overline{V}_{h+1}^\tau$ only depends on the previous trajectories $\left( x_i^j, a_i^j, b_i^j : j \in [\tau-1], i \in [H] \right)$. We denote the corresponding $\sigma$-algebra as $\mathcal{F}_{\tau-1}$, and thus we have $\overline{V}_{h+1}^\tau \in \mathcal{F}_{\tau-1}$. As each $\overline{V}_{h+1}^\tau(x_{h+1}^\tau)$ can be seen as a stochastic sample of $(\mathbb{P}_h \overline{V}_{h+1}^\tau)(z_h^\tau)$, the regularized regression problem of the max-player in (4.1) can be seen as solving a linear bandit problem with context $\phi_{\overline{V}_{h+1}^\tau}(z_h^\tau)$, reward function $(\mathbb{P}_h \overline{V}_{h+1}^\tau)(z_h^\tau)$ and noise term $\overline{V}_{h+1}^\tau(x_{h+1}^\tau) - (\mathbb{P}_h \overline{V}_{h+1}^\tau)(z_h^\tau)$. From the solution to the ridge regression problem (4.1), we can define the upper/lower confidence bound of the action-value functions $Q_h^{*,\nu}, Q_h^{\pi,*}$ respectively. For the simplicity of notation, we define the vectors $\overline{\Psi}_h^t := \left( \phi_{\overline{V}_{h+1}^1}(z_h^1), \ldots \phi_{\overline{V}_{h+1}^{t-1}}(z_h^{t-1}) \right)^\top \in \mathcal{H}^{t-1}$.

For a positive parameter $\beta_t > 0$ that will be chosen in later analysis, the confidence region centered at $\overline{\theta}_h^t$ in the RKHS $\mathcal{H}$ is defined as

$$\overline{\mathcal{C}}_h^t = \left\{ \theta : \sqrt{\lambda \left\| \theta - \overline{\theta}_h^t \right\|_{\mathcal{H}}^2 + \left\| \left\langle \overline{\Psi}_h^t, \theta - \overline{\theta}_h^t \right\rangle_{\mathcal{H}} \right\|^2} \leq \beta_t \right\}, \tag{4.2}$$

We omit the definition of $\underline{\mathcal{C}}_h^t$ which is an analogue of Eq. (4.2) by changing all overline symbols to underline ones. Based on the confidence regions, we construct an optimistic/pessimistic estimate of $Q_h^{*,\nu}$ as

$$\overline{Q}_h^t := \Pi_{[-H,H]} \left[ r_h + \max_{\theta \in \overline{\mathcal{C}}_h^t} \left\langle \phi_{\overline{V}_{h+1}^t}, \theta \right\rangle_{\mathcal{H}} \right], \quad \underline{Q}_h^t := \Pi_{[-H,H]} \left[ r_h + \min_{\theta \in \underline{\mathcal{C}}_h^t} \left\langle \phi_{\underline{V}_{h+1}^t}, \theta \right\rangle_{\mathcal{H}} \right],$$
$$\tag{4.3}$$

where $\Pi_{[-H,H]}$ is the projection operator onto $[-H, H]$, which is by definition the range of value functions. For the convenience of conducting an induction argument we define $\overline{V}_{H+1}^t = \underline{V}_{H+1}^t = 0$, and also $V_{H+1}^{\pi,\nu}(x) = 0$ and $V_{H+1}^{*,\nu^t} = V_{H+1}^{\pi^t,*} = 0$, since there is no more future steps starting from $h = H + 1$. Given the estimation of $\overline{Q}_h^t, \underline{Q}_h^t$, the next step is to estimate the corresponding state value functions $\overline{V}_h^t, \underline{V}_h^t$. We utilize the $\texttt{FIND\_CCE}$ algorithm introduced recently in Xie et al. [2020] to find a coarse-correlated equilibrium of the payoff pair $(\overline{Q}_h^t(z), \underline{Q}_h^t(z))$.

**Computational efficiency.** By substituting the closed-form solutions to the maximization/minimization problems in (4.3), we can derive the analytic-form for $\overline{Q}_h^t$ and $\underline{Q}_h^t$. Take $\overline{Q}_h^t$ as an example, we have

$$\overline{Q}_h^t(z) = \Pi_{[-H,H]}\left[ r_h(z) + \overline{\boldsymbol{k}}_h^t(z)^\top (\overline{\mathbf{K}}_h^t + \lambda \mathbf{I})^{-1} \overline{\boldsymbol{y}}_h^t + \beta_t \cdot \overline{w}_h^t(z) \right], \tag{4.4}$$

where the gram matrix $\overline{\mathbf{K}}_h^t$ and vector-valued function $\overline{\boldsymbol{k}}_h^t$ are defined as

$$\overline{\mathbf{K}}_h^t = \left(\overline{\boldsymbol{\Psi}}_h^t\right)\left(\overline{\boldsymbol{\Psi}}_h^t\right)^\top \in \mathbb{R}^{(t-1)\times(t-1)}, \quad \overline{\boldsymbol{k}}_h^t = \left(\overline{\boldsymbol{\Psi}}_h^t\right)\boldsymbol{\phi}_{\overline{V}_{h+1}^t}(z) = \left(\boldsymbol{k}_{\overline{V}_{h+1}^i, \overline{V}_{h+1}^t}(z_h^i, z)\right)_i \in \mathbb{R}^{t-1}.$$

Also, we have $\overline{\boldsymbol{y}}_h^t := \left[\overline{V}_{h+1}^1(x_h^1), \ldots, \overline{V}_{h+1}^{t-1}(x_h^{t-1})\right]^\top$ and $\overline{w}_h^t(z) = \lambda^{-1/2}\left[\boldsymbol{k}_{\overline{V}_{h+1}^t, \overline{V}_{h+1}^t}(z, z) - \overline{\boldsymbol{k}}_h^t(z)^\top(\overline{\mathbf{K}}_h^t + \lambda\mathbf{I})^{-1}\overline{\boldsymbol{k}}_h^t(z)\right]^{1/2}$. Therefore, by the assumption that the weighted kernel function $\boldsymbol{k}_{V_1, V_2}$ can be evaluated efficiently, and $\overline{Q}_h^t$ and $\underline{Q}_h^t$ can also be computed efficiently. Furthermore, given $\overline{Q}_h^t$ and $\underline{Q}_h^t$, $\texttt{FIND\_CCE}$ can also be implemented efficiently [Xie et al., 2020]. Thus, Algorithm 1 is computationally efficient.

# 5 Main Results

In this section, we present the regret bound of our algorithm for the kernel mixture Markov Game. Recall that for the linear function class, the regret upper bound is characterized by the dimension of the linear function, the horizon of the game, and the number of episodes [Chen et al., 2022]. Our analysis in the RKHS function approximation setting aligns with the linear function approximation setting when $\mathbf{K}(z, z') = \boldsymbol{\phi}(z)^\top \boldsymbol{\phi}(z')$.

When considering the nonlinear function class as an approximator of the value function, we need to develop a new concept analogous to the dimension $d$ that characterizes the intrinsic complexity of the function class $\mathcal{F}$. We do so by making use of the maximal information gain, $\Gamma_{\mathbf{K}}(T, \lambda)$ [Srinivas et al., 2010], where $T$ is the episode number and $H$ is the time horizon. In particular, we define the *effective dimension* of the RKHS $\mathcal{H}$ with respect to the mixture MG as follows:

**Definition 2.** We define the effective dimension $\Gamma_{\mathbf{K}}(T, \lambda)$ as follows:

$$\Gamma_{\mathbf{K}}(T, \lambda) := \sup_{(V_i)_i, (z_i)_i} \frac{1}{2} \log \det\left(\mathbf{I} + \mathbf{K}(\{V_i\}_i, \{z_i\}_i)/\lambda\right),$$

for any $1 \leq i \leq T$, $V_i : \mathcal{S} \to [-H, H]$, $z_i \in \mathcal{Z}$, where $V_i$'s are functions mapping from $\mathcal{S}$ to $[-H, H]$ and $z_i$'s are state-action tuples. Here, $\mathbf{K}(\{V_i\}_i, \{z_i\}_i) \in \mathbb{R}^{T \times T}$ and its $(p, q)$-th entry for any $1 \leq p, q \leq T$ is $[\mathbf{K}(\{V_i\}_i, \{z_i\}_i)]_{(p,q)} = \boldsymbol{k}_{V_p, V_q}(z_p, z_q)$.

By the boundedness of $\phi_V$ as in Section 3.2, it is easy to verify that both the tabular MG and the linear mixture MG enjoy a finite effective dimension. Specifically, for finite RKHS $\mathcal{H}$ with rank $d$, $\Gamma_{\mathbf{K}}(T, \lambda) = O(d \cdot \log T)$ approximately depicts the rank of $\mathcal{H}$. Via a concentration argument, we first present our main lemma for bounding the estimation error when choosing $\beta_t = \beta$ for all $t \geq 1$:

**Lemma 3.** Assuming that for any $h \in [H]$, $\|\boldsymbol{\theta}_h^*\|_{\mathcal{H}} \leq B$. Let $\lambda = 1 + 1/T$ and $\beta$ satisfies $(\beta/H)^2 \geq 2\Gamma_{\mathbf{K}}(T, \lambda) + 2 + 4 \cdot \log(1/\delta) + 2\lambda(B/H)^2$. Then, for any $\delta > 0$, with probability at least $1 - \delta$ the following holds for any $(t, h) \in [T] \times [H]$ and any $(x, a, b) \in \mathcal{S} \times \mathcal{A} \times \mathcal{A}$:

$$\left|\left\langle \boldsymbol{\phi}_{\overline{V}_{h+1}^t}(x, a, b), \overline{\boldsymbol{\theta}}_h^t - \boldsymbol{\theta}_h^* \right\rangle_{\mathcal{H}}\right| \leq \beta \cdot \overline{w}_h^t(x, a, b), \quad \left|\left\langle \boldsymbol{\phi}_{\underline{V}_{h+1}^t}(x, a, b), \underline{\boldsymbol{\theta}}_h^t - \boldsymbol{\theta}_h^* \right\rangle_{\mathcal{H}}\right| \leq \beta \cdot \underline{w}_h^t(x, a, b).$$

We are now ready to present our main theorem.

**Theorem 4** (RKHS function approximation). Under the same conditions as Lemma 3, with probability at least $1 - \delta$, KernelCCE-VTR has the following regret

$$\text{Regret}(T) = O\left(\beta H \sqrt{T \cdot \Gamma_{\mathbf{K}}(T, \lambda)} + 1\right).$$

**Remark 5.** Theorem 4 suggests that by treating the norm $B$ as a constant, KernelCCE-VTR achieves an $\widetilde{O}(\Gamma_{\mathbf{K}}(T, \lambda) H^2 \sqrt{T})$ regret bound. When the RKHS degenerates to the Euclidean space, the regret bound reduces to $\widetilde{O}(dH^2\sqrt{T})$, which matches the $\widetilde{O}(dH^{3/2}\sqrt{T})$ regret for linear mixture MGs presented by Chen et al. [2022] up to a $\sqrt{H}$ factor.

Similar to Xie et al. [2020], by using a standard online-to-batch conversion technique, we can convert the regret bound in Theorem 4 to a PAC bound. For simplicity, let the initial states of each episode be the same, i.e., $x_1^t = x_1$. After $T$ episodes, we select $t_0 \in [T]$ such that

$$t_0 = \underset{t \in [T]}{\text{argmin}} \left\{ \overline{V}_1^t(x_1) - \underline{V}_1^t(x_1) \right\}, \tag{5.1}$$

which yields the following sample complexity guarantee for finding an $\epsilon$-approximate NE policy pair $(\pi^{t_0}, \nu^{t_0})$.

**Corollary 6** (Sample complexity). Under the same condition as Theorem 4, by setting $T = O\left(\beta^2 H^2 \Gamma_{\mathbf{K}}(T, \lambda)/\epsilon^2\right) = \widetilde{O}\left(H^4 \Gamma_{\mathbf{K}}^2(T, \lambda)/\epsilon^2\right)$ and selecting $t_0$ as in (5.1), the policy pair $(\pi^{t_0}, \nu^{t_0})$ is an $\epsilon$-approximate NE.

## 6 Bernstein-type Bonus, Misspecification, and Neural Function Approximation

In this section, we propose several extensions of KernelCCE-VTR. Section 6.1 introduces KernelCCE-VTR with Bernstein-type bonus. Section 6.2 discusses kernel function approximation with misspecification. We also specialize the kernel function approximation with misspecification to the neural function approximation setting, which is deferred to Appendix C.

### 6.1 KernelCCE-VTR with Bernstein-type Bonus

Recall that in KernelCCE-VTR, we need to choose $\beta$ in order to calculate the optimistic and pessimistic state-action value functions $\overline{Q}_h^t(\cdot), \underline{Q}_h^t(\cdot)$ defined in (4.3). The theoretical value of $\beta$ is defined in Lemma 3, which controls the uncertainty of the action-value estimate. Such a choice of $\beta$ is due to a Hoeffding-type concentration used in the proof of Lemma 3. It has been shown in Zhou et al. [2021a] that by using a Bernstein-type bonus and a sharp analysis based on the total variance lemma, one can obtain an improved algorithm with a tighter regret bound. Following this idea, we propose a KernelCCE-VTR+ algorithm, which replaces the Hoeffding-type bonus with a Bernstein-type bonus. To demonstrate the construction of the Bernstein-type bonus, we take the max player for example. In particular, we solve the following weighted kernel ridge regression problem:

$$\overline{\boldsymbol{\theta}}_{h,1}^t = \min_{\boldsymbol{\theta} \in \mathcal{H}} \sum_{\tau=1}^{t-1} \left[ \overline{V}_{h+1}^\tau(x_{h+1}^\tau) - \left\langle \boldsymbol{\phi}_{\overline{V}_{h+1}^\tau}(z_h^\tau), \boldsymbol{\theta} \right\rangle_{\mathcal{H}} \right]^2 / \left( \overline{R}_h^\tau \right)^2 + \lambda_1 \|\boldsymbol{\theta}\|_{\mathcal{H}}^2, \tag{6.1}$$

where the input is the normalized feature mapping $\boldsymbol{\phi}_{\overline{V}_{h+1}^\tau}(z_h^\tau)/\overline{R}_h^\tau$, the output is the normalized value function $\overline{V}_{h+1}^\tau(x_{h+1}^\tau)/\overline{R}_h^\tau$, and the normalization factor $\overline{R}_h^\tau$ is an upper bound on the conditional variance of $\overline{V}_{h+1}^\tau(x_{h+1}^\tau)$. It is straightforward to verify that (6.1) admits a closed-form solution. Given that solution, we can compute the upper confidence bound of the action-value functions $Q_h^{*,\nu}$.

In detail, we define $\overline{\boldsymbol{\Psi}}_{h,1}^t := \left( \boldsymbol{\phi}_{\overline{V}_{h+1}^1}(z_h^1)/\overline{R}_h^1, \ldots, \boldsymbol{\phi}_{\overline{V}_{h+1}^{t-1}}(z_h^{t-1})/\overline{R}_h^{t-1} \right)^\top \in \mathcal{H}^{t-1}$. The gram matrix $\overline{\mathbf{K}}_{h,1}^t$, vector-valued function $\overline{\boldsymbol{k}}_{h,1}^t$ and the confidence region centered at $\overline{\boldsymbol{\theta}}_{h,1}^t$ in the RKHS $\mathcal{H}$ can be calculated the same as in Algorithm 1, except that $\overline{\boldsymbol{\Psi}}_h^t$ is replaced by $\overline{\boldsymbol{\Psi}}_{h,1}^t$. Then the optimistic estimate of the action-value function $Q_h^{*,\nu}$ has the following form:

$$\overline{Q}_h^t(z) = \Pi_{[-H,H]} \left[ r_h(z) + \overline{\boldsymbol{k}}_{h,1}^t(z)^\top (\overline{\mathbf{K}}_{h,1}^t + \lambda \mathbf{I})^{-1} \overline{\boldsymbol{y}}_{h,1}^t + \beta_t \cdot \overline{w}_{h,1}^t(z) \right], \tag{6.2}$$

where $\overline{\boldsymbol{y}}_{h,1}^t := \left[ \overline{V}_{h+1}^1(x_h^1)/\overline{R}_h^1, \ldots \overline{V}_{h+1}^{t-1}(x_h^{t-1})/\overline{R}_h^{t-1} \right]^\top$ and

$$\overline{w}_{h,1}^t(z) = \lambda_1^{-1/2} \cdot \left[ \boldsymbol{k}_{\overline{V}_{h+1}^t, \overline{V}_{h+1}^t}(z,z) - \overline{\boldsymbol{k}}_{h,1}^t(z)^\top \left( \overline{\mathbf{K}}_{h,1}^t + \lambda_1 \cdot \mathbf{I} \right)^{-1} \overline{\boldsymbol{k}}_{h,1}^t(z) \right]^{1/2}.$$

Due to space limit, we defer the conditional variance estimator $\overline{R}_h^t$ to Appendix B. Similarly, we can construct the pessimistic estimate of the action-value function $Q_h^{\pi,*}$ for the min player. We have the following informal result for `KernelCCE-VTR+`. The full algorithm and its formal result can be found in Appendix B.

**Theorem 7** (Informal). Let $d_{\text{eff}} = \Gamma_{\mathbf{K}}(T, \lambda)$, with proper choice of $\overline{R}_h^t, \underline{R}_h^t$ and $\beta_t$, with probability at least $1 - \delta$, `KernelCCE-VTR+` has the following regret

$$\text{Regret}(T) = \widetilde{O}\left( d_{\text{eff}}^2 H^3 + \sqrt{d_{\text{eff}} H^4 + d_{\text{eff}}^2 H^3} \sqrt{T} + \left( d_{\text{eff}}^7 H^7 + d_{\text{eff}}^4 H^9 \right)^{1/4} T^{1/4} \right).$$

**Remark 8.** When $T$ is sufficiently large and $\Gamma_{\mathbf{K}}(T, \lambda)$ is larger than $H$, the regret bound in Theorem 7 is dominated by $\widetilde{O}\left( \Gamma_{\mathbf{K}}(T, \lambda) H^{3/2} \sqrt{T} \right)$, which improves the $\widetilde{O}\left( \Gamma_{\mathbf{K}}(T, \lambda) H^2 \sqrt{T} \right)$ regret derived in Theorem 4 by a factor of $\sqrt{H}$. Compared with the $\widetilde{\Omega}\left( d H^{3/2} \sqrt{T} \right)$ lower bound proposed in Chen et al. [2022], our `KernelCCE-VTR+` algorithm is almost optimal when it reduces to the linear mixture MG.

## 6.2 Kernel Function Approximation with Misspecification

In this subsection, we consider the case where the function class may not be confined to an RKHS, but instead the distance to it can be bounded. This can be formulated as kernel function approximation with misspecification. We assume that there exists a misspecification error between the RKHS $\mathcal{H}$ and the true transition probability $\mathbb{P}_h(s'|z)$.

**Assumption 9.** There exists an $\iota_{\text{mis}} > 0$, an RKHS $\mathcal{H}$ with feature mapping $\phi : \mathcal{Z} \mapsto \mathcal{S} \times \mathcal{H}$, and an unknown true parameter $\boldsymbol{\theta}_h^* \in \mathcal{H}$ satisfying $\|\boldsymbol{\theta}_h^*\|_{\mathcal{H}} \leq B$ such that for any $h \in [H]$, the distance of the transition probability $\mathbb{P}_h$ to $\mathcal{H}$ can be bounded by $\iota_{\text{mis}}$, which is $\left\| \mathbb{P}_h(\cdot|z) - \langle \phi(\cdot|z), \boldsymbol{\theta}_h^* \rangle_{\mathcal{H}} \right\|_{\text{TV}} \leq \iota_{\text{mis}}$.

In order to deal with model misspecification, the key idea is to enlarge $\beta_t$ in the definition of the optimistic action-value function in (4.3). More specifically, we will add an extra $\mathcal{O}(H \iota_{\text{mis}} \sqrt{t})$ term brought by misspecification error to $\beta$ specified in Lemma 3. We can show that `KernelCCE-VTR` with such enlarged $\beta$ will have a sublinear regret in the presence of misspecification.

**Theorem 10** (RKHS function approximation with misspecification). Assuming that for any $h \in [H]$, $\|\boldsymbol{\theta}_h^*\|_{\mathcal{H}} \leq B$. Set $\lambda = 1 + 1/T$ in the `KernelCCE-VTR` Algorithm. For any $\delta > 0$ and any $\beta_t$ satisfying $(\beta_t/H)^2 \geq 2\Gamma_{\mathbf{K}}(T, \lambda) + 3 + 6 \cdot \log(1/\delta) + 3\lambda (B/H)^2 + 3\iota_{\text{mis}}^2 t$, there exists a global constant $c > 0$ such that with probability at least $1 - \delta$, we have

$$\text{Regret}(T) \leq c \left( \beta_T H \sqrt{T \cdot \Gamma_{\mathbf{K}}(T, \lambda)} + 1 + H^2 T \iota_{\text{mis}} \right).$$

In words, Theorem 10 suggests that in the misspecified case, `KernelCCE-VTR` can achieve the same regret as that in the well-specified case up to an $O\left( \sqrt{\Gamma_{\mathbf{K}}(T, \lambda)} H^2 T \iota_{\text{mis}} \right)$ error. Such a linear dependence on $\iota_{\text{mis}}$ matches the result of single agent RL for the finite dimensional case [Jin et al., 2020, Zanette et al., 2020].

## 7 Conclusions

In this work, we studied learning for two-player mixture MGs using a kernel function approximation. We introduced a new formulation of kernel mixture MGs and proposed an algorithm `KernelCCE-VTR` that exploits the kernel function of the MG. We show that our `KernelCCE-VTR` is able to achieve a sublinear $\widetilde{O}(d_{\mathbf{K}} H^2 \sqrt{T})$ regret. We further improve our algorithm with a *Bernstein-type bonus* and *weighted kernel ridge regression*, which enjoys a better $\widetilde{O}(d_{\mathbf{K}} H^{3/2} \sqrt{T})$ regret and nearly matches

the regret lower bound in Chen et al. [2022] when reducing to linear mixture MGs. Finally, we extend our analysis of the basic RKHS setting to a more general nonlinear function approximation setting with misspecification errors and demonstrate that neural networks can be treated as a special instance of this misspecification framework. We believe our framework and analysis greatly broadens the expressiveness of the function classes used for MGs. We leave the study of learning in general-sum MGs by kernel function approximation as future work.

## Acknowledgments and Disclosure of Funding

We thank the anonymous reviewers and area chair for their helpful comments. DZ and QG are supported in part by the National Science Foundation CAREER Award 1906169 and the Sloan Research Fellowship. MIJ is supported in part by the Mathematical Data Science program of the Office of Naval Research under grant number N00014-18-1-2764 and by the Vannevar Bush Faculty Fellowship program under grant number N00014-21-1-2941 and NSF grant IIS-1901252. The views and conclusions contained in this paper are those of the authors and should not be interpreted as representing any funding agencies.

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
