## Appendix

The appendix is organized as follows. In Appendix A we introduce basic properties of RKHS. In Appendix B we discuss the implementation details of `KernelCCE-VTR+`. In Appendix C we introduce more details for applying our algorithm to the neural function approximation setting. In Appendix D we prove results for `KernelCCE-VTR`. In Appendix E we prove results for `KernelCCE-VTR+`. In Appendix F we prove results for `KernelCCE-VTR` with misspecification. In Appendix G we prove results for `KernelCCE-VTR` with neural function approximation. In Appendix H we prove the remaining auxiliary lemmas. Finally, in Appendix I we discuss the implementation details of `FIND_CCE` as an instance of linear programming.

## A  Properties of the Reproducing Kernel Hilbert Spaces

Recall that in Section 4, we define the update rule of $\overline{Q}_h^t, \underline{Q}_h^t$ in Eq. (4.3), where each term is defined in the sense of computational accessibility. For convenience of theoretical analysis, in this section we provide the equivalent forms of the $Q$-update on the RKHS. We have the following simple facts:

**Lemma 11.** Define covariance matrices $\overline{\mathbf{\Lambda}}_h^t, \underline{\mathbf{\Lambda}}_h^t : \mathcal{H} \mapsto \mathcal{H}$ as

$$\overline{\mathbf{\Lambda}}_h^t := \lambda \mathbf{I}_{\mathcal{H}} + \left(\overline{\mathbf{\Psi}}_h^t\right)^\top \left(\overline{\mathbf{\Psi}}_h^t\right), \quad \underline{\mathbf{\Lambda}}_h^t := \lambda \mathbf{I}_{\mathcal{H}} + \left(\underline{\mathbf{\Psi}}_h^t\right)^\top \left(\underline{\mathbf{\Psi}}_h^t\right), \tag{A.1}$$

where $\mathbf{I}_{\mathcal{H}}$ is the identity mapping on $\mathcal{H}$. Then the following holds:

(a) $\overline{\boldsymbol{\theta}}_h^t := \left(\overline{\mathbf{\Psi}}_h^t\right)^\top \left[\overline{\mathbf{K}}_h^t + \lambda \mathbf{I}\right]^{-1} \overline{\mathbf{y}}_h^t = \left(\overline{\mathbf{\Lambda}}_h^t\right)^{-1} \left(\overline{\mathbf{\Psi}}_h^t\right)^\top \overline{\mathbf{y}}_h^t \in \mathcal{H}$ and the same holds for $\underline{\boldsymbol{\theta}}_h^t$;

(b) $\overline{w}_h^t = \left[\phi_{\overline{V}_{h+1}^t}(z)^\top \overline{\mathbf{\Lambda}}_h^t \phi_{\overline{V}_{h+1}^t}(z)\right]^{1/2}$ and the same holds for $\underline{w}_h^t$;

(c) $\phi_{\overline{V}_{h+1}^t}(z) = \left(\overline{\mathbf{\Psi}}_h^t\right)^\top (\overline{\mathbf{K}}_h^t + \lambda \mathbf{I})^{-1} \overline{\mathbf{k}}_h^t(z) + \lambda \cdot (\overline{\mathbf{\Lambda}}_h^t)^{-1} \phi_{\overline{V}_{h+1}^t}(z)$.

*Proof.* We prove the statements as follows.

(a) By definition of $\overline{\mathbf{K}}_h^t$ in Section 4, we note that

$$\left(\overline{\mathbf{\Psi}}_h^t\right)^\top \left[\overline{\mathbf{K}}_h^t + \lambda \mathbf{I}\right] = \left(\overline{\mathbf{\Psi}}_h^t\right)^\top \left[\left(\overline{\mathbf{\Psi}}_h^t\right) \left(\overline{\mathbf{\Psi}}_h^t\right)^\top + \lambda \mathbf{I}\right] = \left[\left(\overline{\mathbf{\Psi}}_h^t\right)^\top \left(\overline{\mathbf{\Psi}}_h^t\right) + \lambda \mathbf{I}_{\mathcal{H}}\right] \left(\overline{\mathbf{\Psi}}_h^t\right)^\top.$$

Taking the inverse operation on both sides of the second equality, we conclude

$$\left[\left(\overline{\mathbf{\Psi}}_h^t\right) \left(\overline{\mathbf{\Psi}}_h^t\right)^\top + \lambda \mathbf{I}\right]^{-1} \left(\overline{\mathbf{\Psi}}_h^t\right)^{-\top} = \left(\overline{\mathbf{\Psi}}_h^t\right)^{-\top} \left[\left(\overline{\mathbf{\Psi}}_h^t\right)^\top \left(\overline{\mathbf{\Psi}}_h^t\right) + \lambda \mathbf{I}_{\mathcal{H}}\right]^{-1},$$

and hence we arrive at an equality on space $\mathcal{H} \times \mathbb{R}^t$ that:

$$\left(\overline{\mathbf{\Psi}}_h^t\right)^\top \left[\overline{\mathbf{K}}_h^t + \lambda \mathbf{I}\right]^{-1} = \left(\overline{\mathbf{\Psi}}_h^t\right)^\top \left[\left(\overline{\mathbf{\Psi}}_h^t\right) \left(\overline{\mathbf{\Psi}}_h^t\right)^\top + \lambda \mathbf{I}\right]^{-1}$$

$$= \left[\left(\overline{\mathbf{\Psi}}_h^t\right)^\top \left(\overline{\mathbf{\Psi}}_h^t\right) + \lambda \mathbf{I}_{\mathcal{H}}\right]^{-1} \left(\overline{\mathbf{\Psi}}_h^t\right)^\top = \left(\overline{\mathbf{\Lambda}}_h^t\right)^{-1} \left(\overline{\mathbf{\Psi}}_h^t\right)^\top.$$

Multiplying both sides by $\overline{\mathbf{y}}_h^t$ we have that the close form solution of Eq. (4.1) satisfies

$$\overline{\boldsymbol{\theta}}_h^t := \left(\overline{\mathbf{\Psi}}_h^t\right)^\top \left[\overline{\mathbf{K}}_h^t + \lambda \mathbf{I}\right]^{-1} \overline{\mathbf{y}}_h^t = \left(\overline{\mathbf{\Lambda}}_h^t\right)^{-1} \left(\overline{\mathbf{\Psi}}_h^t\right)^\top \overline{\mathbf{y}}_h^t \in \mathcal{H},$$

which proves item $(a)$ of our results. The same argument holds for $\underline{\boldsymbol{\theta}}_h^t$.

**Algorithm 2** `KernelCCE-VTR+`

---

1: **Input:** bonus parameter $\lambda_1, \lambda_2 > 0$.
2: **for** episode $t = 1, 2, \ldots, T$ **do**
3:    Receive initial state $x_1^t$
4:    **for** step $h = H, H-1, \ldots, 1$ **do**
5:       Estimate $\overline{R}_h^t, \underline{R}_h^t$ as in Eq. (B.2)
6:       Calculate $\overline{Q}_h^t(\cdot), \underline{Q}_h^t(\cdot)$ as in Eq. (6.2)
7:       For each $x$, let $\sigma_h^t(x) = \texttt{FIND\_CCE}(\overline{Q}_h^t, \underline{Q}_h^t, x)$
8:       Let $\overline{V}_h^t(x) = \mathbb{E}_{(a,b) \sim \sigma_h^t(x)} \overline{Q}_h^t(x, a, b)$ and $\underline{V}_h^t(x) = \mathbb{E}_{(a,b) \sim \sigma_h^t(x)} \underline{Q}_h^t(x, a, b)$
9:    **end for**
10:   **for** step $h = 1, 2, \ldots, T$ **do**
11:      Sample $(a_h^t, b_h^t) \sim \sigma_h^t(x_h^t)$
12:      $P_1$ takes action $a_h^t$, $P_2$ takes action $b_h^t$
13:      Observe next state $x_{h+1}^t$
14:    **end for**
15: **end for**

---

(b) By definition of $\overline{w}_h^t$, $\overline{k}_h^t$ and $\overline{\mathbf{K}}_h^t$, we have

$$
\begin{aligned}
\overline{w}_h^t(z) &= \lambda^{-1/2} \cdot \left[ \boldsymbol{k}_{\overline{V}_{h+1}^t, \overline{V}_{h+1}^t}(z,z) - \overline{\boldsymbol{k}}_h^t(z)^\top \left( \overline{\mathbf{K}}_h^t + \lambda \mathbf{I} \right)^{-1} \overline{\boldsymbol{k}}_h^t(z) \right]^{1/2} \\
&= \lambda^{-1/2} \cdot \left[ \boldsymbol{k}_{\overline{V}_{h+1}^t, \overline{V}_{h+1}^t}(z,z) - \boldsymbol{\phi}_{\overline{V}_{h+1}^t}^\top(z) \left( \overline{\boldsymbol{\Psi}}_h^t \right)^\top \left( \overline{\mathbf{K}}_h^t + \lambda \mathbf{I} \right)^{-1} \left( \overline{\boldsymbol{\Psi}}_h^t \right) \boldsymbol{\phi}_{\overline{V}_{h+1}^t}(z) \right]^{1/2} \\
&= \lambda^{-1/2} \cdot \left[ \boldsymbol{k}_{\overline{V}_{h+1}^t, \overline{V}_{h+1}^t}(z,z) - \boldsymbol{\phi}_{\overline{V}_{h+1}^t}^\top(z) \left( \overline{\boldsymbol{\Lambda}}_h^t \right)^{-1} \left( \overline{\boldsymbol{\Psi}}_h^t \right)^\top \left( \overline{\boldsymbol{\Psi}}_h^t \right) \boldsymbol{\phi}_{\overline{V}_{h+1}^t}(z) \right]^{1/2} \\
&= \lambda^{-1/2} \cdot \left[ \boldsymbol{\phi}_{\overline{V}_{h+1}^t}^\top \left( \overline{\boldsymbol{\Lambda}}_h^t \right)^{-1} \left( \overline{\boldsymbol{\Lambda}}_h^t \right) \boldsymbol{\phi}_{\overline{V}_{h+1}^t}(z) - \boldsymbol{\phi}_{\overline{V}_{h+1}^t}^\top \left( \overline{\boldsymbol{\Lambda}}_h^t \right)^{-1} \left( \overline{\boldsymbol{\Psi}}_h^t \right)^\top \left( \overline{\boldsymbol{\Psi}}_h^t \right) \boldsymbol{\phi}_{\overline{V}_{h+1}^t}(z) \right]^{1/2} \\
&= \left[ \boldsymbol{\phi}_{\overline{V}_{h+1}^t}(z) (\overline{\boldsymbol{\Lambda}}_h^t)^{-1} \boldsymbol{\phi}_{\overline{V}_{h+1}^t}(z) \right]^{1/2}.
\end{aligned}
$$

This concludes the proof of item $(b)$ of our result. The same argument holds for $\underline{w}_h^t(z)$.

(c) Noting that from the definition of $\overline{\boldsymbol{\Lambda}}_h^t$ in Eq. (A.1),

$$
\begin{aligned}
\boldsymbol{\phi}_{\overline{V}_{h+1}^t}(z) &= \left( \overline{\boldsymbol{\Lambda}}_h^t \right)^{-1} \left( \overline{\boldsymbol{\Lambda}}_h^t \right) \boldsymbol{\phi}_{\overline{V}_{h+1}^t}(z) = \left( \overline{\boldsymbol{\Lambda}}_h^t \right)^{-1} \left( \lambda \mathbf{I}_{\mathcal{H}} + \left( \overline{\boldsymbol{\Psi}}_h^t \right)^\top \left( \overline{\boldsymbol{\Psi}}_h^t \right) \right) \boldsymbol{\phi}_{\overline{V}_{h+1}^t}(z) \\
&= \left( \overline{\boldsymbol{\Lambda}}_h^t \right)^{-1} \left( \overline{\boldsymbol{\Psi}}_h^t \right)^\top \left( \overline{\boldsymbol{\Psi}}_h^t \right) \boldsymbol{\phi}_{\overline{V}_{h+1}^t}(z) + \lambda \cdot \left( \overline{\boldsymbol{\Lambda}}_h^t \right)^{-1} \boldsymbol{\phi}_{\overline{V}_{h+1}^t}(z).
\end{aligned}
$$

Applying the results in the proof of item $(a)$ on $\left( \overline{\boldsymbol{\Lambda}}_h^t \right)^{-1} \left( \overline{\boldsymbol{\Psi}}_h^t \right)^\top$, we have that

$$
\begin{aligned}
\boldsymbol{\phi}_{\overline{V}_{h+1}^t}(z) &= \left( \overline{\boldsymbol{\Psi}}_h^t \right)^\top \left[ \overline{\mathbf{K}}_h^t + \lambda \mathbf{I} \right]^{-1} \left( \overline{\boldsymbol{\Psi}}_h^t \right) \boldsymbol{\phi}_{\overline{V}_{h+1}^t}(z) + \lambda \cdot \left( \overline{\boldsymbol{\Lambda}}_h^t \right)^{-1} \boldsymbol{\phi}_{\overline{V}_{h+1}^t}(z) \qquad \text{(A.2)} \\
&= \left( \overline{\boldsymbol{\Psi}}_h^t \right)^\top \left[ \overline{\mathbf{K}}_h^t + \lambda \mathbf{I} \right]^{-1} \overline{\boldsymbol{k}}_h^t(z) + \lambda \cdot \left( \overline{\boldsymbol{\Lambda}}_h^t \right)^{-1} \boldsymbol{\phi}_{\overline{V}_{h+1}^t}(z),
\end{aligned}
$$

which concludes the proof of item $(c)$.

$\square$

# B   Details of `KernelCCE-VTR+`

In this section, we propose more details for the algorithm `KernelCCE-VTR+`. We consider the following ridge regression problem with each term weighed by its estimated variance:

$$\overline{\boldsymbol{\theta}}_{h,1}^t = \min_{\boldsymbol{\theta}\in\mathcal{H}} \sum_{\tau=1}^{t-1} \left[\overline{V}_{h+1}^\tau(x_{h+1}^\tau) - \left\langle \phi_{\overline{V}_{h+1}^\tau}(z_h^\tau), \boldsymbol{\theta}\right\rangle_\mathcal{H}\right]^2 / \left(\overline{R}_h^\tau\right)^2 + \lambda_1 \|\boldsymbol{\theta}\|_\mathcal{H}^2,$$

$$\underline{\boldsymbol{\theta}}_{h,1}^t = \min_{\boldsymbol{\theta}\in\mathcal{H}} \sum_{\tau=1}^{t-1} \left[\underline{V}_{h+1}^\tau(x_{h+1}^\tau) - \left\langle \phi_{\underline{V}_{h+1}^\tau}(z_h^\tau), \boldsymbol{\theta}\right\rangle_\mathcal{H}\right]^2 / (\underline{R}_h^\tau)^2 + \lambda_1 \|\boldsymbol{\theta}\|_\mathcal{H}^2.$$

Here we use $\overline{R}_h^\tau, \underline{R}_h^\tau$ to denote upper bounds on the conditional variance of $\overline{V}_{h+1}^\tau(x_{h+1}^\tau)$ and $\underline{V}_{h+1}^\tau(x_{h+1}^\tau)$ respectively, which we will specify in later subsections. Next we define the necessary quantities in estimating the regret bound. Similarily as in previous sections, we define

$$\overline{\boldsymbol{\Psi}}_{h,1}^t := \left(\phi_{\overline{V}_{h+1}^1}(z_h^1)/\overline{R}_h^1, \ldots \phi_{\overline{V}_{h+1}^{t-1}}(z_h^{t-1})/\overline{R}_h^{t-1}\right)^\top \in \mathcal{H}^{t-1}, \quad \text{and}$$

$$\underline{\boldsymbol{\Psi}}_{h,1}^t := \left(\phi_{\underline{V}_{h+1}^1}(z_h^1)/\underline{R}_h^1, \ldots \phi_{\underline{V}_{h+1}^{t-1}}(z_h^{t-1})/\underline{R}_h^{t-1}\right)^\top \in \mathcal{H}^{t-1}.$$

The gram matrix $\overline{\mathbf{K}}_{h,1}^t$, vector-valued function $\overline{\boldsymbol{k}}_{h,1}^t$ and the confidence region centered at $\overline{\boldsymbol{\theta}}_{h,1}^t$ in the RKHS $\mathcal{H}$ are defined accordingly by replacing $\overline{\boldsymbol{\Psi}}_h^t, \underline{\boldsymbol{\Psi}}_h^t$ by $\overline{\boldsymbol{\Psi}}_{h,1}^t, \underline{\boldsymbol{\Psi}}_{h,1}^t$ respectively. The optimistic (pessimistic version can be defined accordingly) estimates of the action value function have the following closed form solution:

$$\overline{Q}_h^t(z) = \Pi_{[-H,H]}[r_h(z) + \overline{\boldsymbol{k}}_{h,1}^t(z)^\top (\overline{\mathbf{K}}_{h,1}^t + \lambda\mathbf{I})^{-1}\overline{\boldsymbol{y}}_{h,1}^t + \beta_t \cdot \overline{w}_{h,1}^t(z)], \quad\quad (\text{B.1})$$

where

$$\overline{\boldsymbol{y}}_{h,1}^t := \left[\overline{V}_{h+1}^1(x_h^1)/\overline{R}_h^1, \ldots \overline{V}_{h+1}^{t-1}(x_h^{t-1})/\overline{R}_h^{t-1}\right]^\top,$$

and

$$\overline{w}_{h,1}^t(z) = \lambda_1^{-1/2} \cdot \left[\boldsymbol{k}_{\overline{V}_{h+1}^t, \overline{V}_{h+1}^t}(z,z) - \overline{\boldsymbol{k}}_{h,1}^t(z)^\top \left(\overline{\mathbf{K}}_{h,1}^t + \lambda_1 \cdot \mathbf{I}\right)^{-1} \overline{\boldsymbol{k}}_{h,1}^t(z)\right]^{1/2}.$$

The full version of the algorithm is presented formally in Algorithm 2.

## B.1   Variance Estimator

In order to determine the values of $\overline{R}_h^\tau, \underline{R}_h^\tau$, we note that we can solve a ridge regression problem for estimating the expected square of the value function:

$$\overline{\boldsymbol{\theta}}_{h,2}^t = \min_{\boldsymbol{\theta}\in\mathcal{H}} \sum_{\tau=1}^{t-1} \left[\left(\overline{V}_{h+1}^\tau(x_{h+1}^\tau)\right)^2 - \left\langle \phi_{(\overline{V}_{h+1}^\tau)^2}(z_h^\tau), \boldsymbol{\theta}\right\rangle_\mathcal{H}\right]^2 + \lambda_2 \|\boldsymbol{\theta}\|_\mathcal{H}^2,$$

$$\underline{\boldsymbol{\theta}}_{h,2}^t = \min_{\boldsymbol{\theta}\in\mathcal{H}} \sum_{\tau=1}^{t-1} \left[(\underline{V}_{h+1}^\tau(x_{h+1}^\tau))^2 - \left\langle \phi_{(\underline{V}_{h+1}^\tau)^2}(z_h^\tau), \boldsymbol{\theta}\right\rangle_\mathcal{H}\right]^2 + \lambda_2 \|\boldsymbol{\theta}\|_\mathcal{H}^2.$$

By defining

$$\overline{\boldsymbol{\Psi}}_{h,2}^t := \left(\phi_{(\overline{V}_{h+1}^1)^2}(z_h^1), \ldots \phi_{(\overline{V}_{h+1}^{t-1})^2}(z_h^{t-1})\right)^\top \in \mathcal{H}^{t-1},$$

$$\underline{\boldsymbol{\Psi}}_{h,2}^t := \left(\phi_{(\underline{V}_{h+1}^1)^2}(z_h^1), \ldots \phi_{(\underline{V}_{h+1}^{t-1})^2}(z_h^{t-1})\right)^\top \in \mathcal{H}^{t-1},$$

we can define the gram matrix $\overline{\mathbf{K}}_{h,2}^t$, vector-valued function $\overline{\boldsymbol{k}}_{h,2}^t$, and

$$\overline{w}_{h,2}^t(z) = \lambda_2^{-1/2} \cdot \left[ \boldsymbol{k}_{(\overline{V}_{h+1}^t)^2,(\overline{V}_{h+1}^t)^2}(z,z) - \overline{\boldsymbol{k}}_{h,2}^t(z)^\top \left( \overline{\mathbf{K}}_{h,2}^t + \lambda_2 \cdot \mathbf{I} \right)^{-1} \overline{\boldsymbol{k}}_{h,2}^t(z) \right]^{1/2}.$$

The variance estimator is thus defined as:

$$\mathbb{V}^{\mathrm{est}} \overline{V}_{h+1}^t(z_h^t) := \left\langle \boldsymbol{\phi}_{(\overline{V}_{h+1}^t)^2}(z_h^t), \overline{\boldsymbol{\theta}}_{h,2}^t \right\rangle_{\mathcal{H}} - \left( \left\langle \boldsymbol{\phi}_{(\overline{V}_{h+1}^t)^2}(z_h^t), \overline{\boldsymbol{\theta}}_{h,1}^t \right\rangle_{\mathcal{H}} \right)^2$$

$$\approx \mathbb{P}_h \left( \underline{V}_{h+1}^t(x_{h+1}^t) \right)^2 - \left( \mathbb{P}_h \underline{V}_{h+1}^t(x_{h+1}^t) \right)^2,$$

and

$$\left( \overline{R}_h^t \right)^2 := \max\{ \mathbb{V}^{\mathrm{est}} \overline{V}_{h+1}^t(z_h^t) + \overline{E}_h^t, (\alpha_t)^2 \},$$

$$\overline{E}_h^t := \min \left\{ H^2, \beta_t^{(2)} \overline{w}_{h,2}^t \right\} + \min \left\{ H^2, 2H\beta_t^{(1)} \overline{w}_{h,1}^t \right\}. \tag{B.2}$$

Up till now, we have finished the definition of the variance estimator for the upper-value estimator. The lower-value estimator can be defined in a similar fashion, and we omit the details.

## B.2   Main Results

In this section, we provide theoretical results for the regret bound under the weighted setting described above. First we propose a key lemma which suggests that our constructed $\overline{\boldsymbol{\theta}}_{h,1}^t$ and $\overline{\boldsymbol{\theta}}_{h,2}^t$ are good estimates to $\boldsymbol{\theta}_h^*$ with high probability.

**Lemma 12.** Assuming that for any $h \in [H]$, $\|\boldsymbol{\theta}_h^*\|_{\mathcal{H}} \leq B$. Let $\alpha_t, \beta_t^{(1)}, \beta_t^{(2)}$ satisfy $\alpha_t = \alpha$,

$$\beta_t^{(1)} = (16H/\alpha)\sqrt{\Gamma_{\mathbf{K}}(T,\lambda_1\alpha^2)}\sqrt{\log(4t^2H/\delta)} + (8H/\alpha)\log(4t^2H/\delta) + \sqrt{\lambda_1} \cdot B, \tag{B.3}$$

$$\beta_t^{(2)} = 16H^2\sqrt{\Gamma_{\mathbf{K}}(T,\lambda_2/H^2)}\sqrt{\log(4t^2H/\delta)} + 8H^2\log(4t^2H/\delta) + \sqrt{\lambda_2} \cdot B, \tag{B.4}$$

then for any $\delta > 0$, there exists an event $\mathcal{E}$ satisfying $\mathbb{P}(\mathcal{E}) \geq 1 - 2\delta$ such that on $\mathcal{E}$ the following holds for any $(t,h) \in [T] \times [H]$ and any $(x,a,b) \in \mathcal{S} \times \mathcal{A} \times \mathcal{A}$:

$$\left| \left\langle \boldsymbol{\phi}_{\overline{V}_{h+1}^t}(z_h^t), \boldsymbol{\theta}_h^* - \overline{\boldsymbol{\theta}}_{h,1}^t \right\rangle_{\mathcal{H}} \right| \leq \beta_t^{(1)} \cdot \overline{w}_{h,1}^k(z_h^k),$$

and

$$\left| \left\langle \boldsymbol{\phi}_{(\overline{V}_{h+1}^t)^2}(z_h^t), \boldsymbol{\theta}_h^* - \overline{\boldsymbol{\theta}}_{h,2}^t \right\rangle_{\mathcal{H}} \right| \leq \beta_t^{(2)} \cdot \overline{w}_{h,2}^k(z_h^k).$$

We now propose our main theorem, which is the formal version of Theorem 7 and suggests that the regret bound of Algorithm 2 is upper bounded by $\widetilde{O}\left( \Gamma_{\mathbf{K}}(T,\lambda)H^2\sqrt{T} \right)$.

**Theorem 13.** Assuming that for any $h \in [H]$, $\|\boldsymbol{\theta}_h^*\|_{\mathcal{H}} \leq B$. Let $\lambda = 1/B^2$, $d_{\mathrm{eff}} = \Gamma_{\mathbf{K}}(T,\lambda)$, $\lambda_1 = d_{\mathrm{eff}}/(B^2H^2)$, $\lambda_2 = H^2/B^2$, and taking $\beta_t, \beta_t^{(1)}, \beta_t^{(2)}$ as in Eq. (E.5), (B.3) and (B.4), then with probability at least $1 - \delta$, the following holds that:

$$\mathrm{Regret}(T) := \sum_{t=1}^T V_1^{*,\nu^t}(x_h^t) - V_1^{\pi^t,*}(x_h^t)$$

$$\leq \widetilde{O}\left( d_{\mathrm{eff}}^2 H^3 + \sqrt{d_{\mathrm{eff}}H^4 + d_{\mathrm{eff}}^2 H^3}\sqrt{T} + \left( d_{\mathrm{eff}}^7 H^8 + d_{\mathrm{eff}}^4 H^9 \right)^{1/4} T^{1/4} \right).$$

Proofs of Lemma 12 and Theorem 13 are deferred to Section E.

**Algorithm 3** NeuralCCE-VTR

1: **Input:** bonus parameter $\beta_t > 0$.
2: **for** episode $t = 1, 2, \ldots, T$ **do**
3:     Receive initial state $x_1^t$
4:     **for** step $h = H, H-1, \ldots, 1$ **do**
5:         Solve the optimization problem (C.1)
6:         Calculate $\overline{Q}_h^t(\cdot), \underline{Q}_h^t(\cdot)$ as in Eq. (C.4)
7:         For each $x$, let $\sigma_h^t(x) = \texttt{FIND\_CCE}(\overline{Q}_h^t, \underline{Q}_h^t, x)$
8:         Let $\overline{V}_h^t(x_h^t) = \mathbb{E}_{(a,b) \sim \sigma_h^t(x_h^t)} \overline{Q}_h^t(x_h^t, a, b)$ and $\underline{V}_h^t(x_h^t) = \mathbb{E}_{(a,b) \sim \sigma_h^t(x_h^t)} \underline{Q}_h^t(x_h^t, a, b)$
9:     **end for**
10:    **for** step $h = 1, 2, \ldots, T$ **do**
11:       Sample $(a_h^t, b_h^t) \sim \sigma_h^t(x_h^t)$
12:       $P_1$ takes action $a_h^t$, $P_2$ takes action $b_h^t$
13:       Observe next state $x_{h+1}^t$
14:    **end for**
15: **end for**

## C  Neural Network (NN) Function Approximation

### C.1  Neural Function Approximation

In this subsection, we show that neural network function approximation can be treated as a special case of kernel function approximation with misspecification. We denote $z := (x, a, b)$ as a vector in $\mathbb{R}^d$ that satisfies $\|z\| = 1$ and represent the parameters of a $L$-Layer fully connected neural network $f$ by $\boldsymbol{\theta} := \left[\text{vec}(\mathbf{W}_1)^\top, \text{vec}(\mathbf{W}_2)^\top, \ldots, \text{vec}(\mathbf{W}_L)^\top\right]^\top$, where $\mathbf{W}_1 \in \mathbb{R}^{m \times d}$, $\mathbf{W}_l \in \mathbb{R}^{m \times m}$ for $2 \leq l \leq L-1$ and $\mathbf{W}_L \in \mathbb{R}^{1 \times m}$. The neural network $f(z; \boldsymbol{\theta})$ with parameter set $\boldsymbol{\theta}$ can be defined as:

$$f(z; \boldsymbol{\theta}) = \sqrt{m} \mathbf{W}_L G\left(\cdots G\left(\mathbf{W}_2 G\left(\mathbf{W}_1 z\right)\right)\right),$$

where $G(\cdot) : \mathbb{R} \mapsto \mathbb{R}$ is an activation function. For $1 \leq l \leq L-1$, $\mathbf{W}_l$ is initialized as $\mathbf{W}_l = (\mathbf{W}, \mathbf{0}; \mathbf{0}, \mathbf{W})$, where each entry of $\mathbf{W}$ is generated independently from normal distribution $N(0, 4/m)$; $\mathbf{W}_L$ is initialized as $\mathbf{W}_L = (\boldsymbol{w}^\top, -\boldsymbol{w}^\top)$, where each entry of $\boldsymbol{w}$ is generated independently from $N(0, 2/m)$. Given the initialized parameter $\boldsymbol{\theta}^{(0)}$, we choose the feature map as the gradient of $f$ at $\boldsymbol{\theta}^{(0)}$:

$$\boldsymbol{\phi}(z) = \nabla_{\boldsymbol{\theta}} f(z; \boldsymbol{\theta}^{(0)})/\sqrt{m}.$$

Then we define the weighted kernel function $\boldsymbol{k}_{V_1, V_2}(\cdot, \cdot)$ in Definition 1 with $\boldsymbol{\phi}(z)$. Similarly, we define the effective dimension $\Gamma_{\mathbf{K}}(T, \lambda)$ with respect to the kernel function $\boldsymbol{k}_{V_1, V_2}(\cdot, \cdot)$, in the same fashion of Definition 2. Our assumption is that for $\forall h \in [H]$ our transition probability $\mathbb{P}_h$ can be modeled by the neural network with parameter $\boldsymbol{\theta}_h^*$ satisfying $\left\|\boldsymbol{\theta}_h^* - \boldsymbol{\theta}^{(0)}\right\|_2 \leq B$:

$$\mathbb{P}_h(x'|z) = f(x', z; \boldsymbol{\theta}_h^*).$$

Now we show the details of our Algorithm 3. Similarly as in Eq. (4.1), we solve penalized ridge regression problem for the min-player and the max-player respectively

$$\overline{\boldsymbol{\theta}}_h^t = \min_{\boldsymbol{\theta} \in \mathbb{R}^P} \sum_{\tau=1}^{t-1} \left[\overline{V}_{h+1}^\tau(x_{h+1}^\tau) - f_{\overline{V}_{h+1}^\tau}(z_h^\tau; \boldsymbol{\theta})\right]^2 + \lambda \cdot \left\|\boldsymbol{\theta} - \boldsymbol{\theta}^{(0)}\right\|^2,$$

$$\underline{\boldsymbol{\theta}}_h^t = \min_{\boldsymbol{\theta} \in \mathbb{R}^P} \sum_{\tau=1}^{t-1} \left[\underline{V}_{h+1}^\tau(x_{h+1}^\tau) - f_{\underline{V}_{h+1}^\tau}(z_h^\tau; \boldsymbol{\theta})\right]^2 + \lambda \cdot \left\|\boldsymbol{\theta} - \boldsymbol{\theta}^{(0)}\right\|^2, \tag{C.1}$$

where $p = md + m^2(L-2) + m$ is the dimension of the parameter space, and $f_{\overline{V}_{h+1}^\tau}, f_{\underline{V}_{h+1}^\tau}$ are defined similarly as $\boldsymbol{\phi}_{\overline{V}_{h+1}^\tau}$ as follows:

$$f_{\overline{V}_{h+1}^\tau}(z; \boldsymbol{\theta}) = \sum_{s' \in \mathcal{S}} \overline{V}_{h+1}^\tau(s') f(s', z; \boldsymbol{\theta}), \qquad f_{\underline{V}_{h+1}^\tau}(z; \boldsymbol{\theta}) = \sum_{s' \in \mathcal{S}} \underline{V}_{h+1}^\tau(s') f(s', z; \boldsymbol{\theta}).$$

For given $\overline{\boldsymbol{\theta}}_h^t, \underline{\boldsymbol{\theta}}_h^t$, we define

$$\overline{\boldsymbol{\Psi}}_h^t := \left( \boldsymbol{\phi}_{\overline{V}_{h+1}^1}(z_h^1; \overline{\boldsymbol{\theta}}_h^2), \dots \boldsymbol{\phi}_{\overline{V}_{h+1}^{t-1}}(z_h^{t-1}; \overline{\boldsymbol{\theta}}_h^t) \right)^\top,$$

$$\underline{\boldsymbol{\Psi}}_h^t := \left( \boldsymbol{\phi}_{\underline{V}_{h+1}^1}(z_h^1; \underline{\boldsymbol{\theta}}_h^2), \dots \boldsymbol{\phi}_{\underline{V}_{h+1}^{t-1}}(z_h^{t-1}; \underline{\boldsymbol{\theta}}_h^t) \right)^\top. \tag{C.2}$$

Furthermore,

$$\overline{\boldsymbol{\Lambda}}_h^t := \lambda \mathbf{I} + (\overline{\boldsymbol{\Psi}}_h^t)^\top \overline{\boldsymbol{\Psi}}_h^t, \qquad \underline{\boldsymbol{\Lambda}}_h^t := \lambda \mathbf{I} + (\underline{\boldsymbol{\Psi}}_h^t)^\top \underline{\boldsymbol{\Psi}}_h^t,$$

and

$$\overline{w}_h^t(z) := \left[ \boldsymbol{\phi}_{\overline{V}_{h+1}^t}(z; \overline{\boldsymbol{\theta}}_h^t)^\top (\overline{\boldsymbol{\Lambda}}_h^t)^{-1} \boldsymbol{\phi}_{\overline{V}_{h+1}^t}(z; \overline{\boldsymbol{\theta}}_h^t) \right]^{1/2},$$

$$\underline{w}_h^t(z) := \left[ \boldsymbol{\phi}_{\underline{V}_{h+1}^t}(z; \underline{\boldsymbol{\theta}}_h^t)^\top (\underline{\boldsymbol{\Lambda}}_h^t)^{-1} \boldsymbol{\phi}_{\underline{V}_{h+1}^t}(z; \underline{\boldsymbol{\theta}}_h^t) \right]^{1/2}. \tag{C.3}$$

Using the $\overline{\boldsymbol{\Lambda}}_h^t, \underline{\boldsymbol{\Lambda}}_h^t, \overline{w}_h^t, \underline{w}_h^t$, we estimate the optimal value functions as

$$\overline{Q}_h^t(z) = \Pi_{[-H,H]}\{r_h(z) + f_{\overline{V}_{h+1}^t}(z; \overline{\boldsymbol{\theta}}_h^t) + \beta \cdot \overline{w}_h^t(z)\},$$

$$\underline{Q}_h^t(z) = \Pi_{[-H,H]}\{r_h(z) + f_{\underline{V}_{h+1}^t}(z; \underline{\boldsymbol{\theta}}_h^t) - \beta \cdot \underline{w}_h^t(z)\}. \tag{C.4}$$

Combining with the procedures of finding a CCE, we present the full version of our algorithm as in Algorithm 3.

We have the following result on the neural network at initialization.

**Lemma 14.** There exist constants $C_i > 0$ such that for any $\delta \in (0,1)$, if $B$ satisfies that

$$B \geq C_1 m^{-1} L^{-3/2} \max\{\log^{-3/2} m, \log^{3/2}(|\mathcal{Z}|HL^2/\delta)\},$$

$$B \leq C_2 L^{-6}(\log m)^{-3/2},$$

then with probability at least $1 - \delta$, we have for all $z \in \mathcal{Z}$, $h \in [H]$ and $V_h : \mathcal{S} \to [-1,1]$,

$$|\mathbb{P}_h V_h(z) - \langle \boldsymbol{\phi}_{V_h}(z), \boldsymbol{\theta}_h^* - \boldsymbol{\theta}^{(0)} \rangle| \leq C_3 |\mathcal{S}| B^{4/3} m^{-1/6} L^3 \sqrt{\log m},$$

and

$$\|\boldsymbol{\phi}_{V_h}(z)\|_2 \leq C := C_4 |\mathcal{S}| \sqrt{L}.$$

Lemma 14, whose proof is deferred to Appendix H, suggests that in the NN approximation setting, Assumption 9 for the misspecified kernel approximation setting is satisfied with $\iota_{\mathrm{mis}} = C_3 |\mathcal{S}| B^{4/3} m^{-1/6} L^3 \sqrt{\log m}$ and with probability at least $1 - \delta$. The misspecified error is sufficiently small when $m$ is large. We note that the definition of $\phi(z)$ in the NN setting does not match the boundedness assumption in Section 3.2. We balance the scale of $\phi(z)$ by the constant $C$ in Lemma 14 which goes into the choice of $\lambda = C^2(1 + 1/T)$. With these at hand, we are ready to present our main result for NN approximation:

**Theorem 15** (NN approximation). Let $C$ be the constant in Lemma 14. Assuming that for any $h \in [H]$, $\|\boldsymbol{\theta}_h^* - \boldsymbol{\theta}^{(0)}\|_2 \leq B$. Set $\lambda = C^2(1 + 1/T)$ in the `KernelCCE-VTR` Algorithm. For any $\delta > 0$ and any $\beta_t$ satisfying

$$\left( \frac{\beta_t}{H} \right)^2 \geq 2\Gamma_{\mathbf{K}}(T, \lambda) + 3 + 6 \cdot \log\left( \frac{1}{\delta} \right) + 3\lambda \left( \frac{B}{H} \right)^2 + 3 \cdot C^2 \cdot B^{8/3} \cdot m^{-1/12} \cdot t \cdot \log m,$$

there exists a global constant $c > 0$ such that with probability at least $1 - 2\delta$, we have

$$\mathrm{Regret}(T) \leq c \left( \beta_T H \sqrt{T \cdot \Gamma_{\mathbf{K}}(T, \lambda)} + 1 + B^{4/3} H^2 T m^{-1/6} \sqrt{\log m} \right).$$

Theorem 15 suggests that when we use an overparameterized deep neural network ($m \gg 1$) to approximate the transition dynamic, `KernelCCE-VTR` achieves an $\widetilde{O}(\Gamma_{\mathbf{K}}(T, \lambda)H^2\sqrt{T})$ regret, which is of the same order as that in Theorem 4. We defer the proof of Theorem 15 to Appendix G.

# D Proof of Results for `KernelCCE-VTR`

In this subsection, we provide the proof of our main Theorem 4 on RKHS.

## D.1 Proof of Theorem 4

We recall that the duality gap is defined as $\sum_{t=1}^{T} V_1^{*,\nu^t}(x_1^t) - V_1^{\pi^t,*}(x_1^t)$. As can be seen in our algorithm, we maintain an optimistic estimate of $V_h^{*,\upsilon^t}(\cdot)$ as $\overline{V}_{h+1}^t(\cdot)$ and a pessimistic estimate of $V_h^{\pi^t,*}(\cdot)$ as $\underline{V}_{h+1}^t(\cdot)$. Hence the term $\overline{V}_{h+1}^t(x_h^t) - \underline{V}_{h+1}^t(x_h^t)$ is approximately the upper bound of the duality gap. We write the decomposition formally as below:

$$V_h^{*,\nu^t}(x_h^t) - V_h^{\pi^t,*}(x_h^t) = \underbrace{\overline{V}_h^t(x_h^t) - \underline{V}_h^t(x_h^t)}_{\text{I}} - \underbrace{\left(V_h^{\pi^t,*}(x_h^t) - \underline{V}_h^t(x_h^t)\right)}_{\text{II}} - \underbrace{\left(\overline{V}_h^t(x_h^t) - V_h^{*,\nu^t}(x_h^t)\right)}_{\text{III}}.$$

$$\text{(D.1)}$$

We use $\overline{\delta}_h^t$ to denote an important quantity $\left\langle \phi_{\overline{V}_{h+1}^t}(z_h^t), \boldsymbol{\theta}_h^* - \overline{\boldsymbol{\theta}}_h^t \right\rangle_{\mathcal{H}}$ (and $\left\langle \phi_{\underline{V}_{h+1}^t}(z_h^t), \boldsymbol{\theta}_h^* - \underline{\boldsymbol{\theta}}_h^t \right\rangle_{\mathcal{H}}$) in estimating the duality gap. In the rest of the proof we aim to show that all of the above three terms can be bounded by a quantity related to $\overline{\delta}_h^t$ ($\underline{\delta}_h^t$) and a stochastic random variable that forms a martingale difference sequence when considering for all $h \in [H], t \in [T]$.

For bounding term I, we first define two sequences of zero mean variables:

$$\gamma_h^t := \overline{Q}_h^t(x_h^t, a_h^t, b_h^t) - \underline{Q}_h^t(x_h^t, a_h^t, b_h^t) - \mathbb{E}_{(a,b)}\left[\overline{Q}_h^t(x, a, b) - \underline{Q}_h^t(x, a, b)\right],$$

$$\xi_h^t := \left(\mathbb{P}_h(\overline{V}_{h+1}^t - \underline{V}_{h+1}^t)\right)(x_h^t, a_h^t, b_h^t) - \left(\overline{V}_{h+1}^t(x_{h+1}^t) - \underline{V}_{h+1}^t(x_{h+1}^t)\right),$$

$$\text{(D.2)}$$

where $\gamma_h^t$ depicts the stochastic error with respect to the policy and $\xi_h^t$ depicts the stochastic error with respect to the transition. We refer the readers to the proof of Lemma 16 for detailed explanations on these two error term. Given the above definition, we have the following Lemma 16.

**Lemma 16.** Under the settings of Lemma 3, we have the following recursive bound for $\forall h \in [H]$:

$$\overline{V}_h^t(x_h^t) - \underline{V}_h^t(x_h^t)$$
$$\leq \overline{V}_{h+1}^t(x_{h+1}^t) - \underline{V}_{h+1}^t(x_{h+1}^t) + 2\beta_t \min\{1, \overline{w}_h^t(x_h^t)\} + 2\beta_t \min\{1, \underline{w}_h^t(x_h^t)\} + \xi_h^t + \gamma_h^t.$$

*Proof of Lemma 16.* By the update rule of Algorithm 3, we have the following relation:

$$\overline{V}_h^t(x_h^t) - \underline{V}_h^t(x_h^t) = \mathbb{E}_{(a,b)\sim\sigma_h^t(x_h^t)}\left[\overline{Q}_h^t(x_h^t, a, b) - \underline{Q}_h^t(x_h^t, a, b)\right]. \qquad \text{(D.3)}$$

We note that the RHS of Eq. (D.3) is an expectation over the CCE distribution $\sigma_h^t(x_h^t)$, which can be decomposed into one sample from the distribution plus a noise term as follows:

$$\overline{V}_h^t(x_h^t) - \underline{V}_h^t(x_h^t) = \overline{Q}_h^t(x_h^t, a_h^t, b_h^t) - \underline{Q}_h^t(x_h^t, a_h^t, b_h^t) + \gamma_h^t, \qquad \text{(D.4)}$$

where

$$\gamma_h^t := \overline{Q}_h^t(x_h^t, a_h^t, b_h^t) - \underline{Q}_h^t(x_h^t, a_h^t, b_h^t) - \mathbb{E}_{(a,b)}\left[\overline{Q}_h^t(x, a, b) - \underline{Q}_h^t(x, a, b)\right].$$

Furthermore, for bounding the difference between the upper confidence $Q$ estimation and the lower confidence $Q$ estimation, we have

$$\overline{Q}_h^t(z_h^t) - \underline{Q}_h^t(z_h^t)$$
$$\leq \left\langle \phi_{\overline{V}_{h+1}^t}(z_h^t), \overline{\boldsymbol{\theta}}_h^t \right\rangle_{\mathcal{H}} - \left\langle \phi_{\underline{V}_{h+1}^t}(z_h^t), \underline{\boldsymbol{\theta}}_h^t \right\rangle_{\mathcal{H}} + \beta_t \overline{w}_h^t(z_h^t) + \beta_t \underline{w}_h^t(z_h^t)$$
$$= \left\langle \phi_{\overline{V}_{h+1}^t}(z_h^t), \overline{\boldsymbol{\theta}}_h^t \right\rangle_{\mathcal{H}} - \left\langle \phi_{\overline{V}_{h+1}^t}(z_h^t), \boldsymbol{\theta}_h^* \right\rangle_{\mathcal{H}} + \left\langle \phi_{\overline{V}_{h+1}^t}(z_h^t), \boldsymbol{\theta}_h^* \right\rangle_{\mathcal{H}} - \left\langle \phi_{\underline{V}_{h+1}^t}(z_h^t), \boldsymbol{\theta}_h^* \right\rangle_{\mathcal{H}}$$
$$+ \left\langle \phi_{\underline{V}_{h+1}^t}(z_h^t), \boldsymbol{\theta}_h^* \right\rangle_{\mathcal{H}} - \left\langle \phi_{\underline{V}_{h+1}^t}(z_h^t), \underline{\boldsymbol{\theta}}_h^t \right\rangle_{\mathcal{H}} + \beta_t \overline{w}_h^t(z_h^t) + \beta_t \underline{w}_h^t(z_h^t)$$

$$= \left\langle \phi_{\overline{V}^t_{h+1}}(z^t_h), \overline{\boldsymbol{\theta}}^t_h - \boldsymbol{\theta}^*_h \right\rangle_{\mathcal{H}} + \left( \mathbb{P}_h(\overline{V}^t_{h+1} - \underline{V}^t_{h+1}) \right)(z^t_h) + \left\langle \phi_{\underline{V}^t_{h+1}}(z^t_h), \boldsymbol{\theta}^*_h - \underline{\boldsymbol{\theta}}^t_h \right\rangle_{\mathcal{H}}$$
$$+ \beta_t \overline{w}^t_h(z^t_h) + \beta_t \underline{w}^t_h(z^t_h).$$

By utilizing Lemma 3, we further arrive at:

$$\overline{Q}^t_h(z^t_h) - \underline{Q}^t_h(z^t_h) \leq \left( \mathbb{P}_h(\overline{V}^t_{h+1} - \underline{V}^t_{h+1}) \right)(z^t_h) + 2\beta_t \overline{w}^t_h(z^t_h) + 2\beta_t \underline{w}^t_h(z^t_h),$$

where again by extracting the sequence

$$\xi^t_h := \left( \mathbb{P}_h(\overline{V}^t_{h+1} - \underline{V}^t_{h+1}) \right)(x^t_h, a^t_h, b^t_h) - \left( \overline{V}^t_{h+1}(x^t_{h+1}) - \underline{V}^t_{h+1}(x^t_{h+1}) \right),$$

we have

$$\overline{Q}^t_h(z^t_h) - \underline{Q}^t_h(z^t_h) \leq \overline{V}^t_{h+1}(x^t_{h+1}) - \underline{V}^t_{h+1}(x^t_{h+1}) + 2\beta \overline{w}^t_h(z^t_h) + 2\beta \underline{w}^t_h(z^t_h) + \xi^t_h. \tag{D.5}$$

Combining Eq. (D.4) and (D.5) concludes the following recursive bound:

$$\overline{V}^t_h(x^t_h) - \underline{V}^t_h(x^t_h) \leq \overline{V}^t_{h+1}(x^t_{h+1}) - \underline{V}^t_{h+1}(x^t_{h+1}) + 2\beta_t \overline{w}^t_h(z^t_h) + 2\beta_t \underline{w}^t_h(z^t_h) + \xi^t_h + \gamma^t_h$$

Moreover, due to the fact that $\overline{V}^t_h(x^t_h) - \underline{V}^t_h(x^t_h) \leq 2H$, we rewrite the above inequality into:

$$\overline{V}^t_h(x^t_h) - \underline{V}^t_h(x^t_h)$$
$$\leq \min \left\{ 2H, \overline{V}^t_{h+1}(x^t_{h+1}) - \underline{V}^t_{h+1}(x^t_{h+1}) + 2\beta_t \overline{w}^t_h(z^t_h) + 2\beta_t \underline{w}^t_h(z^t_h) + \xi^t_h + \gamma^t_h \right\}$$
$$\leq \min \left\{ 2H, 2\beta_t \overline{w}^t_h(z^t_h) + 2\beta_t \underline{w}^t_h(z^t_h) \right\} + \overline{V}^t_{h+1}(x^t_{h+1}) - \underline{V}^t_{h+1}(x^t_{h+1}) + \xi^t_h + \gamma^t_h$$
$$\leq 2\beta_t \min\{1, \overline{w}^t_h(z^t_h)\} + 2\beta_t \min\{1, \underline{w}^t_h(z^t_h)\} + \overline{V}^t_{h+1}(x^t_{h+1}) - \underline{V}^t_{h+1}(x^t_{h+1}) + \xi^t_h + \gamma^t_h,$$

where the last inequality is due to the choice of $\beta$ satisfying $\beta/H \geq 1$. This completes the proof of Lemma 16. $\qquad \square$

For bounding II and III, we use induction to prove that III $\geq 0$ for every $h$, that is,

$$\overline{V}^t_h(x^t_h) - V^{*,\nu^t}_h(x^t_h) \geq 0. \tag{D.6}$$

Then the same statement will also hold for II due to the symmetric property. The statement holds for $h = H+1$, where III $= 0$ (since $\overline{V}^t_{H+1} = V^{*,\nu^t}_{H+1} = 0$ by definition). Suppose the statement holds for $h+1$. Let $(a, b) \in \mathcal{A}_1 \times \mathcal{A}_2$ and $z := (x^t_h, a, b)$. If $\overline{Q}^t_h(z) \geq H$, then by definition, III $\geq 0$. Suppose $\overline{Q}^t_h(z) < H$, then by definition of $\overline{Q}^t_h(z)$, we have

$$\overline{Q}^t_h(z) - Q^{*,\nu^t}_h(z) = \left\langle \phi_{\overline{V}^t_{h+1}}(z), \overline{\boldsymbol{\theta}}^t_h - \boldsymbol{\theta}^*_h \right\rangle_{\mathcal{H}} + \left( \mathbb{P}_h(\overline{V}^t_{h+1} - V^{*,\nu^t}_{h+1}) \right)(z) + \beta_t \overline{w}^t_h(z)$$
$$\geq -\beta_t \overline{w}^t_h(z) + \beta_t \overline{w}^t_h(z) = 0, \tag{D.7}$$

where the first inequality holds due to the statement holds for $h+1$, which leads to $\overline{V}^t_{h+1} - V^{*,\nu^t}_{h+1} \geq 0$, and Lemma 3 that gives a bound for $\left\langle \phi_{\overline{V}^t_{h+1}}(z), \overline{\boldsymbol{\theta}}^t_h - \boldsymbol{\theta}^*_h \right\rangle_{\mathcal{H}}$. Next, we have

$$\overline{V}^t_h(x^t_h) - V^{*,\nu^t}_h(x^t_h) = \mathbb{E}_{(a,b)\sim\sigma^t_h(x^t_h)} \overline{Q}^t_h(x^t_h, a, b) - \mathbb{E}_{a\sim\mathrm{br}(\nu^t_h), b\sim\nu^t_h} Q^{*,\nu^t}_h(x^t_h, a, b)$$
$$\geq \mathbb{E}_{a\sim\mathrm{br}(\nu^t_h), b\sim\nu^t_h} \overline{Q}^t_h(x^t_h, a, b) - \mathbb{E}_{a\sim\mathrm{br}(\nu^t_h), b\sim\nu^t_h} Q^{*,\nu^t}_h(x^t_h, a, b)$$
$$= \mathbb{E}_{a\sim\mathrm{br}(\nu^t_h), b\sim\nu^t_h} \left[ \overline{Q}^t_h(x^t_h, a, b) - Q^{*,\nu^t}_h(x^t_h, a, b) \right] \geq 0,$$

where $\nu^t_h := \mathcal{P}_2\sigma^t_h$ and $\pi^t_h := \mathcal{P}_1\sigma^t_h$ is the projection of $\sigma^t_h$ on the first and second coordinate respectively and br is the best response policy of a given distribution. The last inequality holds due to (D.7). Therefore, the statement holds for $h$, which suggests that the induction holds.

Combining with Eq. (D.1), we arrive at a bound in terms of $\overline{w}^t_h(z^t_h), \underline{w}^t_h(z^t_h)$ and the martingale difference sequences:

$$V^{*,\nu^t}_1(x^t_h) - V^{\pi^t,*}_1(x^t_h) \leq \sum_{h=1}^{H} \left( 2\beta_t \min\{1, \overline{w}^t_h(z^t_h)\} + 2\beta_t \min\{1, \underline{w}^t_h(z^t_h)\} + \xi^t_h + \gamma^t_h \right), \tag{D.8}$$

where $\nu^t$ is the policy that operates according to $\nu_h^t$ at time $h$ and $\pi^t$ is the sequence of $\pi_h^t$ accordingly. The rest of the proof follows by bounding $\sum_{t=1}^T \sum_{h=1}^H \min\{1, \overline{w}_h^t(z_h^t)\}, \sum_{t=1}^T \sum_{h=1}^H \min\{1, \underline{w}_h^t(z_h^t)\}$ and the martingale difference sequences.

The bound of $\sum_{t=1}^T \sum_{h=1}^H \min\{1, \overline{w}_h^t(z_h^t)\}$ and $\sum_{t=1}^T \sum_{h=1}^H \min\{1, \underline{w}_h^t(z_h^t)\}$ comes directly from the following lemma 17 which can be simply derived from Lemma 11 in Abbasi-Yadkori et al. [2011] and is an analogue of Lemma E.3 of Yang et al. [2020]:

**Lemma 17** (Lemma E.3 of Yang et al. [2020]). For any sequence $\{\mathbf{x}_t\}_{t\geq 1}$ taking values on the RKHS $\mathcal{H}$ satisfying $\forall t, \|\mathbf{x}_t\|_{\mathcal{H}} \leq L$. Let $I_{\mathcal{H}}$ be the identity operator on $\mathcal{H}$ and $\mathbf{\Lambda}_0 := \lambda I_{\mathcal{H}}$ the multiplication operator by $\lambda$. Furthermore, if we let $\mathbf{\Lambda}_t := \mathbf{\Lambda}_0 + \sum_{i=1}^t \mathbf{x}_i \mathbf{x}_i^\top$ be a positive definite operator from $\mathcal{H}$ to $\mathcal{H}$ and $\mathbf{K}_t \in \mathbb{R}^{t \times t}$ the gram matrix of $\mathcal{H}$ obtained from $\{\mathbf{x}_t\}_{t\geq 1}$. Then the following holds for $\forall t > 0$:

$$\sum_{i=1}^t \min\{1, \mathbf{x}_i^\top \mathbf{\Lambda}_{t-1}^{-1} \mathbf{x}_i\} \leq 2\log\det(\mathbf{I} + \mathbf{K}_t/\lambda).$$

We recall that by Lemma 11, $\overline{w}_h^t = \left[ \phi_{\overline{V}_{h+1}^t}(z)^\top (\overline{\mathbf{\Lambda}}_h^t)^{-1} \phi_{\overline{V}_{h+1}^t}(z) \right]^{1/2}$, and the same holds for $\underline{w}_h^t$. Let $\mathbf{x}_t = \phi_{\overline{V}_{h+1}^t}(z_h^t)$ and $\mathbf{\Lambda}_0 = \lambda \mathbf{I}$ in Lemma 17 and by applying the Cauchy-Schwarz inequality, we have that

$$\sum_{h=1}^H \sum_{t=1}^T \beta_t \min\{1, \overline{w}_h^t(z_h^t)\}, \quad \sum_{h=1}^H \sum_{t=1}^T \beta_t \min\{1, \underline{w}_h^t(z_h^t)\} \leq 2\beta H \cdot \sqrt{T} \sqrt{\Gamma_{\mathbf{K}}(T, \lambda)}, \qquad \text{(D.9)}$$

where $\beta_t = \beta$ takes values as in Lemma 3 for each $t$ and $\Gamma_{\mathbf{K}}(T, \lambda)$ is defined as the supremum over all $V$'s and $z$'s as in Definition 2. For the martingale difference sequence $\xi_h^t + \gamma_h^t$, as $|\xi_h^t + \gamma_h^t| \leq 4H$, we bound it by Azuma-Hoeffding which gives us with probability at least $1 - \delta$:

$$\sum_{t=1}^T \sum_{h=1}^H \xi_h^t + \gamma_h^t \leq \mathcal{O}\left( H\sqrt{TH} \cdot \log(1/\delta) \right).$$

Combining the above inequality with the bound in (D.9) and (D.8) concludes our proof of Theorem 4.

### D.2 Proof of Corollary 6

Due to the selection of $t_0$, we have

$$V_1^{*,\nu^{t_0}}(x_1) - V_1^{\pi^{t_0},*}(x_1) \leq \overline{V}_1^{t_0}(x_1) - \underline{V}_1^{t_0}(x_1) \leq \frac{1}{T} \sum_{t=1}^T \overline{V}_1^t(x_1) - \underline{V}_1^t(x_1) \leq \sqrt{\frac{\beta^2 H^2 \Gamma_{\mathbf{K}}(T, \lambda)}{T}},$$
$$\text{(D.10)}$$

where the first inequality holds due to (D.6) and its counterpart for $\underline{V}_1^t$, the second one holds due to the selection of $t_0$. From (D.10) we can see that by selecting $T$ as what our statement suggests, the $\epsilon$-approximate NE can be guaranteed.

## E Proof of Results for `KernelCCE-VTR+`

In this section we give the proof of results in Appendix B. One of the key results of this paper is the following Bernstein self-normalized concentration inequality:

**Theorem 18** (Bernstein inequality for vector-valued martingales). Let $\{\mathcal{G}_t\}_{t=1}^\infty$ be a filtration, $\{\mathbf{x}_t, \eta_{t+1}\}_{t\geq 1}$ be a stochastic process so that $\mathbf{x}_t \in \mathbb{R}^d$ is $\mathcal{G}_t$-measurable and $\eta_{t+1} \in \mathbb{R}$ is $\mathcal{G}_{t+1}$-measurable. Fix $R, L, \sigma, \lambda > 0, \boldsymbol{\mu}^* \in \mathbb{R}^d$. For $t \geq 1$ we observe $\langle \boldsymbol{\mu}^*, \mathbf{x}_t \rangle + \eta_{t+1}$ and suppose that $\eta_{t+1}, \mathbf{x}_t$ also satisfy

$$|\eta_{t+1}| \leq R, \quad \mathbb{E}[\eta_{t+1}|\mathcal{G}_t] = 0, \quad \mathbb{E}[\eta_{t+1}^2|\mathcal{G}_t] \leq \sigma^2, \quad \|\mathbf{x}_t\|_2 \leq L.$$

Then, for any $\delta \in (0,1)$, with probability at least $1 - \delta$ we have

$$\left\| \sum_{i=1}^{t} \mathbf{x}_i \eta_{i+1} \right\|_{\mathbf{Z}_t^{-1}} \leq \beta_t, \quad \forall t > 0, \tag{E.1}$$

where for each $t \geq 1$, $\mathbf{Z}_t = \lambda \mathbf{I} + \sum_{i=1}^{t} \mathbf{x}_i \mathbf{x}_i^\top$, and

$$\beta_t = 8\sigma \sqrt{\log \det(\mathbf{I} + \mathbf{K}_t/\lambda) \log(4t^2/\delta)} + 4R \log(4t^2/\delta), \qquad [\mathbf{K}_t]_{i,j} = \langle \mathbf{x}_i, \mathbf{x}_j \rangle_{\mathcal{H}}.$$

*Proof.* The proof can be derived by following the proof of Theorem 2 in Zhou et al. [2021a]. We only need to replace Lemma 12 in Zhou et al. [2021a] with Lemma 17, then the remaining of the proof goes through the same as Zhou et al. [2021a]. □

We first give the proof of Lemma 12.

### E.1 Proof of Lemma 12

*Proof.* We only provide the proof of the max-player, and results of the min-player can be derived similarily. We recall that we define $\overline{\mathbf{y}}_{h,1}^t$ to be the vector of regression targets

$$\left( \overline{V}_{h+1}^1(x_{h+1}^1)/\overline{R}_h^1, \ldots, \overline{V}_{h+1}^{t-1}(x_{h+1}^{t-1})/\overline{R}_h^{t-1} \right)^\top \in \mathbb{R}^{t-1}.$$

Furthermore by Lemma 11 in Section A, we know that $\overline{\boldsymbol{\theta}}_{h,1}^t = \left( \overline{\boldsymbol{\Psi}}_{h,1}^t \right)^\top \left[ \overline{\mathbf{K}}_{h,1}^t + \lambda_1 \cdot \mathbf{I} \right]^{-1} \overline{\mathbf{y}}_{h,1}^t$ and $\boldsymbol{\phi}_{\overline{V}_{h+1}^t}(z) = \left( \overline{\boldsymbol{\Psi}}_{h,1}^t \right)^\top (\overline{\mathbf{K}}_{h,1}^t + \lambda_1 \cdot \mathbf{I})^{-1} \overline{\mathbf{k}}_{h,1}^t(z) + \lambda_1 \cdot (\overline{\boldsymbol{\Lambda}}_{h,1}^t)^{-1} \boldsymbol{\phi}_{\overline{V}_{h+1}^t}(z)$, which enable us to bound the difference $\left\langle \boldsymbol{\phi}_{\overline{V}_{h+1}^t}(z), \overline{\boldsymbol{\theta}}_{h,1}^t - \boldsymbol{\theta}_h^* \right\rangle_{\mathcal{H}}$ as follows:

$$\left\langle \boldsymbol{\phi}_{\overline{V}_{h+1}^t}(z), \overline{\boldsymbol{\theta}}_{h,1}^t - \boldsymbol{\theta}_h^* \right\rangle_{\mathcal{H}} = \boldsymbol{\phi}_{\overline{V}_{h+1}^t}(z)^\top \left( \overline{\boldsymbol{\Psi}}_{h,1}^t \right)^\top \left[ \overline{\mathbf{K}}_{h,1}^t + \lambda_1 \cdot \mathbf{I} \right]^{-1} \overline{\mathbf{y}}_{h,1}^t$$

$$- \left( \boldsymbol{\theta}_h^* \right)^\top \left[ \left( \overline{\boldsymbol{\Psi}}_{h,1}^t \right)^\top \left[ \overline{\mathbf{K}}_{h,1}^t + \lambda_1 \cdot \mathbf{I} \right]^{-1} \overline{\mathbf{k}}_{h,1}^t(z) + \lambda_1 \cdot (\overline{\boldsymbol{\Lambda}}_{h,1}^t)^{-1} \boldsymbol{\phi}_{\overline{V}_{h+1}^t}(z) \right]$$

$$= \underbrace{(\overline{\mathbf{k}}_{h,1}^t)^\top \left[ \overline{\mathbf{K}}_{h,1}^t + \lambda_1 \cdot \mathbf{I} \right]^{-1} \left[ \overline{\mathbf{y}}_{h,1}^t - \overline{\boldsymbol{\Psi}}_{h,1}^t \boldsymbol{\theta}_h^* \right]}_{\mathrm{I}_1} - \underbrace{\lambda_1 \cdot \boldsymbol{\phi}_{\overline{V}_{h+1}^t}(z)^\top (\overline{\boldsymbol{\Lambda}}_{h,1}^t)^{-1} \boldsymbol{\theta}_h^*}_{\mathrm{I}_2}.$$

For bounding $\mathrm{I}_2$, we apply the Cauchy-Schwarz inequality and have

$$\lambda_1 \cdot \boldsymbol{\phi}_{\overline{V}_{h+1}^t}(z)^\top (\overline{\boldsymbol{\Lambda}}_h^t)^{-1} \boldsymbol{\theta}_h^* \leq \left\| \lambda_1 \cdot \boldsymbol{\phi}_{\overline{V}_{h+1}^t}(z)^\top (\overline{\boldsymbol{\Lambda}}_{h,1}^t)^{-1} \right\|_{\mathcal{H}} \cdot \| \boldsymbol{\theta}_h^* \|_{\mathcal{H}}$$

$$\overset{(a)}{\leq} B \cdot \sqrt{\lambda_1 \boldsymbol{\phi}_{\overline{V}_{h+1}^t}(z)^\top (\overline{\boldsymbol{\Lambda}}_{h,1}^t)^{-1} \lambda_1 (\overline{\boldsymbol{\Lambda}}_{h,1}^t)^{-1} \boldsymbol{\phi}_{\overline{V}_{h+1}^t}(z)} \overset{(b)}{\leq} \sqrt{\lambda_1} B \cdot \overline{w}_{h,1}^t(z),$$

where $(a)$ is due to the assumption that $\| \boldsymbol{\theta}_h^* \|_{\mathcal{H}} \leq B$, and $(b)$ is by the definition of $\overline{w}_h^t$ and the fact that $\left( \overline{\boldsymbol{\Lambda}}_{h,1}^t \right)^{-1}$ is a self-adjoint mapping on the RKHS $\mathcal{H}$ satisfying $\left\| \left( \overline{\boldsymbol{\Lambda}}_{h,1}^t \right)^{-1} \right\|_{op} \leq \frac{1}{\lambda_1}$.

For bounding $\mathrm{I}_1$, we observe the following equality:

$$(\overline{\mathbf{k}}_{h,1}^t)^\top \left[ \overline{\mathbf{K}}_{h,1}^t + \lambda_1 \cdot \mathbf{I} \right]^{-1} \left[ \overline{\mathbf{y}}_{h,1}^t - \overline{\boldsymbol{\Psi}}_{h,1}^t \boldsymbol{\theta}_h^* \right]$$

$$= \boldsymbol{\phi}_{\overline{V}_{h+1}^t}(z)^\top (\overline{\boldsymbol{\Lambda}}_{h,1}^t)^{-1} (\overline{\boldsymbol{\Psi}}_{h,1}^t)^\top \left[ \overline{\mathbf{y}}_{h,1}^t - \boldsymbol{\Psi}_{h,1}^t \boldsymbol{\theta}_h^* \right]$$

$$= \boldsymbol{\phi}_{\overline{V}_{h+1}^t}(z)^\top (\overline{\boldsymbol{\Lambda}}_{h,1}^t)^{-1} \sum_{\tau=1}^{t-1} \boldsymbol{\phi}_{\overline{V}_{h+1}^\tau}(z_h^\tau) \left[ \overline{V}_{h+1}^\tau(x_{h+1}^\tau) - (\mathbb{P}_h \overline{V}_{h+1}^\tau)(z_h^\tau) \right] / \left( \overline{R}_h^\tau \right)^2.$$

Again by applying the Cauchy-Schwarz inequality, we bound the RHS of the above equality as

$$|\mathbf{I}_1| \leq \left\| \phi_{\overline{V}^t_{h+1}}(z) \right\|_{(\overline{\boldsymbol{\Lambda}}^t_{h,1})^{-1}} \cdot \left\| \sum_{\tau=1}^{t-1} \phi_{\overline{V}^\tau_{h+1}}(z^\tau_h) \left[ \overline{V}^\tau_{h+1}(x^\tau_{h+1}) - (\mathbb{P}_h \overline{V}^\tau_{h+1})(z^\tau_h) \right] / \left( \overline{R}^\tau_h \right)^2 \right\|_{(\overline{\boldsymbol{\Lambda}}^t_{h,1})^{-1}}.$$

We note that for the given $h$ considered in Lemma 12. If we define $\{\mathcal{F}_t\}_{t\geq 0}$ as the $\sigma$-algebra generated by all data before iteration $t-1$ along with all data before time $h$ at iteration $t$, then

$$\eta_{\tau+1} := \left( \overline{V}^\tau_{h+1}(x^\tau_{h+1}) - (\mathbb{P}_h \overline{V}^\tau_{h+1})(z^\tau_h) \right) / \overline{R}^\tau_h \in \mathcal{F}_{\tau+1} \tag{E.2}$$

is a mean zero random variable with respect to filtration $\mathcal{F}_\tau$. By the choice of $\overline{R}_h$ such that $\overline{R}_h \geq \alpha_t$, we can bound the absolute value of $\eta_{\tau+1}$ by $\left| \left( \overline{V}^\tau_{h+1}(x^\tau_{h+1}) - (\mathbb{P}_h \overline{V}^\tau_{h+1})(z^\tau_h) \right) / \overline{R}^\tau_h \right| \leq 2H/\alpha_\tau$. We take $\eta_{\tau+1}$ as in Eq. (E.2) in Theorem 18 and $\{\mathbf{x}_t\}_{t\geq 1}$ is $\left\{ \phi_{\overline{V}^t_{h+1}}(z^t_h)/\overline{R}^t_h \right\}_{t\geq 1} \in \mathcal{F}_\tau$. Then by directly utilizing Theorem 18, we have that the following inequality holds with probability at least $1 - \delta/H$,

$$\left\| \sum_{\tau=1}^{t-1} \phi_{\overline{V}^\tau_{h+1}}(z^\tau_h) \left[ \overline{V}^t_{h+1}(x^\tau_{h+1}) - (\mathbb{P}_h \overline{V}^t_{h+1})(z^\tau_h) \right] / \overline{R}^\tau_h \right\|^2_{(\overline{\boldsymbol{\Lambda}}^t_{h,1})^{-1}}$$

$$\leq 16H/\alpha \sqrt{\log \det(\mathbf{I} + \mathbf{K}^{(1)}_t/\lambda_1) \log(4t^2 H/\delta)} + 8H/\alpha \log(4t^2 H/\delta)$$

$$\leq 16H/\alpha \sqrt{\log \det(\mathbf{I} + \mathbf{K}_t/(\lambda_1(\alpha_t)^2)) \log(4t^2 H/\delta)} + 8H/\alpha \log(4t^2 H/\delta)$$

$$\leq 16H/\alpha \sqrt{\Gamma_{\mathbf{K}}(T, \lambda_1(\alpha_t)^2)} \sqrt{\log(4t^2 H/\delta)} + 8H/\alpha \log(4t^2 H/\delta),$$

where $\mathbf{K}^{(1)}_t$ is the gram matrix for $\{\mathbf{x}_\tau\}_{\tau\in[t-1]} = \left\{ \phi_{\overline{V}^t_{h+1}}(z^\tau_h)/\overline{R}^\tau_h \right\}_{\tau\in[t-1]}$, $\mathbf{K}_t$ is the gram matrix for $\left\{ \phi_{\overline{V}^t_{h+1}}(z^\tau_h) \right\}_{\tau\in[t-1]}$.

On the other hand, when estimating $\mathbb{P}_h \left( \overline{V}^t_{h+1} \right)^2$, we have the following result regarding $\overline{\boldsymbol{\theta}}^t_{h,2}$ holds with probability at least $1 - \delta/H$:

$$\left| \left\langle \phi_{(\overline{V}^t_{h+1})^2}(z), \overline{\boldsymbol{\theta}}^t_{h,2} - \boldsymbol{\theta}^*_h \right\rangle_{\mathcal{H}} \right| \leq 16H^2 \sqrt{\Gamma_{\mathbf{K}}(T, \lambda_2/H^2)} \sqrt{\log(4t^2 H/\delta)} + 8H^2 \log(4t^2 H/\delta) + \sqrt{\lambda_2} \cdot B.$$

By letting

$$\beta^{(1)}_t = 16H/\alpha \sqrt{\Gamma_{\mathbf{K}}(T, \lambda_1(\alpha_t)^2)} \sqrt{\log(4t^2 H/\delta)} + 8H/\alpha \log(4t^2 H/\delta) + \sqrt{\lambda_1} \cdot B,$$

and

$$\beta^{(2)}_t = 16H^2 \sqrt{\Gamma_{\mathbf{K}}(T, \lambda_2/H^2)} \sqrt{\log(4t^2 H/\delta)} + 8H^2 \log(4t^2 H/\delta) + \sqrt{\lambda_2} \cdot B,$$

we have that from the above theoretical derivation and by taking union bounds over $h \in [H]$,

$$\left| \phi_{\overline{V}^\tau_{h+1}}(z)^\top (\boldsymbol{\theta}^*_h - \overline{\boldsymbol{\theta}}^t_{h,1}) \right| \leq \beta^{(1)}_t \cdot \overline{w}^t_{h,1}(z),$$

and

$$\left| \phi_{(\overline{V}^\tau_{h+1})^2}(z)^\top (\boldsymbol{\theta}^*_h - \overline{\boldsymbol{\theta}}^t_{h,2}) \right| \leq \beta^{(2)}_t \cdot \overline{w}^t_{h,2}(z), \tag{E.3}$$

with probability at least $1 - 2\delta$. This concludes our proof. $\qquad\square$

From Lemma 12, we can prove that $\overline{R}^t_h$ is an upper bound of the actual variance of $\overline{V}^t_{h+1}$.

**Lemma 19.** Following the setting of Lemma 12 and assume that event $\mathcal{E}$ occurs, the following holds for any $(t, h) \in T \times H$:

$$\left| \mathbb{V}^{\text{est}} \overline{V}^t_{h+1}(z^t_h) - \mathbb{V} \overline{V}^t_{h+1}(z^t_h) \right| \leq \min \left\{ H^2, \beta^{(2)}_t \overline{w}^t_{h,2} \right\} + \min \left\{ H^2, 2H\beta^{(1)}_t \overline{w}^t_{h,1} \right\}.$$

*Proof.* By the triangle inequality we have that

$$|\mathbb{V}^{\text{est}}\overline{V}_{h+1}^t(z_h^t) - \mathbb{V}\overline{V}_{h+1}^t(z_h^t)|$$

$$\leq \left|\langle\boldsymbol{\phi}_{(\overline{V}_{h+1}^t)^2}(z_h^t),\boldsymbol{\theta}_h^*\rangle_{\mathcal{H}} - \left[\langle\boldsymbol{\phi}_{(\overline{V}_{h+1}^t)^2}(z_h^t),\overline{\boldsymbol{\theta}}_{h,2}^t\rangle_{\mathcal{H}}\right]_{[0,H^2]}\right|$$

$$+ \left|(\langle\boldsymbol{\phi}_{\overline{V}_{h+1}^t}(z_h^t),\boldsymbol{\theta}_h^*\rangle_{\mathcal{H}})^2 - \left[\langle\boldsymbol{\phi}_{\overline{V}_{h+1}^t}(z_h^t),\overline{\boldsymbol{\theta}}_{h,1}^t\rangle_{\mathcal{H}}\right]_{[-H,H]}^2\right|$$

$$\leq \min\left\{H^2, \left|\langle\boldsymbol{\phi}_{(\overline{V}_{h+1}^t)^2}(z_h^t),\boldsymbol{\theta}_h^* - \overline{\boldsymbol{\theta}}_{h,2}^t\rangle_{\mathcal{H}}\right|\right\} + \min\left\{H^2, 2H\left|\langle\boldsymbol{\phi}_{\overline{V}_{h+1}^t}(z_h^t),\boldsymbol{\theta}_h^* - \overline{\boldsymbol{\theta}}_{h,1}^t\rangle_{\mathcal{H}}\right|\right\}$$

$$\leq \min\left\{H^2, \beta_t^{(2)}\overline{w}_{h,2}^t\right\} + \min\left\{H^2, 2H\beta_t^{(1)}\overline{w}_{h,1}^t\right\},$$

(E.4)

where the last inequality directly comes from Lemma 12. $\square$

**Lemma 20** (Fine-tuned bound). *Assuming that for any $h \in [H]$, $\|\boldsymbol{\theta}_h^*\|_{\mathcal{H}} \leq B$. Let $\beta_t$ satisfy*

$$\beta_t \geq 16\sqrt{\Gamma_{\mathbf{K}}(T, \lambda_1(\alpha_t)^2)}\sqrt{\log(4t^2H/\delta)} + 8H/\alpha\log(4t^2H/\delta) + \sqrt{\lambda_1}\cdot B. \qquad \text{(E.5)}$$

*Then on the event defined in Lemma 12, there exists an event $\mathcal{E}_1$ such that the following holds with probability at least $1 - \delta$ for any $(t,h) \in [T] \times [H]$ and any $(x,a,b) \in \mathcal{S} \times \mathcal{A} \times \mathcal{A}$:*

$$\left|\langle\boldsymbol{\phi}_{\overline{V}_{h_1}^t}(z_h^t),\boldsymbol{\theta}_h^* - \overline{\boldsymbol{\theta}}_{h,1}^t\rangle_{\mathcal{H}}\right| \leq \beta_t \cdot \overline{w}_{h,1}^k(z_h^k).$$

*Proof.* From the definition of $\overline{R}_h$ and $\overline{E}_h^t$, we know that

$$\overline{R}_h^t \geq \mathbb{V}^{\text{est}}\overline{V}_{h+1}^t(z_h^t) + \min\left\{H^2, \beta_t^{(2)}\overline{w}_{h,2}^t\right\} + \min\left\{H^2, 2H\beta_t^{(1)}\overline{w}_{h,1}^t\right\}.$$

Combining with the result in Lemma 19 where we bound the absolute difference between the estimated variance and the true variance in Eq. (E.4), we have that on the event $\mathcal{E}$ defined in Lemma 12:

$$\overline{R}_h^t \geq \mathbb{V}\overline{V}_{h+1}^t(z_h^t).$$

We derive a fine-tuned bound on the variance of $\eta_{t+1}$ defined in Eq. (E.2) that on event $\mathcal{E}$:

$$\mathbb{E}\left[\eta_{t+1}^2 \mid \mathcal{F}_t\right] = \mathbb{V}\overline{V}_{h+1}^t(z_h^t)/\left(\overline{R}_h^t\right)^2 \leq 1.$$

The rest of the proof follows by a direct application of Theorem 18. $\square$

**Lemma 21.** *On the event $\mathcal{E} \cap \mathcal{E}_1$, there exists an event $\mathcal{E}_2$ such that $\mathcal{E}_2$ holds with probability at least $1 - \delta$, we have*

$$\sum_{t=1}^T\sum_{h=1}^H\left(\overline{R}_h^t\right)^2 \leq HT\alpha^2 + 3(H^2T + H^3\log(1/\delta)) + 4H\sum_{t=1}^T\sum_{h=1}^H\mathbb{P}_h[\overline{V}_{h+1}^t - V_{h+1}^{\boldsymbol{\mu}^t}]$$

$$+ 2\beta_T^{(2)}\sqrt{TH}\cdot\sqrt{2H\Gamma_{\mathbf{K}}(T,\lambda_2/H^2)}$$

$$+ 7\beta_T^{(1)}H^2\sqrt{TH}\cdot\sqrt{2H\Gamma_{\mathbf{K}}(T,\lambda_1/\alpha^2)}.$$

*Proof.* First by considering the definition of $\overline{R}_h$, we know that

$$\sum_{t=1}^T\sum_{h=1}^H\left(\overline{R}_h^t\right)^2$$

$$\leq \sum_{t=1}^T\sum_{h=1}^H\left(\alpha_t^2 + \mathbb{V}^{\text{est}}\overline{V}_{h+1}^t(z_h^t) + \overline{E}_h^t\right)$$

$$= \sum_{t=1}^T\sum_{h=1}^H\left(\alpha_t^2 + \mathbb{V}^{\text{est}}\overline{V}_{h+1}^t(z_h^t) + \min\left\{H^2, \beta_t^{(2)}\overline{w}_{h,2}^t\right\} + \min\left\{H^2, 2H\beta_t^{(1)}\overline{w}_{h,1}^t\right\}\right)$$

$$\leq HT\alpha^2 + \sum_{t=1}^{T}\sum_{h=1}^{H}\left(\mathbb{V}\overline{V}_{h+1}^t(z_h^t) + 2\min\left\{H^2, \beta_t^{(2)}\overline{w}_{h,2}^t\right\} + 2\min\left\{H^2, 2H\beta_t^{(1)}\overline{w}_{h,1}^t\right\}\right)$$

$$\leq HT\alpha^2 + \underbrace{\sum_{t=1}^{T}\sum_{h=1}^{H}\left[\mathbb{V}\overline{V}_{h+1}^t(z_h^t) - \mathbb{V}\overline{V}_{h+1}^{\pi_t}(z_h^t)\right]}_{\text{I}} + \underbrace{\sum_{t=1}^{T}\sum_{h=1}^{H}\mathbb{V}\overline{V}_{h+1}^{\pi_t}(z_h^t)}_{\text{II}}$$

$$+ \underbrace{\sum_{t=1}^{T}\sum_{h=1}^{H}2\min\left\{H^2, \beta_t^{(2)}\overline{w}_{h,2}^t\right\} + \sum_{t=1}^{T}\sum_{h=1}^{H}2\min\left\{H^2, 2H\beta_t^{(1)}\overline{w}_{h,1}^t\right\}}_{\text{III}}.$$

The rest of the proof for bounding I, II, III goes the same as in the proof of Lemma A.6 in Chen et al. [2022], except that we replace Lemma B.4 in Chen et al. [2022] with Lemma 17. $\qquad\square$

*Proof of Theorem 13.* The first part of the proof follows almost the same as in the proof of Theorem 4 by replacing $\overline{w}_h^t$ with $\overline{w}_{h,1}^t$ and $\underline{w}_h^t$ with $\underline{w}_{h,1}^t$, except that now we have $\beta\overline{R}_h^t \geq 2H$ so that we have (E.6) instead of (D.8).

$$V_1^{*,\nu^t}(x_1^t) - V_1^{\pi^t,*}(x_1^t)$$
$$\leq \sum_{h=1}^{H}\left(4\beta_t\overline{R}_h^t\min\{1, \overline{w}_{h,1}^t(z_h^t)/\overline{R}_h^t\} + 4\beta_t\underline{R}_h^t\min\{1, \underline{w}_{h,1}^t(z_h^t)/\overline{R}_h^t\} + \xi_h^t + \gamma_h^t\right), \tag{E.6}$$

Similarily, for any $1 \leq h' \leq H$, we have

$$\overline{V}_{h'}^t(x_{h'}^t) - \underline{V}_{h'}^t(x_{h'}^t)$$
$$\leq \sum_{h=1}^{H}\left(2\beta_t\overline{R}_h^t\min\{1, \overline{w}_{h,1}^t(z_h^t)/\overline{R}_h^t\} + 2\beta_t\underline{R}_h^t\min\{1, \underline{w}_{h,1}^t(z_h^t)/\overline{R}_h^t\} + \xi_h^t + \gamma_h^t\right).$$

Applying Azuma-Hoeffding inequality onto (E.7), we have with probability at least $1 - \delta$,

$$\sum_{t=1}^{T}\sum_{h=1}^{H}\mathbb{P}_h[\overline{V}_{h+1}^t - \underline{V}_{h+1}^t](x_h^t, a_h^t, b_h^t)$$
$$\leq \sum_{t=1}^{T}\sum_{h=1}^{H}\left(2\beta_t\overline{R}_h^t\min\{1, \overline{w}_{h,1}^t(z_h^t)/\overline{R}_h^t\} + 2\beta_t\underline{R}_h^t\min\{1, \underline{w}_{h,1}^t(z_h^t)/\overline{R}_h^t\} + \xi_h^t + \gamma_h^t\right) + \sum_{t=1}^{T}\sum_{h=1}^{H}\xi_h^t. \tag{E.7}$$

Then we estimate the two summation terms $\sum_{t=1}^{T}\sum_{h=1}^{H}\overline{R}_h^t\min\{1, \overline{w}_{h,1}^t(z_h^t)/\overline{R}_h^t\}$ and $\sum_{t=1}^{T}\sum_{h=1}^{H}\overline{R}_h^t\min\{1, \underline{w}_{h,1}^t(z_h^t)/\overline{R}_h^t\}$ separately. By definitions in Lemma 11, Section A, we know that

$$\sum_{t=1}^{T}\sum_{h=1}^{H}\overline{R}_h^t\min\{1, \overline{w}_{h,1}^t(z_h^t)/\overline{R}_h^t\}$$
$$= \sum_{t=1}^{T}\sum_{h=1}^{H}\overline{R}_h^t\min\left\{1, \left[\phi_{\overline{V}_{h+1}^t}(z)^\top\overline{\mathbf{\Lambda}}_{h,1}^t\phi_{\overline{V}_{h+1}^t}(z)\right]^{1/2}/\overline{R}_h^t\right\}$$
$$\leq \sqrt{\sum_{t=1}^{T}\sum_{h=1}^{H}\left(\overline{R}_h^t\right)^2}\sqrt{\sum_{t=1}^{T}\sum_{h=1}^{H}\min\left\{1, \left[\phi_{\overline{V}_{h+1}^t}(z)^\top\overline{\mathbf{\Lambda}}_{h,1}^t\phi_{\overline{V}_{h+1}^t}(z)\right]/\overline{R}_h^t\right\}}$$
$$\leq \sqrt{\sum_{t=1}^{T}\sum_{h=1}^{H}\left(\overline{R}_h^t\right)^2}\cdot\sqrt{2H\Gamma_{\mathbf{K}}(T, \lambda_1\alpha^2)}.$$

Similarly,

$$\sum_{t=1}^{T}\sum_{h=1}^{H}\underline{R}_h^t \min\{1, \underline{w}_{h,1}^t(z_h^t)/\underline{R}_h^t\} \le \sqrt{\sum_{t=1}^{T}\sum_{h=1}^{H}\left(\underline{R}_h^t\right)^2} \cdot \sqrt{2H\Gamma_{\mathbf{K}}(T, \lambda_1\alpha^2)}.$$

By Lemma 21, we have

$$\sum_{t=1}^{T}\sum_{h=1}^{H}\left(\overline{R}_h^t\right)^2 + \left(\underline{R}_h^t\right)^2$$

$$= O\bigg(HT\alpha^2 + H^2T + H^3\log(1/\delta) + H\sum_{t=1}^{T}\sum_{h=1}^{H}\mathbb{P}_h[\overline{V}_{h+1}^t - \underline{V}_{h+1}^t]$$

$$+ \beta_T^{(2)}\sqrt{TH}\sqrt{H\Gamma_{\mathbf{K}}(T, \lambda_2/H^2)} + \beta_t^{(1)}H^2\sqrt{TH}\sqrt{H\Gamma_{\mathbf{K}}(T, \lambda_1/\alpha^2)}\bigg)$$

$$\le O\bigg(HT\alpha^2 + H^2T + H^3\log(1/\delta))$$

$$+ H^2\beta_t\sqrt{\sum_{t=1}^{T}\sum_{h=1}^{H}\left(\overline{R}_h^t\right)^2 + \left(\underline{R}_h^t\right)^2} \cdot \sqrt{H \cdot \Gamma_{\mathbf{K}}(T, \lambda_1\alpha^2)} + H^3\sqrt{HT\log(H/\delta)}$$

$$+ \beta_t^{(2)}\sqrt{TH}\sqrt{H\Gamma_{\mathbf{K}}(T, \lambda_2/H^2)} + \beta_t^{(1)}H^2\sqrt{TH}\sqrt{H\Gamma_{\mathbf{K}}(T, \lambda_1\alpha^2)}\bigg), \qquad \text{(E.8)}$$

where the inequality holds due to Cauchy-Schwarz inequality. Next, by taking

$$\alpha = H/\sqrt{\Gamma_{\mathbf{K}}(T, 1/B^2)}, \ \lambda_2 = H^2/B^2, \ \lambda_1 = 1/(\alpha^2 B^2),$$

we have

$\beta_t^{(1)} = 16H/\alpha\sqrt{\Gamma_{\mathbf{K}}(T, \lambda_1(\alpha)^2)}\sqrt{\log(4t^2H/\delta)} + 8H/\alpha\log(4t^2H/\delta) + \sqrt{\lambda_1} \cdot B = \widetilde{O}(\Gamma_{\mathbf{K}}(T, H^2/B^2))$,

$\beta_t^{(2)} = 16H^2\sqrt{\Gamma_{\mathbf{K}}(T, \lambda_2/H^2)}\sqrt{\log(4t^2H/\delta)} + 8H^2\log(4t^2H/\delta) + \sqrt{\lambda_2} \cdot B = \widetilde{O}(H^2)$,

$\beta_t = 16\sqrt{\Gamma_{\mathbf{K}}(T, \lambda_1(\alpha)^2)}\sqrt{\log(4t^2H/\delta)} + 8H/\alpha\log(4t^2H/\delta) + \sqrt{\lambda_1} \cdot B = \widetilde{O}(\sqrt{\Gamma_{\mathbf{K}}(T, H^2/B^2)})$.

For simplicity, let $d_{\text{eff}} := \Gamma_{\mathbf{K}}(T, H^2/B^2)$, then by (E.8) we have

$$\sum_{t=1}^{T}\sum_{h=1}^{H}\left(\overline{R}_h^t\right)^2 + \left(\underline{R}_h^t\right)^2$$

$$\le \widetilde{O}\left(\sqrt{\sum_{t=1}^{T}\sum_{h=1}^{H}\left(\overline{R}_h^t\right)^2 + \left(\underline{R}_h^t\right)^2 H^{5/2}d_{\text{eff}}} + H^3 d_{\text{eff}}^{3/2}T^{1/2} + H^{7/2}T^{1/2} + H^3 T/d_{\text{eff}} + H^2 T\right).$$

With the fact that $x \le a\sqrt{x} + b$ leads to $x = O(a^2 + b)$, we have

$$\sum_{t=1}^{T}\sum_{h=1}^{H}\left(\overline{R}_h^t\right)^2 + \left(\underline{R}_h^t\right)^2 = \widetilde{O}(d_{\text{eff}}^2 H^5 + H^3 d_{\text{eff}}^{3/2}T^{1/2} + H^{7/2}T^{1/2} + H^3 T/d_{\text{eff}} + H^2 T). \quad \text{(E.9)}$$

Finally, substituting (E.9) into (E.6) and bound the summation of $\xi_h^t, \gamma_h^t$ by Azuma-Hoeffding inequality, we have

$$\sum_{t=1}^{T}V_1^{*,\nu^t}(x_h^t) - V_1^{\pi^t,*}(x_h^t)$$

$$\le \widetilde{O}\left(\beta_t\sqrt{\sum_{t=1}^{T}\sum_{h=1}^{H}\left(\overline{R}_h^t\right)^2 + \left(\underline{R}_h^t\right)^2} \cdot \sqrt{H \cdot d_{\text{eff}}} + H\sqrt{2HT\log(H/\delta)}\right)$$

$$= \widetilde{O}\left(d_{\text{eff}}^2 H^3 + d_{\text{eff}}^{1.75}H^2 T^{0.25} + d_{\text{eff}}H^{2.25}T^{0.25} + \sqrt{d_{\text{eff}}}H^2\sqrt{T} + d_{\text{eff}}H^{1.5}\sqrt{T}\right)$$

$$= \widetilde{O}\left(d_{\text{eff}}^2 H^3 + \sqrt{d_{\text{eff}}H^4 + d_{\text{eff}}^2 H^3}\sqrt{T} + \left(d_{\text{eff}}^7 H^8 + d_{\text{eff}}^4 H^9\right)^{1/4}T^{1/4}\right).$$

This completes the proof of the theorem. $\qquad\square$

# F   Proof of Results for `KernelCCE-VTR` with Misspecification

In this section we prove Theorem 10.

**Lemma 22.** Assuming that for any $h \in [H]$, $\|\boldsymbol{\theta}_h^*\|_{\mathcal{H}} \leq B$. Let $\lambda = 1 + 1/T$ and $\beta_t$ satisfies

$$\left(\frac{\beta_t}{H}\right)^2 \geq 3\Gamma_{\mathbf{K}}(T, \lambda) + 3 + 6 \cdot \log\left(\frac{1}{\delta}\right) + 3\lambda \left(\frac{B}{H}\right)^2 + 3\iota_{\mathrm{mis}}^2 t. \tag{F.1}$$

Then for any $\delta > 0$, with probability at least $1 - \delta$ the following holds for any $(t, h) \in [T] \times [H]$ and any $z \in \mathcal{Z}$:

$$\left| \left\langle \phi_{\overline{V}_{h+1}^t}(z), \overline{\boldsymbol{\theta}}_h^t \right\rangle_{\mathcal{H}} - \mathbb{P}_h \overline{V}_{h+1}^t(z) \right| \leq \beta_t \cdot \overline{w}_h^t(z) + H \cdot \iota_{\mathrm{mis}},$$

$$\left| \left\langle \phi_{\underline{V}_{h+1}^t}(z), \underline{\boldsymbol{\theta}}_h^t \right\rangle_{\mathcal{H}} - \mathbb{P}_h \underline{V}_{h+1}^t(z) \right| \leq \beta_t \cdot \underline{w}_h^t(z) + H \cdot \iota_{\mathrm{mis}}.$$

The proof of Theorem 10 shares similar techniques with the proof of Theorem 4, except that we need Lemma 22 instead of Lemma 3. We detail the whole proof for completeness.

We recall that the duality gap is defined as $\sum_{t=1}^T V_1^{*,\nu^t}(x_1^t) - V_1^{\pi^t,*}(x_1^t)$. As can be seen in our Algorithm, we maintain an optimistic estimate of $V_h^{*,\nu^t}(\cdot)$ as $\overline{V}_{h+1}^t(\cdot)$ and a pessimistic estimate of $V_h^{\pi^t,*}(\cdot)$ as $\underline{V}_{h+1}^t(\cdot)$. Hence the term $\overline{V}_{h+1}^t(x_h^t) - \underline{V}_{h+1}^t(x_h^t)$ is approximately the upper bound of the duality gap. We write the decomposition formally as below:

$$V_h^{*,\nu^t}(x_h^t) - V_h^{\pi^t,*}(x_h^t) = \underbrace{\overline{V}_h^t(x_h^t) - \underline{V}_h^t(x_h^t)}_{\mathrm{I}} - \underbrace{\left( V_h^{\pi^t,*}(x_h^t) - \underline{V}_h^t(x_h^t) \right)}_{\mathrm{II}} - \underbrace{\left( \overline{V}_h^t(x_h^t) - V_h^{*,\nu^t}(x_h^t) \right)}_{\mathrm{III}}. \tag{F.2}$$

We use $\overline{\delta}_h^t$ to denote an important quantity $\left\langle \phi_{\overline{V}_{h+1}^t}(z_h^t), \boldsymbol{\theta}_h^* - \overline{\boldsymbol{\theta}}_h^t \right\rangle_{\mathcal{H}}$ (and $\left\langle \phi_{\underline{V}_{h+1}^t}(z_h^t), \boldsymbol{\theta}_h^* - \underline{\boldsymbol{\theta}}_h^t \right\rangle_{\mathcal{H}}$) in estimating the duality gap. In the rest of the proof we aim to show that all of the above three terms can be bounded by a quantity related to $\overline{\delta}_h^t$ ($\underline{\delta}_h^t$) and a stochastic random variable that forms a martingale difference sequence when considering for all $h \in [H], t \in [T]$.

For bounding term I, we first define two sequences of zero mean variables:

$$\gamma_h^t := \overline{Q}_h^t(x_h^t, a_h^t, b_h^t) - \underline{Q}_h^t(x_h^t, a_h^t, b_h^t) - \mathbb{E}_{(a,b)}\left[ \overline{Q}_h^t(x, a, b) - \underline{Q}_h^t(x, a, b) \right],$$

$$\xi_h^t := \left( \mathbb{P}_h(\overline{V}_{h+1}^t - \underline{V}_{h+1}^t) \right)(x_h^t, a_h^t, b_h^t) - \left( \overline{V}_{h+1}^t(x_{h+1}^t) - \underline{V}_{h+1}^t(x_{h+1}^t) \right), \tag{F.3}$$

where $\gamma_h^t$ depicts the stochastic error with respect to the policy and $\xi_h^t$ depicts the stochastic error with respect to the transition. We refer the readers to the proof of Lemma 23 for detailed explanations on these two error term. Given the above definition, we have the following Lemma 23.

**Lemma 23.** Under the settings of Lemma 22, we have the following recursive bound for $\forall h \in [H]$:

$$\overline{V}_h^t(x_h^t) - \underline{V}_h^t(x_h^t)$$
$$\leq \overline{V}_{h+1}^t(x_{h+1}^t) - \underline{V}_{h+1}^t(x_{h+1}^t) + 2\beta_t \min\{1, \overline{w}_h^t(x_h^t)\} + 2\beta_t \min\{1, \underline{w}_h^t(x_h^t)\} + 2H \cdot \iota_{\mathrm{mis}} + \xi_h^t + \gamma_h^t. \tag{F.4}$$

*Proof of Lemma 23.* Follows the same derivative as in Lemma 16 as it still holds that $\beta_t \geq H$ in Lemma 22 $\qquad\square$

For bounding II and III, we observe that the gap between two consecutive points of the values of term II can be decomposed as

$$
\left(\overline{V}_h^t(x_h^t) - V_h^{*,\nu_h^t}(x_h^t)\right) - \left(\overline{V}_{h+1}^t(x_{h+1}^t) - V_{h+1}^{*,\nu_h^t}(x_{h+1}^t)\right)
$$
$$
= \underbrace{\left(\overline{V}_h^t(x_h^t) - V_h^{*,\nu_h^t}(x_h^t)\right) - \left(\overline{Q}_h^t(z_h^t) - Q_h^{*,\nu_h^t}(z_h^t)\right)}_{\mathrm{I}_1}
$$
$$
+ \underbrace{\left(\overline{Q}_h^t(z_h^t) - Q_h^{*,\nu_h^t}(z_h^t)\right) - \left(\mathbb{P}_h(\overline{V}_{h+1}^t - V_{h+1}^{*,\nu_h^t})\right)(z_h^t)}_{\mathrm{I}_2} \tag{F.5}
$$
$$
+ \underbrace{\left(\mathbb{P}_h(\overline{V}_{h+1}^t - V_{h+1}^{*,\nu_h^t})\right)(z_h^t) - \left(\overline{V}_{h+1}^t(x_{h+1}^t) - V_{h+1}^{*,\nu_h^t}(x_{h+1}^t)\right)}_{\mathrm{I}_3}.
$$

In the two-player game setting II and III are symmetric, so the terms $\mathrm{I}_1, \mathrm{I}_2, \mathrm{I}_3$ has their correspondence denoted as $\mathrm{I}_1', \mathrm{I}_2', \mathrm{I}_3'$ separately. Utilizing the bound in Lemma 3, we can bound $|\mathrm{I}_2|$ and $|\mathrm{I}_2'|$ by $2\beta_t \min\{1, \overline{w}_h^t(z_h^t)\} + H \cdot \iota_{\mathrm{mis}}$ and $2\beta_t \min\{1, \underline{w}_h^t(z_h^t)\} + H \cdot \iota_{\mathrm{mis}}$ respectively via similar techniques as in the Proof of Lemma 16 and the fact that $\left|\overline{Q}_h^t(z_h^t) - Q_h^{*,\nu_h^t}(z_h^t)\right| \le 2H$, $\left|\underline{Q}_h^t(z_h^t) - Q_h^{\pi_h^t,*}(z_h^t)\right| \le 2H$ as well as the following inequalities:

$$
\left|\left(\overline{Q}_h^t(z_h^t) - Q_h^{*,\nu^t}(z_h^t)\right) - \left(\mathbb{P}_h(\overline{V}_{h+1}^t - V_{h+1}^{*,\nu^t})\right)(z_h^t)\right| \le 2\beta_t \overline{w}_h^t(z_h^t) + H \cdot \iota_{\mathrm{mis}},
$$
$$
\left|\left(\underline{Q}_h^t(z_h^t) - Q_h^{\pi^t,*}(z_h^t)\right) - \left(\mathbb{P}_h(\underline{V}_{h+1}^t - V_{h+1}^{\pi^t,*})\right)(z_h^t)\right| \le 2\beta_t \underline{w}_h^t(z_h^t) + H \cdot \iota_{\mathrm{mis}}. \tag{F.6}
$$

For $\mathrm{I}_3$ and $\mathrm{I}_3'$, we note that both terms are stochastic noises with mean zero, where the stochasticity lies in the transition probability. We denote them as $\alpha_{h,t}^1$ and $\alpha_{h,t}^2$ respectively.

Finally for bounding $\mathrm{I}_1$ and $\mathrm{I}_1'$, we utilize the properties of the CCE. (For simplicity we only prove the bound for $\mathrm{I}_1$, it is trivial to generalize to the bound for $\mathrm{I}_1$)

$$
\overline{V}_h^t(x_h^t) - V_h^{*,\nu_h^t}(x_h^t) = \mathbb{E}_{(a,b)\sim\sigma_h^t(x_h^t)}\overline{Q}_h^t(x_h^t, a, b) - \mathbb{E}_{a\sim\mathrm{br}(\nu_h^t),b\sim\nu_h^t}Q_h^{*,\nu_h^t}(x_h^t, a, b),
$$
$$
\ge \mathbb{E}_{a\sim\mathrm{br}(\nu_h^t),b\sim\nu_h^t}\overline{Q}_h^t(x_h^t, a, b) - \mathbb{E}_{a\sim\mathrm{br}(\nu_h^t),b\sim\nu_h^t}Q_h^{*,\nu_h^t}(x_h^t, a, b)
$$
$$
= \mathbb{E}_{a\sim\mathrm{br}(\nu_h^t),b\sim\nu_h^t}\left[\overline{Q}_h^t(x_h^t, a, b) - Q_h^{*,\nu_h^t}(x_h^t, a, b)\right],
$$

where $\nu_h^t := \mathcal{P}_2\sigma_h^t$ and $\pi_h^t := \mathcal{P}_1\sigma_h^t$ is the projection of $\sigma_h^t$ on the first and second coordinate respectively and br is the best response policy of a given distribution. Defining

$$
\zeta_{h,t}^1 := \mathbb{E}_{a\sim\mathrm{br}(\nu_h^t),b\sim\nu_h^t}\left[\overline{Q}_h^t(x_h^t, a, b) - Q_h^{*,\nu_h^t}(x_h^t, a, b)\right] - \left[\overline{Q}_h^t(x_h^t, a_h^t, b_h^t) - Q_h^{*,\nu_h^t}(x_h^t, a_h^t, b_h^t)\right],
$$
$$
\zeta_{h,t}^2 := \mathbb{E}_{a\sim\pi_h^t,b\sim\mathrm{br}(\pi_h^t)}\left[Q_h^{\pi_h^t,*}(x_h^t, a, b) - \underline{Q}_h^t(x_h^t, a, b)\right] - \left[Q_h^{\pi_h^t,*}(x_h^t, a_h^t, b_h^t) - \underline{Q}_h^t(x_h^t, a_h^t, b_h^t)\right],
$$

we are at the conclusion that

$$
\overline{V}_h^t(x_h^t) - V_h^{*,\nu_h^t}(x_h^t) \ge \overline{Q}_h^t(x_h^t, a_h^t, b_h^t) - Q_h^{*,\nu_h^t}(x_h^t, a_h^t, b_h^t) + \zeta_{h,t}^1,
$$
$$
V_h^{\pi_h^t,*}(x_h^t) - \underline{V}_h^t(x_h^t) \ge Q_h^{\pi_h^t,*}(x_h^t, a_h^t, b_h^t) - \underline{Q}_h^t(x_h^t, a_h^t, b_h^t) + \zeta_{h,t}^2.
$$

Bringing this lower bound result together with the previous absolute bound (F.6) into Eq. (F.5) and its counterpart for the min-player, we have

$$
\left(V_h^{*,\nu_h^t}(x_h^t) - \overline{V}_h^t(x_h^t)\right) - \left(V_{h+1}^{*,\nu_h^t}(x_{h+1}^t) - \overline{V}_{h+1}^t(x_{h+1}^t)\right)
$$
$$
+ \left(\underline{V}_h^t(x_h^t) - V_h^{\pi_h^t,*}(x_h^t)\right) - \left(\underline{V}_{h+1}^t(x_{h+1}^t) - V_{h+1}^{\pi_h^t,*}(x_{h+1}^t)\right) \tag{F.7}
$$
$$
\ge \alpha_{h,t}^1 + \alpha_{h,t}^2 + \zeta_{h,t}^1 + \zeta_{h,t}^2 - 2\beta_t \min\{1, \overline{w}_h^t(z_h^t)\} - 2\beta_t \min\{1, \underline{w}_h^t(z_h^t)\} - 2H \cdot \iota_{\mathrm{mis}}.
$$

Combining with Eq. (F.2), we arrive at a bound in terms of $\overline{w}_h^t(z_h^t), \underline{w}_h^t(z_h^t)$ and the martingale difference sequences:

$$\sum_{h=1}^H V_h^{*,\nu^t}(x_h^t) - V_h^{\pi^t,*}(x_h^t)$$

$$\leq \sum_{h=1}^H \left( 4\beta_t \min\{1, \overline{w}_h^t(x_h^t)\} + 4\beta_t \min\{1, \underline{w}_h^t(x_h^t)\} + 4H \cdot \iota_{\mathrm{mis}} + \xi_h^t + \gamma_h^t + \alpha_{h,t}^1 + \alpha_{h,t}^2 + \zeta_{h,t}^1 + \zeta_{h,t}^2 \right),$$

where $\nu^t$ is the policy that operates according to $\nu_h^t$ at time $h$ and $\pi^t$ is the sequence of $\pi_h^t$ accordingly. The rest of the proof follows by bounding $\sum_{t=1}^T \sum_{h=1}^H \min\{1, \overline{w}_h^t(z_h^t)\}, \sum_{t=1}^T \sum_{h=1}^H \min\{1, \underline{w}_h^t(z_h^t)\}$ and the martingale difference sequences. Following the same techniques as in the proof of Theorem 4, we again apply Lemma 17 and Cauchy-Schwarz inequality, together with the Azuma-Hoeffding for bounded martingale differences, we arrive at our final result.

## G  Proof of Results for `KernelCCE-VTR` with Neural Approximation

In this section, we proof our result for the neural network approximation.

*Proof of Theorem 15.* Let $C$ be defined in Lemma 14. We define a rescaled version of $\phi$ as $\widetilde{\phi}/C$, then we know that for any bounded value functions $V_h(\cdot) : \mathcal{S} \mapsto [-1, 1]$,

$$|\mathbb{P}_h V_h(z) - \langle \widetilde{\phi}_{V_h}(z), C(\boldsymbol{\theta}_h^* - \boldsymbol{\theta}^{(0)}) \rangle| \leq C_1 |\mathcal{S}| B^{4/3} m^{-1/6} L^3 \sqrt{\log m}, \qquad \|\widetilde{\phi}_{V_h}(z)\|_2 \leq 1.$$

Defining $\widetilde{\boldsymbol{\theta}}_h^* := C\boldsymbol{\theta}_h^*$ and $\widetilde{\boldsymbol{\theta}}^{(0)} := C\boldsymbol{\theta}^{(0)}$ and taking $\iota_{\mathrm{mis}} := C \cdot B^{4/3} \cdot m^{-1/6} \cdot \sqrt{\log m}$, by Theorem 10 we know that for $\widetilde{\lambda} = 1 + \frac{1}{T}$, any $\delta > 0$ and any $\beta$ satisfying

$$\left( \frac{\beta}{H} \right)^2 \geq 2\Gamma_{\widetilde{K}}(T, \widetilde{\lambda}) + 3 + 6 \cdot \log\left( \frac{1}{\delta} \right) + 3\lambda \left( \frac{\widetilde{B}}{H} \right)^2 + 3C^2 \cdot B^{8/3} \cdot m^{-1/12} \cdot \log m \cdot t,$$

there exists a global constant $c > 0$ such that with probability at least $1 - \delta$, we have

$$\mathrm{Regret}(T) \leq c \left( \beta H \sqrt{T \cdot \Gamma_{\widetilde{K}}(T, \lambda)} + 1 + H^2 T \iota_{\mathrm{mis}} \right).$$

where $\widetilde{B} = C \cdot B$, and

$$\Gamma_{\widetilde{K}}(T, \lambda) := \sup_{(V_i)_i, (z_i)_i} \frac{1}{2} \log \det(\mathbf{I} + \widetilde{\mathbf{K}}(\{V_i\}_i, \{z_i\}_i)/\lambda), \tag{G.1}$$

where $\widetilde{\mathbf{K}} \in \mathbb{R}^{T \times T}$ is the matrix based on the kernel function $k$ induced by the feature mapping $\widetilde{\phi}$, where

$$\boldsymbol{k}_{V_1, V_2}(z_1, z_2) = \langle \widetilde{\phi}_{V_1}(z_1), \widetilde{\phi}_{V_2}(z_2) \rangle.$$

Rescaling gives

$$\Gamma_{\widetilde{K}}(T, \widetilde{\lambda}) = \Gamma_{\mathbf{K}}(T, C^2 \widetilde{\lambda}).$$

Choosing $\lambda := C^2(1 + \frac{1}{T})$ completes our proof of Theorem 15.

$\square$

## H  Proof of Auxillary Lemmas

In this section, we prove the essential lemmas in the proof of our main theorems. First of all we present a concentration bound for self-normalized processes in an RKHS $\mathcal{H}$, which is critical in determining the main term of the regret bound.

**Theorem 24** (Self-Normalized Concentration Bounds for RKHS [Chowdhury and Gopalan, 2017, Yang et al., 2020]). Let $\{\mathbf{x}_t\}_{t \geq 1}$ be a discrete time stochastic process taking values in $\mathcal{Z}$, $\mathcal{H}$ is an RKHS with kernel $\mathbf{K}(\cdot, \cdot) : \mathcal{Z} \times \mathcal{Z} \mapsto \mathbb{R}$, $\{\mathcal{F}_t\}_{t \geq 0}$ is a given filtration. We assume that $\mathbf{x}_t$ is $\mathcal{F}_{t-1}$ measurable in the sense that for $\forall t \geq 1$, $\mathbf{x}_t \in \mathcal{F}_{t-1}$. Furthermore, $\{\epsilon_t\}_{t \geq 1}$ is a real-valued stochastic process with each $\epsilon_t$ $\mathcal{F}_t$ measurable and $\sigma$-sub-Gaussian. Define $\mathbf{K}_t \in \mathbb{R}^{(t-1) \times (t-1)}$ as the Gram matrix for data $\{\mathbf{x}_\tau\}_{\tau \in [t-1]}$ of the RKHS $\mathcal{H}$. Then for any $\lambda > 1$ and $\delta \in (0, 1)$, with probability at least $1 - \delta$, the following holds simultaneosly for all $t \geq 0$:

$$\|\varepsilon_{1:t-1}\|^2_{\left((\mathbf{K}_t + (\lambda-1) \cdot \mathbf{I})^{-1} + \mathbf{I}\right)^{-1}} \leq 2\sigma^2 \log \frac{\sqrt{\det(\lambda \mathbf{I} + \mathbf{K}_t)}}{\delta}.$$

*Proof.* See Lemma E.1 in Yang et al. [2020] and Theorem 1 in Chowdhury and Gopalan [2017] for the detailed proof. $\qquad \square$

## H.1 Proof of Lemma 3

*Proof of Lemma 3.* We only provide the proof of the max-player, and results of the min-player can be derived similarly. We recall the definition of $\overline{\mathbf{y}}_h^t$ is the vector of regression targets $\left(\overline{V}_{h+1}^1(x_{h+1}^1), \ldots, \overline{V}_{h+1}^{t-1}(x_{h+1}^{t-1})\right)^\top \in \mathbb{R}^{t-1}$. Furthermore by Lemma 11 in Section A, we know that $\overline{\theta}_h^t = \left(\overline{\boldsymbol{\Psi}}_h^t\right)^\top \left[\overline{\mathbf{K}}_h^t + \lambda \mathbf{I}\right]^{-1} \overline{\mathbf{y}}_h^t$ and $\phi_{\overline{V}_{h+1}^t}(z) = \left(\overline{\boldsymbol{\Psi}}_h^t\right)^\top (\overline{\mathbf{K}}_h^t + \lambda \mathbf{I})^{-1} \overline{\mathbf{k}}_h^t(z) + \lambda \cdot (\overline{\boldsymbol{\Lambda}}_h^t)^{-1} \phi_{\overline{V}_{h+1}^t}(z)$, which enable us to bound the difference $\left\langle \phi_{\overline{V}_{h+1}^t}(z), \overline{\theta}_h^t - \theta_h^* \right\rangle_{\mathcal{H}}$ as follows:

$$\begin{aligned}
\left\langle \phi_{\overline{V}_{h+1}^t}(z), \overline{\theta}_h^t - \theta_h^* \right\rangle_{\mathcal{H}} &= \phi_{\overline{V}_{h+1}^t}(z)^\top \left(\overline{\boldsymbol{\Psi}}_h^t\right)^\top \left[\overline{\mathbf{K}}_h^t + \lambda \mathbf{I}\right]^{-1} \overline{\mathbf{y}}_h^t \\
&\quad - \left(\overline{\theta}_h^*\right)^\top \left[\left(\overline{\boldsymbol{\Psi}}_h^t\right)^\top \left[\overline{\mathbf{K}}_h^t + \lambda \mathbf{I}\right]^{-1} \overline{\mathbf{k}}_h^t(z) + \lambda \cdot (\overline{\boldsymbol{\Lambda}}_h^t)^{-1} \phi_{\overline{V}_{h+1}^t}(z)\right] \\
&= \underbrace{(\overline{\mathbf{k}}_h^t)^\top \left[\overline{\mathbf{K}}_h^t + \lambda \mathbf{I}\right]^{-1} \left[\overline{\mathbf{y}}_h^t - \overline{\boldsymbol{\Psi}}_h^t \theta_h^*\right]}_{\mathbf{I}_1} - \underbrace{\lambda \cdot \phi_{\overline{V}_{h+1}^t}(z)^\top (\overline{\boldsymbol{\Lambda}}_h^t)^{-1} \theta_h^*}_{\mathbf{I}_2}.
\end{aligned}$$

For bounding $\mathbf{I}_2$, we apply the Cauchy-Schwarz inequality and have

$$\begin{aligned}
\lambda \cdot \phi_{\overline{V}_{h+1}^t}(z)^\top (\overline{\boldsymbol{\Lambda}}_h^t)^{-1} \theta_h^* &\leq \left\| \lambda \cdot \phi_{\overline{V}_{h+1}^t}(z)^\top (\overline{\boldsymbol{\Lambda}}_h^t)^{-1} \right\|_{\mathcal{H}} \cdot \|\theta_h^*\|_{\mathcal{H}} \\
&\overset{(a)}{\leq} B \cdot \sqrt{\lambda \phi_{\overline{V}_{h+1}^t}(z)^\top (\overline{\boldsymbol{\Lambda}}_h^t)^{-1} \lambda (\overline{\boldsymbol{\Lambda}}_h^t)^{-1} \phi_{\overline{V}_{h+1}^t}(z)} \overset{(b)}{\leq} \sqrt{\lambda} B \cdot \overline{w}_h^t(z),
\end{aligned}$$

where $(a)$ is due to the assumption that $\|\theta_h^*\|_{\mathcal{H}} \leq B$ and $(b)$ is by the definition of $\overline{w}_h^t$ and the fact that $\left(\overline{\boldsymbol{\Lambda}}_h^t\right)^{-1}$ is a self-adjoint mapping on the RKHS $\mathcal{H}$ satisfying $\left\| \left(\overline{\boldsymbol{\Lambda}}_h^t\right)^{-1} \right\|_{op} \leq \frac{1}{\lambda}$.

For bounding $\mathbf{I}_1$, we observe the following equality:

$$\begin{aligned}
(\overline{\mathbf{k}}_h^t)^\top \left[\overline{\mathbf{K}}_h^t + \lambda \mathbf{I}\right]^{-1} \left[\overline{\mathbf{y}}_h^t - \overline{\boldsymbol{\Psi}}_h^t \theta_h^*\right] &= \phi_{\overline{V}_{h+1}^t}(z)^\top (\overline{\boldsymbol{\Lambda}}_h^t)^{-1} (\overline{\boldsymbol{\Psi}}_h^t)^\top \left[\overline{\mathbf{y}}_h^t - \boldsymbol{\Psi}_h^t \theta_h^*\right] \\
&= \phi_{\overline{V}_{h+1}^t}(z)^\top (\overline{\boldsymbol{\Lambda}}_h^t)^{-1} \sum_{\tau=1}^{t-1} \phi_{\overline{V}_{h+1}^\tau}(z_h^\tau) \left[\overline{V}_{h+1}^\tau(x_{h+1}^\tau) - (\mathbb{P}_h \overline{V}_{h+1}^\tau)(z_h^\tau)\right].
\end{aligned}$$

Again by applying the Cauchy-Schwarz inequality, we bound the RHS of the above equality as

$$|\mathbf{I}_1| \leq \left\| \phi_{\overline{V}_{h+1}^t}(z) \right\|_{(\boldsymbol{\Lambda}_h^t)^{-1}} \cdot \left\| \sum_{\tau=1}^{t-1} \phi_{\overline{V}_{h+1}^\tau}(z_h^\tau) \left[\overline{V}_{h+1}^\tau(x_{h+1}^\tau) - (\mathbb{P}_h \overline{V}_{h+1}^\tau)(z_h^\tau)\right] \right\|_{(\boldsymbol{\Lambda}_h^t)^{-1}}.$$

We note that for a given $h$ that we consider in Lemma 3. If we define $\{\mathcal{F}_t\}_{t \geq 0}$ as the $\sigma$-algebra generated by all data before iteration $t$ and all data before step $h$ at iteration $t + 1$, $\overline{V}_{h+1}^\tau(x_{h+1}^\tau) - (\mathbb{P}_h \overline{V}_{h+1}^\tau)(z_h^\tau) \in \mathcal{F}_\tau$ is a mean zero random variable with respect to filtration

$\mathcal{F}_{\tau-1}$ with $\left|\overline{V}_{h+1}^{\tau}(x_{h+1}^{\tau}) - (\mathbb{P}_h\overline{V}_{h+1}^{\tau})(z_h^{\tau})\right| \le H$. We take $\epsilon_{\tau} = \overline{V}_{h+1}^{\tau}(x_{h+1}^{\tau}) - (\mathbb{P}_h\overline{V}_{h+1}^{\tau})(z_h^{\tau})$ in Theorem 24 and $\{\mathbf{x}_t\}_{t\ge 1}$ is $\left\{\phi_{\overline{V}_{h+1}^t}(z_h^t)\right\}_{t\ge 1}$. Then by directly utilizing Theorem 24, we have that

$$\left\|\sum_{\tau=1}^{t-1}\phi_{\overline{V}_{h+1}^{\tau}}(z_h^{\tau})\left[\overline{V}_{h+1}^t(x_{h+1}^{\tau}) - (\mathbb{P}_h\overline{V}_{h+1}^t)(z_h^{\tau})\right]\right\|_{(\boldsymbol{\Lambda}_h^t)^{-1}}^2 = \left\|(\boldsymbol{\Psi}_h^t)^{\top}\epsilon_h^t\right\|_{(\boldsymbol{\Lambda}_h^t)^{-1}}^2 = (\epsilon_h^t)^{\top}\boldsymbol{\Psi}_h^t\boldsymbol{\Lambda}_t^{-1}(\boldsymbol{\Psi}_h^t)^{\top}\epsilon_h^t.$$

By Lemma 11 and Theorem 24 again, we arrive at the following:

$$(\epsilon_h^t)^{\top}\boldsymbol{\Psi}_h^t\boldsymbol{\Lambda}_t^{-1}(\boldsymbol{\Psi}_h^t)^{\top}\epsilon_h^t \le (\epsilon_h^t)^{\top}\mathbf{K}_t(\mathbf{K}_t + \lambda\mathbf{I})^{-1}\epsilon_h^t \le (\epsilon_h^t)^{\top}\left(\mathbf{K}_t + (\lambda-1)\cdot\mathbf{I}\right)(\mathbf{K}_t + \lambda\mathbf{I})^{-1}\epsilon_h^t$$

$$= (\epsilon_h^t)^{\top}\left(\left(\mathbf{K}_t + (\lambda-1)\cdot\mathbf{I}\right)^{-1} + \mathbf{I}\right)^{-1}\epsilon_h^t$$

$$\le H^2\log\frac{\sqrt{\det(\lambda\mathbf{I} + \mathbf{K}_t)}}{\delta} \le 2H^2\cdot\text{logdet}(\lambda\mathbf{I} + \mathbf{K}_t) + 2H^2\cdot\log\frac{1}{\delta}.$$

By taking $\lambda = 1 + \frac{1}{T}$,

$$\left|\phi(z)^{\top}(\boldsymbol{\theta}_h^* - \overline{\boldsymbol{\theta}}_h^t)\right|$$

$$\le \left\{\left[H^2\cdot\text{logdet}\left[\lambda\mathbf{I} + \mathbf{K}_t\right] + 2H^2\cdot\log\left(\frac{1}{\delta}\right)\right]^{1/2} + \sqrt{\lambda}B\right\}\cdot b_h^t(z)$$

$$\le \left\{\left[H^2\cdot\text{logdet}\left[\mathbf{I} + \mathbf{K}_t/\lambda\right] + (\lambda-1)tH^2 + 2H^2\cdot\log\left(\frac{1}{\delta}\right)\right]^{1/2} + \sqrt{\lambda}B\right\}\cdot b_h^t(z)$$

$$\le \left\{\left[H^2\cdot\Gamma_{\mathbf{K}}(T,\lambda) + H^2 + 2H^2\cdot\log\left(\frac{1}{\delta}\right)\right]^{1/2} + \sqrt{\lambda}B\right\}\cdot b_h^t(z).$$

Let $(\beta/H)^2 = 2\Gamma_{\mathbf{K}}(T,\lambda) + 2 + 4\cdot\log\left(\frac{1}{\delta}\right) + 2\lambda\left(\frac{B}{H}\right)^2$, we know from the above theoretical derivation that $\left|\phi(z)^{\top}(\boldsymbol{\theta}_h^* - \overline{\boldsymbol{\theta}}_h^t)\right| \le \beta\cdot b_h^t(z)$ with probability at least $1 - \delta$, which concludes our proof. $\square$

## H.2 Proof of Lemma 22

*Proof of Lemma 22.* The first half of the proof of Lemma 22 follows exactly the same as in the proof of Lemma 3, we restate the proof for completeness and analyze the error terms brought by misspecification in the later half of the proof. We only provide the proof of the max-player, and results of the min-player can be derived similarly. We recall the definition of $\overline{\mathbf{y}}_h^t$ is the vector of regression targets $\left(\overline{V}_{h+1}^1(x_{h+1}^1), \ldots, \overline{V}_{h+1}^{t-1}(x_{h+1}^{t-1})\right)^{\top} \in \mathbb{R}^{t-1}$. Furthermore by Lemma 11 in Section A, we know that $\overline{\theta}_h^t = \left(\overline{\boldsymbol{\Psi}}_h^t\right)^{\top}\left[\overline{\mathbf{K}}_h^t + \lambda\mathbf{I}\right]^{-1}\overline{\mathbf{y}}_h^t$ and $\phi_{\overline{V}_{h+1}^t}(z) = \left(\overline{\boldsymbol{\Psi}}_h^t\right)^{\top}(\overline{\mathbf{K}}_h^t + \lambda\mathbf{I})^{-1}\overline{\mathbf{k}}_h^t(z) + \lambda\cdot(\overline{\boldsymbol{\Lambda}}_h^t)^{-1}\phi_{\overline{V}_{h+1}^t}(z)$, which enable us to bound the difference $\left\langle\phi_{\overline{V}_{h+1}^t}(z), \overline{\theta}_h^t - \theta_h^*\right\rangle_{\mathcal{H}}$ as follows:

$$\left\langle\phi_{\overline{V}_{h+1}^t}(z), \overline{\theta}_h^t - \theta_h^*\right\rangle_{\mathcal{H}} = \phi_{\overline{V}_{h+1}^t}(z)^{\top}\left(\overline{\boldsymbol{\Psi}}_h^t\right)^{\top}\left[\overline{\mathbf{K}}_h^t + \lambda\mathbf{I}\right]^{-1}\overline{\mathbf{y}}_h^t$$

$$- \left(\overline{\theta}_h^*\right)^{\top}\left[\left(\overline{\boldsymbol{\Psi}}_h^t\right)^{\top}\left[\overline{\mathbf{K}}_h^t + \lambda\mathbf{I}\right]^{-1}\overline{\mathbf{k}}_h^t(z) + \lambda\cdot(\overline{\boldsymbol{\Lambda}}_h^t)^{-1}\phi_{\overline{V}_{h+1}^t}(z)\right]$$

$$= \underbrace{(\overline{\mathbf{k}}_h^t)^{\top}\left[\overline{\mathbf{K}}_h^t + \lambda\mathbf{I}\right]^{-1}\left[\overline{\mathbf{y}}_h^t - \overline{\boldsymbol{\Psi}}_h^t\theta_h^*\right]}_{\text{I}_1} - \underbrace{\lambda\cdot\phi_{\overline{V}_{h+1}^t}(z)^{\top}(\overline{\boldsymbol{\Lambda}}_h^t)^{-1}\theta_h^*}_{\text{I}_2}.$$

For bounding $\text{I}_2$, we apply the Cauchy-Schwarz inequality and have

$$\lambda\cdot\phi_{\overline{V}_{h+1}^t}(z)^{\top}(\overline{\boldsymbol{\Lambda}}_h^t)^{-1}\theta_h^* \le \left\|\lambda\cdot\phi_{\overline{V}_{h+1}^t}(z)^{\top}(\overline{\boldsymbol{\Lambda}}_h^t)^{-1}\right\|_{\mathcal{H}}\cdot\|\theta_h^*\|_{\mathcal{H}}$$

$$\overset{(a)}{\le} B\cdot\sqrt{\lambda\phi_{\overline{V}_{h+1}^t}(z)^{\top}(\overline{\boldsymbol{\Lambda}}_h^t)^{-1}\lambda(\overline{\boldsymbol{\Lambda}}_h^t)^{-1}\phi_{\overline{V}_{h+1}^t}(z)} \overset{(b)}{\le} \sqrt{\lambda}B\cdot\overline{w}_h^t(z),$$

where $(a)$ is due to the assumption that $\|\boldsymbol{\theta}_h^*\|_{\mathcal{H}} \leq B$ and $(b)$ is by the definition of $\overline{w}_h^t$ and the fact that $\left(\overline{\boldsymbol{\Lambda}}_h^t\right)^{-1}$ is a self-adjoint mapping on the RKHS $\mathcal{H}$ satisfying $\left\|\left(\overline{\boldsymbol{\Lambda}}_h^t\right)^{-1}\right\|_{op} \leq \frac{1}{\lambda}$.

For bounding $\mathrm{I}_1$, we observe the following equality:

$$
(\overline{\boldsymbol{k}}_h^t)^\top \left[\overline{\mathbf{K}}_h^t + \lambda\mathbf{I}\right]^{-1} \left[\overline{\mathbf{y}}_h^t - \overline{\boldsymbol{\Psi}}_h^t \boldsymbol{\theta}_h^*\right]
$$
$$
= \phi_{\overline{V}_{h+1}^t}(z)^\top (\overline{\boldsymbol{\Lambda}}_h^t)^{-1} (\overline{\boldsymbol{\Psi}}_h^t)^\top \left[\overline{\mathbf{y}}_h^t - \boldsymbol{\Psi}_h^t \boldsymbol{\theta}_h^*\right]
$$
$$
= \phi_{\overline{V}_{h+1}^t}(z)^\top (\overline{\boldsymbol{\Lambda}}_h^t)^{-1} \sum_{\tau=1}^{t-1} \phi_{\overline{V}_{h+1}^\tau}(z_h^\tau) \left[\overline{V}_{h+1}^\tau(x_{h+1}^\tau) - (\mathbb{P}_h\overline{V}_{h+1}^\tau)(z_h^\tau)\right]
$$
$$
+ \phi_{\overline{V}_{h+1}^t}(z)^\top (\overline{\boldsymbol{\Lambda}}_h^t)^{-1} \sum_{\tau=1}^{t-1} \phi_{\overline{V}_{h+1}^\tau}(z_h^\tau) \left[(\mathbb{P}_h\overline{V}_{h+1}^\tau)(z_h^\tau) - \left\langle \phi_{\overline{V}_{h+1}^\tau}, \boldsymbol{\theta}_h^*\right\rangle_{\mathcal{H}}\right].
$$

Again by applying the Cauchy-Schwarz inequality, we bound the RHS of the above equality as

$$
|\mathrm{I}_1| \leq \left\|\phi_{\overline{V}_{h+1}^t}(z)\right\|_{(\boldsymbol{\Lambda}_h^t)^{-1}} \cdot \underbrace{\left\|\sum_{\tau=1}^{t-1} \phi_{\overline{V}_{h+1}^\tau}(z_h^\tau) \left[\overline{V}_{h+1}^\tau(x_{h+1}^\tau) - (\mathbb{P}_h\overline{V}_{h+1}^\tau)(z_h^\tau)\right]\right\|_{(\boldsymbol{\Lambda}_h^t)^{-1}}}_{A_1}
$$
$$
\tag{H.1}
$$
$$
+ \left\|\phi_{\overline{V}_{h+1}^t}(z)\right\|_{(\boldsymbol{\Lambda}_h^t)^{-1}} \cdot \underbrace{\left\|\sum_{\tau=1}^{t-1} \phi_{\overline{V}_{h+1}^\tau}(z_h^\tau) \left[(\mathbb{P}_h\overline{V}_{h+1}^\tau)(z_h^\tau) - \phi_{\overline{V}_{h+1}^\tau}^\top \boldsymbol{\theta}_h^*\right]\right\|_{(\boldsymbol{\Lambda}_h^t)^{-1}}}_{A_2}.
$$

For bounding $A_1$, we note that for a given $h$ that we consider in Lemma 22. If we define $\{\mathcal{F}_t\}_{t\geq 0}$ as the $\sigma$-algebra generated by all data before iteration $t$ and all data before step $h$ at iteration $t+1$, $\overline{V}_{h+1}^\tau(x_{h+1}^\tau) - (\mathbb{P}_h\overline{V}_{h+1}^\tau)(z_h^\tau) \in \mathcal{F}_\tau$ is a mean zero random variable with respect to filtration $\mathcal{F}_{\tau-1}$ with $\left|\overline{V}_{h+1}^\tau(x_{h+1}^\tau) - (\mathbb{P}_h\overline{V}_{h+1}^\tau)(z_h^\tau)\right| \leq H$. We take $\epsilon_\tau = \overline{V}_{h+1}^\tau(x_{h+1}^\tau) - (\mathbb{P}_h\overline{V}_{h+1}^\tau)(z_h^\tau)$ in Theorem 24 and $\{\mathbf{x}_t\}_{t\geq 1}$ is $\left\{\phi_{\overline{V}_{h+1}^t}(z_h^t)\right\}_{t\geq 1}$. Then by directly utilizing Theorem 24, we have that

$$
\left\|\sum_{\tau=1}^{t-1} \phi_{\overline{V}_{h+1}^\tau}(z_h^\tau) \left[\overline{V}_{h+1}^t(x_{h+1}^\tau) - (\mathbb{P}_h\overline{V}_{h+1}^t)(z_h^\tau)\right]\right\|_{(\boldsymbol{\Lambda}_h^t)^{-1}}^2
$$
$$
= \left\|(\boldsymbol{\Psi}_h^t)^\top \epsilon_h^t\right\|_{(\boldsymbol{\Lambda}_h^t)^{-1}}^2 = (\epsilon_h^t)^\top \boldsymbol{\Psi}_h^t \boldsymbol{\Lambda}_t^{-1} (\boldsymbol{\Psi}_h^t)^\top \epsilon_h^t.
$$

By Lemma 11 and Theorem 24 again, we arrive at the following:

$$
(\epsilon_h^t)^\top \boldsymbol{\Psi}_h^t \boldsymbol{\Lambda}_t^{-1} (\boldsymbol{\Psi}_h^t)^\top \epsilon_h^t \leq (\epsilon_h^t)^\top \mathbf{K}_t (\mathbf{K}_t + \lambda\mathbf{I})^{-1} \epsilon_h^t \leq (\epsilon_h^t)^\top (\mathbf{K}_t + (\lambda-1)\cdot\mathbf{I}) (\mathbf{K}_t + \lambda\mathbf{I})^{-1} \epsilon_h^t
$$
$$
= (\epsilon_h^t)^\top \left((\mathbf{K}_t + (\lambda-1)\cdot\mathbf{I})^{-1} + \mathbf{I}\right)^{-1} \epsilon_h^t
$$
$$
\leq 2H^2 \log \frac{\sqrt{\det(\lambda\mathbf{I} + \mathbf{K}_t)}}{\delta} \leq H^2 \cdot \mathrm{logdet}(\lambda\mathbf{I} + \mathbf{K}_t) + 2H^2 \cdot \log \frac{1}{\delta}.
$$

For bounding the term $A_2$ in Eq. (H.1), we apply the following lemma, which is the RKHS version of the Lemma 8 in Zanette et al. [2020]:

**Lemma 25** (Lemma 8 in Zanette et al. [2020]). *Let $\{\boldsymbol{a}_i\}_{i=1,\dots,t}$ be any sequence of vectors in the RKHS $\mathcal{H}$ and $\{b_i\}_{i=1,\dots,t}$ be any sequence of scalars such that $|b_i| \leq \epsilon \in \mathbb{R}^+$. For any $\lambda \geq 0$ and $t \in \mathbb{N}$ we have:*

$$
\left\|\sum_{i=1}^t \boldsymbol{a}_i b_i\right\|_{\left[\sum_{i=1}^t \boldsymbol{a}_i \boldsymbol{a}_i^\top + \lambda\mathbf{I}\right]^{-1}} \leq t\epsilon^2.
$$

*Proof of Lemma 25.* By defining the feature matrix $\mathbf{A} := (\boldsymbol{a}_1, \ldots, \boldsymbol{a}_t)$ and the vector $\boldsymbol{b} := (b_1, \ldots, b_t)^\top$, we have the following formulation

$$\sum_{t=1}^{t} \boldsymbol{a}_i b_i = \mathbf{A}\boldsymbol{b}, \qquad \left\| \sum_{i=1}^{t} \boldsymbol{a}_i b_i \right\|_{\left[\sum_{i=1}^{t} \boldsymbol{a}_i \boldsymbol{a}_i^\top + \lambda \mathbf{I}\right]^{-1}} = \|\mathbf{A}\boldsymbol{b}\|_{[\mathbf{A}\mathbf{A}^\top + \lambda \mathbf{I}_{\mathcal{H}}]^{-1}} = \boldsymbol{b}^\top \mathbf{A}^\top [\mathbf{A}\mathbf{A}^\top + \lambda \mathbf{I}_{\mathcal{H}}]^{-1} \mathbf{A}\boldsymbol{b}.$$

Via the same reasoning as in the proof of item $(a)$ in Lemma 11,

$$\boldsymbol{b}^\top \mathbf{A}^\top [\mathbf{A}\mathbf{A}^\top + \lambda \mathbf{I}_{\mathcal{H}}]^{-1} \mathbf{A}\boldsymbol{b} = \boldsymbol{b}^\top [\mathbf{A}^\top \mathbf{A} + \lambda \mathbf{I}_{\mathcal{H}}]^{-1} \mathbf{A}^\top \mathbf{A}\boldsymbol{b}$$
$$= \boldsymbol{b}^\top \boldsymbol{b} - \lambda \cdot \boldsymbol{b}^\top [\mathbf{A}^\top \mathbf{A} + \lambda \mathbf{I}_{\mathcal{H}}]^{-1} \boldsymbol{b} \leq \|\boldsymbol{b}\|^2 \leq t\epsilon^2,$$

which concludes our proof of Lemma 25. $\qquad\square$

By letting $b_\tau = \left[ (\mathbb{P}_h \overline{V}_{h+1}^\tau)(z_h^\tau) - \phi_{\overline{V}_{h+1}^t}^\top \boldsymbol{\theta}_h^* \right]$ and $\boldsymbol{a}_\tau = \phi_{\overline{V}_{h+1}^\tau}(z_h^\tau)$ and knowing that $|b_\tau| \leq \iota_{\mathrm{mis}}$, we have the bound for item $A_2$ that $A_2 \leq \iota_{\mathrm{mis}} \cdot \sqrt{t}$. Then by taking $\lambda = 1 + \frac{1}{T}$,

$$\left| \phi(z)^\top (\boldsymbol{\theta}_h^* - \overline{\boldsymbol{\theta}}_h^t) \right|$$

$$\leq \left\{ \left[ H^2 \cdot \log\det [\lambda \mathbf{I} + \mathbf{K}_t] + 2H^2 \cdot \log\left(\frac{1}{\delta}\right) \right]^{1/2} + \sqrt{\lambda} B + H \cdot \iota_{\mathrm{mis}} \sqrt{t} \right\} \cdot b_h^t(z) \qquad (\mathrm{H.2})$$

$$\leq \left\{ \left[ H^2 \cdot \log\det [\mathbf{I} + \mathbf{K}_t/\lambda] + (\lambda - 1)tH^2 + 2H^2 \cdot \log\left(\frac{1}{\delta}\right) \right]^{1/2} + \sqrt{\lambda} B + H \cdot \iota_{\mathrm{mis}} \sqrt{t} \right\} \cdot b_h^t(z)$$
$$(\mathrm{H.3})$$

$$\leq \left\{ \left[ H^2 \cdot \Gamma_{\mathbf{K}}(T, \lambda) + H^2 + 2H^2 \cdot \log\left(\frac{1}{\delta}\right) \right]^{1/2} + \sqrt{\lambda} B + H \cdot \iota_{\mathrm{mis}} \sqrt{t} \right\} \cdot b_h^t(z). \qquad (\mathrm{H.4})$$

Let $(\beta/H)^2 = 3\Gamma_{\mathbf{K}}(T, \lambda) + 3 + 6 \cdot \log\left(\frac{1}{\delta}\right) + 3\lambda \left(\frac{B}{H}\right)^2 + \iota_{\mathrm{mis}} t^2$, $\left| \phi(z)^\top (\boldsymbol{\theta}_h^* - \overline{\boldsymbol{\theta}}_h^t) \right| \leq \beta \cdot b_h^t(z)$ with probability at least $1 - \delta$. We present the last step of our proof in the following part. Equipped with Assumption 9, we are able to bound the distance between the optimal estimated expected value at time $h + 1$ with the true expected value as in the following Lemma 26, which concludes our proof:

**Lemma 26.** *For any bounded value function $V(\cdot) : \mathcal{S} \mapsto [-1, 1]$ and any $z \in \mathcal{Z}$, there exists a $\boldsymbol{\theta}_h^* \in \mathcal{H}$ such that:*

$$|\mathbb{P}_h V(z) - \langle \phi_V(z), \boldsymbol{\theta}_h^* \rangle_{\mathcal{H}}| \leq \iota_{\mathrm{mis}},$$

*where $\phi_V$ is defined in Section 3.2. The proof of Lemma 26 is a simply application of the definition of total variation distance.*

$\qquad\square$

### H.3 Proof of Lemma 14

In this section we prove Lemma 14. We need the following lemmas.

**Lemma 27** (Lemma 4.1 in Cao and Gu 2019, Zhou et al. 2020)**.** *There exist constants $C_i > 0$ such that for any $\delta \in (0, 1)$, if $B$ satisfies that*

$$C_1 m^{-1} L^{-3/2} \max\{\log^{-3/2} m, \log^{3/2}(|\mathcal{Z}|L^2/\delta)\} \leq B \leq C_2 L^{-6} (\log m)^{-3/2},$$

*then with probability at least $1 - \delta$, for all $\boldsymbol{\theta}_1$ and $\boldsymbol{\theta}_2$ satisfying $\boldsymbol{\theta}_1, \boldsymbol{\theta}_2 \in B(\boldsymbol{\theta}^{(0)}, B)$ and all $(s', z) \in \mathcal{S} \times \mathcal{Z}$, we have*

$$|f(s', z; \boldsymbol{\theta}_1) - f(s', z; \boldsymbol{\theta}_2) - \langle \phi(s', z; \boldsymbol{\theta}_2), \boldsymbol{\theta}_1 - \boldsymbol{\theta}_2 \rangle| \leq C_3 B^{4/3} m^{-1/6} L^3 \sqrt{\log m}.$$

**Lemma 28** (Lemma B.3 in Cao and Gu 2019, Zhou et al. 2020)**.** *There exist constants $C_i > 0$ such that for any $\delta \in (0, 1)$, if $B$ satisfies that*

$$C_1 m^{-1} L^{-3/2} \max\{\log^{-3/2} m, \log^{3/2}(|\mathcal{Z}|L^2/\delta)\} \leq B \leq C_2 L^{-6} (\log m)^{-3/2},$$

*then probability at least $1 - \delta$, for all $\boldsymbol{\theta}$ satisfying $\boldsymbol{\theta} \in B(\boldsymbol{\theta}^{(0)}, B)$ and all $(s', z) \in \mathcal{S} \times \mathcal{Z}$, we have $\|\phi(s'|z)\|_2 \leq C_3 \sqrt{L}$.*

*Proof of Lemma 14.* By Lemma 28 we have

$$\|\boldsymbol{\phi}_V(z)\|_2 = \left\| \sum_{s'} V(s')\boldsymbol{\phi}(s'|z) \right\|_2 \leq \sum_{s'} |V(s')| \|\boldsymbol{\phi}(s'|z)\|_2 \leq C_1 |\mathcal{S}| \sqrt{L}.$$

By the assumption of $\mathbb{P}_h(s'|z)$, we have $\boldsymbol{\theta}_h^* \in B(\boldsymbol{\theta}^{(0)}, B)$. Thus, by Lemma 27, we have with probability at least $1 - \delta$, for all $s' \in \mathcal{S}, z \in \mathcal{Z}, h \in [H]$,

$$|\mathbb{P}_h(s'|z) - \langle \boldsymbol{\phi}(s', z; \boldsymbol{\theta}^{(0)}), \boldsymbol{\theta}_h^* - \boldsymbol{\theta}^{(0)} \rangle| = |f(s', z; \boldsymbol{\theta}_h^*) - f(s', z; \boldsymbol{\theta}^{(0)}) - \langle \boldsymbol{\phi}(s', z; \boldsymbol{\theta}^{(0)}), \boldsymbol{\theta}_h^* - \boldsymbol{\theta}^{(0)} \rangle|$$
$$\leq C_2 B^{4/3} m^{-1/6} L^3 \sqrt{\log m},$$

where the equality holds by the assumption of $\mathbb{P}_h$ and $f(s', z; \boldsymbol{\theta}^{(0)}) = 0$ guaranteed by the initialization scheme, the inequality holds due to Lemma 27. Therefore, for any value function $V : \mathcal{S} \to [-1, 1]$, we have

$$|\mathbb{P}_h V(z) - \langle \boldsymbol{\phi}_V(z), \boldsymbol{\theta}_h^* - \boldsymbol{\theta}^{(0)} \rangle| = \left| \sum_{s'} V(s')\mathbb{P}_h(s'|z) - \sum_{s'} V(s')\langle \boldsymbol{\phi}(s'|z), \boldsymbol{\theta}_h^* - \boldsymbol{\theta}^{(0)} \rangle \right|$$
$$\leq \sum_{s'} |V(s')| |\mathbb{P}_h(s'|z) - \langle \boldsymbol{\phi}(s'|z), \boldsymbol{\theta}_h^* - \boldsymbol{\theta}^{(0)} \rangle|$$
$$\leq C_2 |\mathcal{S}| H B^{4/3} m^{-1/6} L^3 \sqrt{\log m},$$

where the second inequality we use the fact that $|V| \leq 1$. $\qquad\square$

## I  Implementation Details of `FIND_CCE`

Suppose that we have $Q_1, Q_2 \in \mathcal{S} \times \mathcal{A} \times \mathcal{A} \to \mathbb{R}$. Given a state $x \in \mathcal{S}$, let $P_1, P_2 \in \mathbb{R}^{|\mathcal{A}| \times |\mathcal{A}|}$ denote the matrices of Q values such that $[P_i]_{m,n} = Q_i(x, a_m, a_n)$ for $i = 1, 2$, where $a_m, a_n$ denote the $m$-th and $n$-th actions of $\mathcal{A}$. Suppose the CCE of $Q_1, Q_2$ given $x$ is denoted by a matrix $\sigma \in \mathbb{R}^{|\mathcal{A}| \times |\mathcal{A}|}$, where $\sigma_{m,n}$ denotes the probability of selecting $m$-th and $n$-th actions. Then $\sigma$ satisfies the following two groups of constraints:

- Since $\sigma$ is a probability matrix, then we have

$$\forall 1 \leq m, n \leq |\mathcal{A}|,\ 0 \leq \sigma_{m,n} \leq 1, \tag{I.1}$$

$$\sum_{i=1}^{|\mathcal{A}|} \sum_{j=1}^{|\mathcal{A}|} \sigma_{i,j} = 1. \tag{I.2}$$

- To satisfy (3.1), we have

$$\forall 1 \leq m \leq |\mathcal{A}|,\ \sum_{i=1}^{|\mathcal{A}|} \sum_{j=1}^{|\mathcal{A}|} \sigma_{i,j}[P_1]_{i,j} \geq \sum_{j=1}^{|\mathcal{A}|} [P_1]_{m,j} \sum_{i=1}^{|\mathcal{A}|} \sigma_{i,j}.$$

$$\Leftrightarrow \forall 1 \leq m \leq |\mathcal{A}|,\ \sum_{i=1}^{|\mathcal{A}|} \sum_{j=1}^{|\mathcal{A}|} \sigma_{i,j}([P_1]_{m,j} - [P_1]_{i,j}) \leq 0 \tag{I.3}$$

- To satisfy (3.2), we have

$$\forall 1 \leq n \leq |\mathcal{A}|,\ \sum_{i=1}^{|\mathcal{A}|} \sum_{j=1}^{|\mathcal{A}|} \sigma_{i,j}[P_2]_{i,j} \leq \sum_{i=1}^{|\mathcal{A}|} [P_2]_{i,n} \sum_{j=1}^{|\mathcal{A}|} \sigma_{i,j}.$$

$$\Leftrightarrow \forall 1 \leq n \leq |\mathcal{A}|,\ \sum_{i=1}^{|\mathcal{A}|} \sum_{j=1}^{|\mathcal{A}|} \sigma_{i,j}([P_2]_{i,j} - [P_2]_{i,n}) \leq 0. \tag{I.4}$$

There are total $|\mathcal{A}|^2$ number of unknown variables ($\sigma_{m,n}$) with 1 equality constraint and $|\mathcal{A}|^2 + 2|\mathcal{A}|$ number of inequality constraints. The above linear system can be converted into a standard linear programming problem with $2|\mathcal{A}|^2$ number of variables $\sigma_{m,n}, \widehat{\sigma}_{m,n}, 1 \leq m, n \leq |\mathcal{A}|$, such that

$$\max_{\sigma_{m,n}, \widehat{\sigma}_{m,n}} 0$$

$$\sigma_{m,n} \geq 0$$
$$\widehat{\sigma}_{m,n} \geq 0$$
$$\sigma_{m,n} + \widehat{\sigma}_{m,n} \leq 1$$
$$-\sigma_{m,n} - \widehat{\sigma}_{m,n} \leq -1$$
$$\sum_{i=1}^{|\mathcal{A}|} \sum_{j=1}^{|\mathcal{A}|} \sigma_{i,j} \leq 1$$
$$-\sum_{i=1}^{|\mathcal{A}|} \sum_{j=1}^{|\mathcal{A}|} \sigma_{i,j} \leq -1$$
$$\sum_{i=1}^{|\mathcal{A}|} \sum_{j=1}^{|\mathcal{A}|} \sigma_{i,j}([P_1]_{m,j} - [P_1]_{i,j}) \leq 0$$
$$\sum_{i=1}^{|\mathcal{A}|} \sum_{j=1}^{|\mathcal{A}|} \sigma_{i,j}([P_2]_{i,j} - [P_2]_{i,n}) \leq 0$$

The above linear system can be solved by Karmarkar's algorithm [Karmarkar, 1984] within $\widetilde{O}(|\mathcal{A}|^7)$ time complexity, or with Stochastic Central Path Method [Cohen et al., 2021] within $\widetilde{O}(|\mathcal{A}|^{2w})$ time complexity, where $w = 2.373...$ is the matrix multiplication constant.