# OpenReview forum: "Learning Two-Player Markov Games: Neural Function Approximation and Correlated Equilibrium"
_NeurIPS.cc/2022/Conference — NeurIPS 2022 Accept_

### Official Review · Reviewer_3y5m · 2022-06-28

**Rating:** 6
**Confidence:** 4
**Soundness:** 3 good
**Presentation:** 2 fair
**Contribution:** 2 fair

**Summary:**

This paper proposes a new algorithm (KernelCCE-VTR) for solving two-player zero-sum Markov games. KernelCCE-VTR can be applied to a general class of transiton model; it only requires the transition dynamic belongs to some RKHS. The regret upper bound is provided and it matches the best-known complexity. Some potential extensions are also discussed: The case of changing the exploration bonus to Bernstein-type, this paper proposes KernelCCE-VTR+ algorithm. Also, when the transition dynamic belongs to some RKHS up to some misspecification error, KernelCCE-VTR can still reach the Nash equilibrium up to such error.

**Questions:**

1. Some typos and unclear notations:
    * In Algorithm 1 (Line 197), "for step $h=1,2,\dots,T$". $T$ should be $H$.
    * I am feeling $b_h^t$ has been used as an action. But there are also many $\bar{b}_{h}^t$ (bonus function) in this paper. It is a little confusing sometimes.
    * In Appendix D (Line 589), $\bar{\Lambda}\_{h}^{t}$ should have an inverse. There must be more typos when evaluating $\bar{b}_{h}^t$ in Lemma 11 (Line 483); please fix.

2. Undefined notation and a missing step:
    * In Appendix D (Line 574), "...for $h=H+1$, where ${\rm III} = 0$". Actually, for $h=H+1$, $\bar{V}^t_h(x_h^t)$ and  ${V}^{\ast,\nu_h^t}_h(x_h^t)$ are not defined. The induction may need to start from $h=H$ but I am not clear why ${\rm III}=0$ holds in this case. Could you provide more explanation for this step?

3. It would be more helpful if you could highlight more differences between this work and [Yang et al. (2020)]. For example, what makes it difficult to deal with the two-player zero-sum MGs? Also, [Yang et al. (2020)] has a strong assumption; they require the images of $Q^\ast_h$ of Bellman operator to be included in the RKHS-norm ball (in their Assumption 4.1). I have not noticed such requirement in this KernelCCE paper; is it automatically solved becaused of the different setting or a different proof stratagy?

**Limitations:**

Since this is a purely theoretical work, there are no any negative impact.

**Strengths And Weaknesses:**

***Strengths:***
* Most of steps in the proof are well explained. The regret bound is sufficiently tight; it matches the best-known result with respect to the number of episodes $T$. The extension part is complete and gives a clear explaination about what will happen when the approximation assumption is not satisfied.
* This paper studies an important approximation setting; it will allow future algorithms to be applied to general state and action spaces rather than the naive tabular MDP.

***Weaknesses:***
1. Some typos and unclear notations. I list them in the Questions section.
2. One step in the proof is not clear to me; possibly wrong (see the next section Q2). This is also my major concern.
3. The algorithm and proof strucutre are similar to the KOVI algorithm from [Yang et al. (2020)]. Generally, it is not a big problem for me since this paper is solving the two-player MG while [Yang et al. (2020)] is solving the standard single-agent optimal control. But this paper only focuses on the zero-sum case, which makes two players symmetric, so in most of cases, it seems to be sufficient to deal with one-side of player.

***Reference:***

[Yang et al. (2020)] Yang, Z., Jin, C., Wang, Z., Wang, M., & Jordan, M. I. (2020). On function approximation in reinforcement learning: Optimism in the face of large state spaces. arXiv preprint arXiv:2011.04622.

---

> ### Author Response · Authors · 2022-08-02
> **To Reviewer 3y5m**
>
> Thanks for recognizing the contributions of our work!
>
> ---
>
> **Q1** Some typos and unclear notations.
>
> **A1** Thanks for pointing them out. We have already fixed them.
>
> ---
>
> **Q2** Major concern on clarifying the proof in line 574. In special, what are the definitions of $\overline{V_{H+1}}, V_{H+1}^{*, \nu^t}$?
>
> **A2** In our algorithm, we define $\overline{V_{H+1}^t} = 0$ and $\underline{V_{H+1}^t} = 0$. For any policy $\pi, \nu$, the value function $V_{h}^{\pi, \nu}(x)$ is defined as the expected accumulated reward from step $h$. Therefore we also have $V_{H+1}^{\pi, \nu}(x) = 0$ and $V_{H+1}^{*, \nu^t} = 0$, since there is no more future steps starting from $h = H+1$. With these definitions, $\mbox{III} = 0$ in line 574 (line 666 in the revised version) holds automatically. We have added the corresponding definitions. We hope this resolves your concern.
>
> ---
>
> **Q3** It would be more helpful if you could highlight more differences between this work and [Yang et al. (2020)]. For example, what makes it difficult to deal with the two-player zero-sum MGs?
>
> **A3** The differences between our work and Yang et al. (2020) are as follows. First, the two-player setting requires us to consider the joint policy of two agents, while the single-player setting (Yang et al., 2020) only needs to consider the policy of a single agent. Second, our considered kernel assumption is made on the transition dynamic, while the similar assumption in Yang et al. (2020) is made over value functions. Such a difference also makes our algorithms different. Finally, our work proposes a more refined KernelCCE+ algorithm which adopts a Bernstein-type exploration bonus and enjoys a better regret guarantee. Yang et al. (2020) does not consider Berstein-type bonus.
>
> ---
>
> **Q4** Yang et al. (2020) require the images of $Q^*$ of Bellman operator to be included in the RKHS-norm ball. Does KernelCCE require that? Is it automatically solved becaused of the different setting or a different proof stratagy?
>
> **A4** KernelCCE also requires a similar assumption. We assume that for the feature mapping $\phi(s’|z)$, for any $V: \mathcal{S} \rightarrow [-1, 1]$ and $z \in \mathcal{Z}$, $\|\phi_{V}(z)\|_{\mathcal{H}} \leq 1$ (above line 169, line 201 in the revised version).
>
> ---

---

> > ### Comment · Reviewer_3y5m · 2022-08-02
> > **Follow-up questions**
> >
> > I agree that initializing $\bar{V}\_{H+1}(x)=\underline{V}\_{H+1}(x)=0$ for all $x$ can fix this problem, but it may cause some other issues I am concerned:
> >
> > ***Q2.1*** After setting $\bar{V}\_{H+1}=0$, the Eq.(D7), Line 576, for $h=H$, becomes
> > $\bar{Q}\_H^t(z) - {Q}\_H^{\ast, \nu\_H^t}(z)=0$;
> > because $\phi_{\bar{V}\_{H+1}^t}=0$ and $\bar{b}\_{H+1}^t(z)=0$. Does it mean that ${Q}\_H^{\ast, \nu\_H^t}(z)$ can be any value less than $H$?
> >
> > ***Q2.2*** After setting $\bar{V}\_{H+1}=\underline{V}\_{H+1}=0$ for all $x$, according to the definition of $\bar{Q}\_h^t$ in (4.3), Line 217, $\bar{Q}_H^t = \underline{Q}_H^t=r_H$. But $\bar{Q}_H^t = \underline{Q}_H^t=0$ by (C.4), Line 547. Should these definitions be equivalent? Which one is used in the proposed algorithm? Also, I don't understand the curly bracket used in (C.4); what does $\prod\_{[-H,H]}\\{\dots, H-h+1\\}$ mean here?
> >
> > ***Q2.3*** When $\bar{Q}_H^t = \underline{Q}_H^t=0$, it will further implie $\bar{V}\_{H}=\underline{V}\_{H}=0$ by Line 193, Algorithm 1; then $\bar{V}\_{h}=\underline{V}\_{h}$ for all $h\in [H]$ by induction. I don't think this is normal; so I put my derivation below. Just feel free to point out my mistakes.
> >
> > Step 1 (base case): $\bar{V}\_{H+1}=\underline{V}\_{H+1}=0$ for all $x$ by initialization.
> >
> > Step 2: Assume  $\bar{V}\_{h+1}=\underline{V}\_{h+1}=0$ for all $x$. Then
> > * $f_{\bar{V}\_{h+1}}=f_{\underline{V}\_{h+1}}=0$ by (C.1) Line 544
> > * $\bar{b}\_h=\underline{b}\_h=0$ by (C.3), Line 546.
> >
> > Then $\bar{Q}\_h=\underline{Q}\_h=0$ (it implies $\bar{V}\_h=\underline{V}\_h=0$).

---

> > > ### Author Response · Authors · 2022-08-03
> > > **Reply to your follow-up questions**
> > >
> > > Thank you for your follow-up questions. We answer them as follows.
> > >
> > > **Re: Q2.1**
> > > Yes, $Q_H^{ \star,\nu_H^t}$ can take some value between -H and H. It is because $Q_H^{\star, \nu_H^t}(z) = {\overline Q_h^t}(z)$ and ${\overline Q_h^t}(z) \in [-H, H]$.
> > >
> > > **Re: Q2.2**
> > > We are sorry for the typo here. The update rule for the neural network case in (C.4) should be
> > > ${\overline Q_h^t}(z)=\Pi_{[-H,H]}[r_h(z) + f_{\overline V_{h+1}^t}(z;{\overline \theta_h^t}) + \beta \cdot {\overline w_h^t}(z)]$.
> > > We missed $r_h$ here. The corrected (C.4) is now identical to (4.3) and (4.4), and both of them suggest that ${\overline Q_H^t}(z) = r_H(z)$. $\Pi_{[-H,H]}[x]$ means to clip value $x$ to the interval $[-H, H]$. We also fixed it in the revised paper. Thank you for catching this typo!
> > >
> > > **Re: Q2.3**
> > > After we fixed the missing reward function $r_h$ in (C.4), there is no issue in the induction anymore.
> > >
> > > Please let us know if you have any other questions. Thanks.

---

> > > > ### Comment · Reviewer_3y5m · 2022-08-03
> > > > **Follow-up questions cont.**
> > > >
> > > > Thanks for your quick response! Just a quick follow-up question. According to the updated (C.4) and the initialization $\bar{V}\_{H+1}=\underline{V}\_{H+1}=0$ for all $(x,a,b)$, the value of $\bar{Q}_H(x,a,b)$ will be exactly $r_H(x,a,b)$ for all $(x,a,b)$. Does it mean that it is assume the reward function $r_H$ is exactly given? Is it normal in other literatures?

---

> > > > > ### Author Response · Authors · 2022-08-04
> > > > > **Reply to 'Follow-up questions cont. '**
> > > > >
> > > > > Yes, in this paper, we assume that the reward functions $r_1,\dots, r_H$ are given. In general, RL with function approximation requires knowledge about the reward functions. It is very common in the RL literature, e.g., UCRL-VTR[1, 2] and Nash-UCRL[3].
> > > > >
> > > > > [1] Ayoub, Alex, et al. "Model-based reinforcement learning with value-targeted regression." International Conference on Machine Learning. PMLR, 2020.
> > > > >
> > > > > [2] Jia, Zeyu, et al. "Model-based reinforcement learning with value-targeted regression." Learning for Dynamics and Control. PMLR, 2020.
> > > > >
> > > > > [3] Chen, Zixiang, Dongruo Zhou, and Quanquan Gu. "Almost Optimal Algorithms for Two-player Zero-Sum Linear Mixture Markov Games." International Conference on Algorithmic Learning Theory. PMLR, 2022.

---

> > > > > > ### Comment · Reviewer_3y5m · 2022-08-04
> > > > > > **Response**
> > > > > >
> > > > > > Thanks for your response! I read the given references and agree that setting the value of $\bar{Q}_H(x,a,b)$ as $r_H(x,a,b)$ has also been applied in other literatures. Now all my concerns are resolved and I decide to raise my score from 5 to 6.

---

> > > > > > > ### Author Response · Authors · 2022-08-04
> > > > > > > **Thank you for raising the score!**
> > > > > > >
> > > > > > > We're glad that we have resolved your questions and concerns.

---

### Official Review · Reviewer_nHpk · 2022-07-10

**Rating:** 7
**Confidence:** 3
**Soundness:** 3 good
**Presentation:** 3 good
**Contribution:** 3 good

**Summary:**

This paper considers two-player, zero-sum Markov games under a kernel mixture MG model where the transition kernel is assumed to be an element in an RKHS. This implies that the expectations of the state and state-action value functions are also elements in an RKHS. The authors propose KernelCCE-VTR to learn the equilibrium of the game. This is achieved by first estimating the state value functions using kernel ridge regression. By relating the regression problem to a linear bandit setting, the authors can derive tractable confidence bounds used to construct optimistic / pessimistic estimates of the state-action value functions. Given the state-action value functions, a CCE can then be computed, and a state value function can be recovered from their expectation under the CCE. The authors are able to prove a regret bound on the duality gap of $\tilde{\mathcal{O}}(H^{3/2} \sqrt{T})$. The authors also study the setting where the transition kernel "approximately" belongs to an RKHS (the error contributing a linear regret to the bound).

**Questions:**

- The abstract and conclusion both mention the use/demonstration of neural networks, however, I did not see it discussed in any significant detail (or any pointers to the appendix). Please remove these claims if you do not explore/discuss them.
- Sec 2 contrasts this work with Qui et al. (2021) where it is assumed that the expectation of the value function is in an RKHS. However, my understanding of this work is that the transitional kernel being in an RKHS implies the value function lives in an RKHS (paragraph 2 of Kernel mixture MG) making that previous work more general. Can you comment?
- Please describe how FIND_CCE works briefly and provide a bit more detail on the game/equilibrium being solved. For example, when the state-action value "Q" functions are calculated via Eq. 4.3, are the "Q" values computed for every state in the MDP? And then is a CCE computed for every possible initial state in the game? I found the single "dot" notation to be a bit confusing given that the Q value depends on state, player 1 action, player 2 action.
- Is the initial state from line 3 of Alg 1 used anywhere in lines 4-8 or can it be moved below to right above line 9?
- I might have missed it, but I did not know that superscripts indicated episodes (e.g., $x^{\tau}_{h+1}$) until I figured that out myself. Please define that notation if you don't already.

**Limitations:**

I don't see any negative societal impact.

**Strengths And Weaknesses:**

The work appears to be original and is mostly clear given the technical difficulty. Learning two-player, zero-sum Markov games with function approximation is an interesting and challenging problem. This work

---

> ### Author Response · Authors · 2022-08-02
> **To Reviewer nHpk**
>
> Thanks for your supportive comments!
>
> ---
>
> **Q1** The abstract and conclusion both mention the use/demonstration of neural networks, however, I did not see it discussed in any significant detail (or any pointers to the appendix). Please remove these claims if you do not explore/discuss them.
>
> **A1** The neural network approximation section is Section C. We will add a pointer to Section 3 in the introduction.
>
> ---
>
> **Q2** Sec 2 contrasts this work with Qui et al. (2021) where it is assumed that the expectation of the value function is in an RKHS. However, my understanding of this work is that the transitional kernel being in an RKHS implies the value function lives in an RKHS (paragraph 2 of Kernel mixture MG) making that previous work more general. Can you comment?
>
> **A2** It is not true. Qiu et al. (2021) considered a kernel MG setting, which implies that the value function belongs to an RKHS space. In contrast, our kernel mixture MG assumption implies the expectation of any value function belongs to an RKHS (above line 166, line 196 in the revised paper), but not the value function itself. Thus, these two settings are different, and the kernel MG setting can not cover our setting.
>
> ---
>
> **Q3** Please describe how FIND_CCE works briefly and provide a bit more detail on the game/equilibrium being solved.
>
> **A3** The detail of FIND_CCE is above line 141 (line 165 to 171 in the revised version). FIND_CCE takes two Q functions $\overline{Q}, \underline{Q}$ and an state $x$ as its input, then it outputs a distribution $\sigma(x)$ over $\mathcal{A} \times \mathcal{A}$. To solve CCE, it suffices to set the joint probability of the distribution $\sigma(x)$ as an unknown matrix $P \in [0,1]^{|\mathcal{A}| \times |\mathcal{A}|}$ and solve out $P$ that satisfy several inequalities decided by the equations above line 141. Such a solving procedure can be achieved efficiently via linear programming.
>
> ---
>
> **Q4** When the state-action value "Q" functions are calculated via Eq. 4.3, are the "Q" values computed for every state in the MDP? And then is a CCE computed for every possible initial state in the game?
>
> **A4** You are right, since our algorithm needs to calculate the weighted kernel function $k_{\overline{V_{h+1}^i}, \overline{V_{h+1}^t}}$, which requires to know $\overline{V}_{h+1}^t(x)$ for every state $x$. This further implies that we should compute corresponding Q values for every state, as well as the CCE.
>
> ---
>
> **Q5** I found the single "dot" notation to be a bit confusing given that the Q value depends on state, player 1 action, player 2 action.
>
> **A5** Thanks for the suggestion. We have used $Q(\cdot, \cdot, \cdot)$ to replace $Q(\cdot)$ to emphasize that the Q function takes a triplet $(x,a,b)$ as input.
>
> ---
>
> **Q6** Is the initial state from line 3 of Alg 1 used anywhere in lines 4-8 or can it be moved below to right above line 9?
>
> **A6** It can be moved to above line 9.
>
> ---
>
> **Q7** I might have missed it, but I did not know that superscripts indicated episodes until I figured that out myself. Please define that notation if you don't already.
>
> **A7** Thanks for your suggestion. We have specified that the superscripts indicate episodes in Section 3.1.
>
> ---

---

> > ### Comment · Reviewer_nHpk · 2022-08-03
> > **Computational efficiency**
> >
> > Thank you for all your explanations. I only have one follow-up question.
> >
> > Q3/A3: Thank you for your comment. I'm also interested in the computational complexity of FIND_CCE. In general, it would be good if the section on Computational efficiency (line 261 in new supp) included specific results in big-O notation.

---

> > > ### Author Response · Authors · 2022-08-04
> > > **Reply to 'Computational efficiency'**
> > >
> > > Thank you for your further question. We’re glad that we have addressed your previous questions.
> > >
> > > Suppose that we have $Q_1, Q_2 \in \mathcal{S} \times \mathcal{A} \times \mathcal{A} \rightarrow \mathbb{R}$. Given a state $x \in \mathcal{S}$, let $P_1, P_2 \in \mathbb{R}^{|\mathcal{A}| \times |\mathcal{A}|}$ denote the matrices of Q values such that
> > >
> > > $[P_i]_{m,n} = Q_i(x, a_m, a_n)$
> > >
> > > for $i = 1,2$, where $a_m, a_n$ denote the $m$-th and $n$-th actions of $\mathcal{A}$. Suppose the CCE of $Q_1, Q_2$ given $x$ is denoted by a matrix $\sigma \in \mathbb{R}^{|\mathcal{A}| \times |\mathcal{A}|}$, where $\sigma_{m,n}$ denotes the probability of selecting $m$-th and $n$-th actions. Then $\sigma$ satisfies the following two groups of constraints:
> > >
> > >
> > > Since $\sigma$ is a probability matrix, then we have
> > >
> > >  $$\forall 1 \leq m, n \leq |\mathcal{A}|,0 \leq \sigma_{m,n} \leq 1, \sum_{i=1}^{|\mathcal{A}|}\sum_{j=1}^{|\mathcal{A}|} \sigma_{i,j} = 1.$$
> > >
> > >
> > > To satisfy (3.1), we have
> > >
> > > \begin{align}
> > > \\forall 1 \\leq m \\leq |\\mathcal{A}|,\\Sigma_{i=1}^{|\\mathcal{A}|}\\Sigma_{j=1}^{|\\mathcal{A}|}\\sigma_{i,j}[P_1]_{i,j}  \geq
> > > \end{align}
> > >
> > > \begin{align}
> > > {\\Sigma_{i=1}^{|\\mathcal{A}|} \\Sigma_{j=1}^{|\\mathcal{A}|}\\sigma_{i,j} [P_1]_{m,j}}
> > > \end{align}
> > >
> > >
> > > To satisfy (3.2), we have
> > >     \begin{align}
> > >         \forall 1 \leq n \leq |\mathcal{A}|,\ \sum_{i=1}^{|\mathcal{A}|}\sum_{j=1}^{|\mathcal{A}|}\sigma_{i,j}[P_2]_{i,j} \leq
> > >     \end{align}
> > >
> > > \begin{align}
> > > \sum_{i=1}^{|\mathcal{A}|} \sum_{j=1}^{|\mathcal{A}|} \sigma_{i,j} [P_2]_{i,n}
> > > \end{align}
> > >
> > >
> > > The above linear system can be solved by Karmarkar's algorithm [1] with $O(|\mathcal{A}|^7)$ time complexity or by [2] with $O(|\mathcal{A}|^{2w})$ time complexity, where $w = 2.373…$ is the matrix multiplication constant.
> > >
> > > [1] Karmarkar, Narendra. "A new polynomial-time algorithm for linear programming." Proceedings of the sixteenth annual ACM symposium on Theory of computing. 1984.
> > >
> > > [2] Cohen, Michael B., Yin Tat Lee, and Zhao Song. "Solving linear programs in the current matrix multiplication time." Journal of the ACM (JACM) 68.1 (2021): 1-39.
> > >
> > > For a more detailed calculation of the time complexity for solving the FIND_CCE, please refer to Appendix I in the revised paper (we just updated it).

---

> > > > ### Comment · Reviewer_nHpk · 2022-08-07
> > > > **FIND_CCE Complexity**
> > > >
> > > > Thank you for adding this detail to the appendix. I maintain my score.

---

### Official Review · Reviewer_QrN9 · 2022-07-10

**Rating:** 6
**Confidence:** 4
**Soundness:** 3 good
**Presentation:** 3 good
**Contribution:** 2 fair

**Summary:**

This work proposes a KernelCCE-VTR algorithm for finding the Nash equilibrium of two-player zero-sum MGs. The algorithm uses kernel function approximation in RKHS to approximate the transition kernel and optimal value function. The algorithm design leverages several techniques from the existing literature, including value-targeted regression, confidence set design as in UCR, CCE solver, etc. These techniques enable the authors to adapt the results from the linear setting to the nonlinear RKHS regime and achieve computation and statistical efficiency.

**Questions:**

Overall this paper develops an efficient algorithm for zero-sum games and provides a comprehensive complexity analysis.

1. Regarding the CCE definition, is the joint policy $\sigma$ supposed to be a product policy?

2. How does this algorithm compare to other value-based algorithms for zero-sum games? For example, the V-learning algorithm in https://openreview.net/pdf?id=Bx-evj5k6x9 (this seems to be a missing related work). One difference is that V-learning requires keeping a V table while the proposed algorithm requires keeping a Q table.

**Limitations:**

Yes

**Strengths And Weaknesses:**

Strength

This paper provides a computation simple and statistical efficient algorithm for solving zero-sum MGs.

Weakness

The algorithm design is based on a combination of many existing techniques; The kernel approximation setting is of limited novelty.

---

> ### Author Response · Authors · 2022-08-02
> **To Reviewer QrN9**
>
> Thank you for recognizing the contributions of our paper!
>
> ---
>
> **Q1** The algorithm design is based on a combination of many existing techniques; The kernel approximation setting is of limited novelty.
>
> **A1** We would like to highlight that the kernel approximation setting is new compared with existing works such as Yang et al. (2020) in the following aspects. Existing works make kernel assumptions on the value function class (Yang et al., 2020), while our assumption is directly made on the transition kernel itself. Such an assumption requires us to consider the weighted kernel function class (Definition 1) and to develop new algorithms using the weighted kernel function class. To our knowledge, both of them are firstly considered in our work.
>
> ---
>
> **Q2** Regarding the CCE definition, is the joint policy supposed to be a product policy?
>
> **A2** No, the joint policy is not necessary to be a product policy. From the definition of CCE (line 141, line 165 in the revised paper), the CCE is a joint policy over the product of action spaces $\mathcal{A}$. When $\sigma$ can be written as a product of two policies over action space $\mathcal{A}$, it is an Nash equilibrium (Xie et al., 2020).
>
> ---
>
> **Q3** How does this algorithm compare to other value-based algorithms for zero-sum games? For example, the V-learning algorithm in https://openreview.net/pdf?id=Bx-evj5k6x9 (this seems to be a missing related work). One difference is that V-learning requires keeping a V table while the proposed algorithm requires keeping a Q table.
>
> **A3** Thanks for pointing out this related reference. After careful checking the V-learning algorithm in Jin et al. (2021) is designed for tabular MGs, while our algorithm is for kernel mixture MGs. Moreover, their algorithm is more like an ‘online’ algorithm where each agent executes the policy of their own and does not take into account other agents’ actions. Our algorithm is more like an ‘offline’ algorithm while it needs a central player to design policies for each agent. Besides, our algorithm actually also only needs to keep a V-table, since our Q function $\overline{Q_h^t}$ can be computed directly based on $r_h$ and $k_{\overline{V_{h+1}^i}, \overline{V_{h+1}^t}}$ (Eq 4.4), which only requires us to know $\overline{V}_{h+1}^i$ beforehand. We will add it as a reference and add a comment in the revised manuscript.
>
> ---

---

### Official Review · Reviewer_ExxC · 2022-07-12

**Rating:** 7
**Confidence:** 2
**Soundness:** 3 good
**Presentation:** 3 good
**Contribution:** 3 good

**Summary:**

The paper studies learning zero-sum Markov Games with kernel function approximation. Specifically, they study kernel mixture Markov Games, which extend the linear mixture MDPs to the kernelized setting. They propose KernelCCE-VTR algorithm based on value-target regression and establish regret bounds in terms of the effective dimension of the kernel. The authors further improve the algorithm with a Bernstein-type bonus and also study the regret bound under model misspecification.


**Questions:**

N/A

**Strengths And Weaknesses:**

The paper is a novel combination of OFU explorations in zero-sum MGs, Value-targeted iteration, and kernelized version of linear mixture MDPs. The paper provides sound and elegant regret bounds in terms of the effective dimension of the kernel. The algorithm is practical since it does not require access to the possibly infinite-dimensional feature map. The paper is clearly written.

---

> ### Author Response · Authors · 2022-08-02
> **Reply to Reviewer ExxC**
>
> Thanks for your positive comments!

---

### Meta-Review · Area_Chair_CbZk · 2022-08-27

**Recommendation:** Accept
**Confidence:** Less certain

**Metareview:**

This paper considers the problem of online reinforcement learning in two-player zero-sum Markov games. They consider a class of Markov games called kernel mixture Markov games, which extend the linear mixture MDP setting considered in prior work to the RKHS setting. The main result is to propose and algorithm called KernelCCE-VTR, which uses the principle of value-targeted regression to achieve regret bounds that scale with the effective dimension of the kernel. The authors also provide an improved variant of the algorithm based on Bernstein-type confidence bonuses.

The reviewers found the setting to be important and found the paper to be well-written, and agreed that the main results are technically challenging and likely to be a useful starting point for future work on learning Markov games with function approximation. The main issue raised by the reviewers is that the algorithm design and analysis appears to be based on a combination of well-known existing techniques for simpler settings---for the final revision, the paper can be strengthened by more strongly advocating for the novelty required in combining these techniques.


**Award:**

No

---

### Decision · Program_Chairs · 2022-09-14

Accept